# Steering Diffusion Models Towards Credible Content Recommendation

**Zhuo Cai**[1]   **Shoujin Wang**[1*]   **Jin Li**[1]   **Peilin Zhou**[2]   **Victor W. Chu**[1]   **Fang Chen**[1]
**Tianqing Zhu**[3]   **Charu C. Aggarwal**[4]
[1]University of Technology Sydney   [2]Hong Kong University of Science and Technology (Guangzhou)
[3]City University of Macau   [4]IBM T. J. Watson Research Center

## Abstract

In recent years, diffusion models (DMs) have achieved remarkable success in recommender systems (RSs), owing to their strong capacity to model the complex distributions of item content and user behaviors. Despite their effectiveness, existing methods pose the danger of generating uncredible content recommendations (e.g., fake news, misinformation) that may significantly harm social well-being, as they primarily emphasize recommendation accuracy while neglecting the credibility of the recommended content. To address this issue, in this paper, we propose `Disco`, a novel method to steer diffusion models towards credible content recommendation. Specifically, we design a novel disentangled diffusion model to mitigate the harmful influence of uncredible content on the generation process while preserving high recommendation accuracy. This is achieved by reformulating the diffusion objective to encourage generation conditioned on preference-related signals while discouraging generation conditioned on uncredible content-related signals. In addition, to further improve the recommendation credibility, we design a progressively enhanced credible subspace projection that suppresses uncredible content by projecting diffusion targets into the null space of uncredible content. Extensive experiments on real-world datasets demonstrate the effectiveness of `Disco` in terms of both accurate and credible content recommendations.

## 1 Introduction

Diffusion models (DMs) have achieved remarkable advances across multiple domains, such as image synthesis (Ho et al., 2020; Dhariwal & Nichol, 2021) and language/text generation (Li et al., 2022; Lovelace et al., 2023). Owing to their strong capability in modeling complex data distributions of user behaviors and diverse item content types (e.g., text, images, and videos), DMs have attracted growing attention in recommender systems (RSs), thereby further driving the innovations in this field (Wang et al., 2023d; Yang et al., 2023b; Liu et al., 2025a).

DM-based recommendation methods generally adopt a diffusion-then-denoising paradigm to model the distributions of users' behaviors and then generate items they are likely to engage with (Yang et al., 2023b; Liu et al., 2025a; Li et al., 2023). Figure 1 illustrates the overall process of existing DM-based methods. A sequence encoder (e.g., Transformer

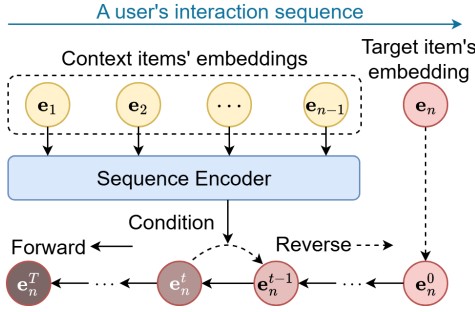

Figure 1: A general paradigm of DM-based sequential recommendation methods. The condition and diffusion target (i.e., target item's embedding) are two core components in DM-based methods.

(Vaswani et al., 2017), GRU (Chung et al., 2014)) is first employed to encode the embeddings of the first $(n-1)$ context items interacted by a user into a unified representation of the user's overall preference, which serve as the condition in the reverse stage. The $n$-th item (i.e., the last item

*Corresponding author: Shoujin.Wang@uts.edu.au

interacted with by the user) is then treated as the diffusion target. In the forward process, noise is gradually added to the diffusion target according to a predefined noise schedule. In the reverse process, the sequence encoder's output serves as the condition to guide the generation of the item embeddings that reflect users' genuine preferences (Li et al., 2025b; Cai et al., 2025).

Although DM-based recommendation methods have achieved remarkable success, they often overlook a critical real-world concern: **the risk of generating uncredible content recommendations that can harm social well-being**. For example, news RSs powered by DMs may produce uncredible recommendations containing fake news (Wang et al., 2022b; 2024a), as these methods typically overlook the credibility of recommended content. Recommending such uncredible content to users not only diminishes users' experience but also poses substantial societal risks. For instance, during the COVID-19 pandemic, news RSs (e.g., online news portals) Wang et al. (2023b); Zhao et al. (2023) were shown to amplify the spread of health-related fake news (e.g., false cures and vaccine conspiracy theories), which fueled public confusion and resistance to medical treatments (Loomba et al., 2021). Hence, from a societal consideration perspective Wang et al. (2024b); Li et al. (2024); Wang et al. (2022a), it is crucial to develop DM tailored for credible content recommendations.

To achieve this goal, we first conduct both empirical and theoretical analysis to investigate why existing DM-based recommendation methods risk generating uncredible content recommendations. The detailed analysis can be found in Appendix C. Our analysis reveals two key factors: **(1) uncredible condition**, which arises when a user has previously interacted with uncredible items (i.e., there are uncrdible items in context items); and **(2) uncredible diffusion target**, which occurs when the target item itself is uncredible. In this paper, an uncredible item refers to an item containing uncredible content, such as fake news and misinformation. These two factors jointly lead existing DM-based methods to generate recommendations that may contain uncredible content.

Hence, to develop a diffusion model tailed for credible content recommendation, it is necessary to carefully address these two factors. A straightforward solution is to remove uncredible items from both context items and diffusion targets, or to apply recommendation unlearning methods (Chen et al., 2022; Zhang et al., 2024b; Li et al., 2026) to erase their impacts. This can ensure the credibility of both the condition and the diffusion target. However, such an approach raises a critical issue: uncredible items may still reflect users' genuine preferences. For instance, if a user reads a sports-related fake news article, it may signal this user's underlying interest in sports topics. In this case, removing the uncredible items entirely would severely harm recommendation accuracy. Thus, **the first challenge lies in how to mitigate the negative impact of uncredible content without sacrificing recommendation accuracy.** An alternative solution is to retain users' preference-related information while removing only the uncredible aspects of content items. However, this approach requires rich supervision (i.e., credibility labels) to ensure accurate and comprehensive removal. In practice, only a small portion of items are verified and labeled. For example, on news portals, some articles may be flagged as fake, while many others remain unverified. Hence, **the second challenge is how to develop a diffusion model that can effectively handle both known and unknown uncredible content under limited label availability.** Existing methods for credible content recommendation (Wang et al., 2022b; 2024a; Ma et al., 2025) typically assume that all uncredible items are fully labeled, which rarely holds in real-world scenarios, leading to suboptimal performance.

To overcome these two challenges and steer diffusion models towards credible content recommendation, we propose a novel framework called `Disco`. Specifically, **to address the uncredible condition and the challenge of preserving recommendation accuracy**, we design a disentangled diffusion model that separates uncredible content from users' preference-related information in items' embeddings. With this disentanglement, the generation process becomes free from the harmful influence of uncredible items, while still preserving high recommendation accuracy by retaining users' genuine preference-related information. In addition, instead of incorporating auxiliary disentanglement networks and constraints which often introduce extra computation cost (Wang et al., 2023e; Qi et al., 2024; Wang et al., 2022b; Ma et al., 2025), the diffusion model itself can serve as an effective disentangler with proper adjustments. Specifically, we reformulate the diffusion objective to encourage the model generation guided by preference-related signals (i..e, signals indicating users' preference, such as content topics), while discouraging the generation conditioned on uncredible content-related signals (i.e., uncredible signals such as inaccurate and misleading information). **To address the uncredible diffusion target**, we introduce a credible subspace projection module to project diffusion targets into the null space of uncredible content features, which maximally excludes uncredible information. **To overcome the challenge of limited labeled data**,

the uncredible content features are progressively enhanced by detecting and incorporating potential uncredible items, making the null space projection progressively more accurate and comprehensive. Comprehensive experiments verify the effectiveness of `Disco` in terms of delivering both accurate and credible content recommendations.

In summary, our contributions can be concluded as follows:

- We propose `Disco`, a novel diffusion model tailored for credible content recommendation. To the best of our knowledge, Disco is the first work designed for credible content recommendation under conditions of limited credibility labels.
- A novel disentangled diffusion model is designed to mitigate the recommendations of uncredible content while preserving high recommendation accuracy.
- We propose a new progressively enhanced credible subspace projection to further suppress and mitigate the harmful impacts of uncredible content contained in diffusion targets.
- Comprehensive experiments on three real-world datasets demonstrate the effectiveness of `Disco` in generating both accurate and credible content recommendation.

## 2 PRELIMINARY

### 2.1 CREDIBLE CONTENT RECOMMENDATION

**Content recommendation.** The content recommendation task in this paper follows the sequential recommendation paradigm (Wang et al., 2019a; Zhang et al., 2018), which aims to infer users' potential interests based on their chronologically ordered interaction sequences with content items (e.g., news, and videos and movies) (Wu et al., 2023a; Deldjoo et al., 2016; Goyani & Chaurasiya, 2020). The set of all sequences is denoted as $\mathcal{S} = \{s_1, s_2, \cdots, s_{|\mathcal{S}|}\}$, where each sequence is represented as $s = \{i_1, \cdots, i_{n-1}, i_n\}$ ($s \in \mathcal{S}$). Here, $\{i_1, \cdots, i_{n-1}\}$ are the context items, and $i_n$ is the target item. Each content item $i_k$ is transformed into an embedding vector $\mathbf{e}_k$ using modality-specific feature extractors, such as language models for textual content or visual encoders for images and videos, yielding a sequence of embeddings $\{\mathbf{e}_1, \cdots, \mathbf{e}_n\}$. Given a user's historical sequence $s$, the goal is to generate a personalized ranking over a set of candidate content items and predict the next item that the user is most likely to engage with (Kang & McAuley, 2018).

**Credible content recommendation.** In this paper, we formulate the task of credible content recommendation as mitigating the exposure of users to uncredible items (Wang et al., 2022b; 2024a). A recommendation model is considered more credible if its generated recommendation lists contain smaller proportions of uncredible items. uncredible items include uncredible information like fake news and misinformation Zhang et al. (2025), which often degrades user experience and leads to adverse societal impacts. Moreover, we focus on a more challenging setting in which only partial credibility labels indicating whether an item contains uncredible content are available during training, reflecting the practical difficulty of obtaining exhaustive annotations in real-world RSs. In contrast, complete labels are provided during testing to ensure an accurate evaluation.

**Definition 1** *Content credibility. Content credibility indicates whether an item contains uncredible information such as false, misleading, or inaccurate content. Items containing such information are regarded uncredible (e.g., fake news, misinformation), whereas all others are regarded credible.*

### 2.2 DIFFUSION MODELS FOR SEQUENTIAL RECOMMENDATION

In sequential recommendation scenarios, DMs are generally utilized on the embedding of the last item (i.e., $\mathbf{e}_n$) in a sequence (Yang et al., 2023b; Liu et al., 2025a). The detailed process is as follows:

***In the forward stage***, DMs gradually add Gaussian noise to embedding $\mathbf{e}_n$ acoording to a noise schedule $[\beta_1, \cdots, \beta_T]$:

$$q(\mathbf{e}_n^t|\mathbf{e}_n^{t-1}) = \mathcal{N}(\mathbf{e}_n^t; \sqrt{1-\beta_t}\mathbf{e}_n^{t-1}, \beta_t\mathbf{I}), \qquad q(\mathbf{e}_n^t|\mathbf{e}_n^0) = \mathcal{N}(\mathbf{e}_n^t; \sqrt{\bar{\alpha}_t}\mathbf{e}_n^0, (1-\bar{\alpha}_t)\mathbf{I}), \quad (1)$$

where $\mathbf{e}_n^0 = \mathbf{e}_n$, $\alpha_t = 1 - \beta_t$ and $\bar{\alpha}_t = \prod_{s=1}^{T} \alpha_s$. The first equation is the step-by-step Markov process from $\mathbf{e}_n^{t-1}$ to $\mathbf{e}_n^t$. The second equation is derived based on the Markov chain principle (Ho

et al., 2020), which can be used to directly derive $\mathbf{e}_n^t$ from $\mathbf{e}_n^0$ in one step. A reparameterization trick is then applied to obtain variable $\mathbf{e}_n^t = \sqrt{\bar{\alpha}_t}\mathbf{e}_n^0 + \sqrt{1 - \bar{\alpha}_t}\boldsymbol{\epsilon}$, where $\boldsymbol{\epsilon} \sim \mathcal{N}(\mathbf{0}, \mathbf{I})$.

***In the reverse stage***, DMs progressively recover the diffusion target step by step starting from a Gaussian noise $p(\mathbf{e}_n^T) = \mathcal{N}(\mathbf{0}, \mathbf{I})$:

$$p_\theta(\mathbf{e}_n^{t-1}|\mathbf{e}_n^t, \mathbf{c}) = \mathcal{N}(\mathbf{e}_n^{t-1}; \boldsymbol{\mu}_\theta(\mathbf{e}_n^t, \mathbf{c}, t), \boldsymbol{\Sigma}_\theta(\mathbf{e}_n^t, \mathbf{c}, t)), \tag{2}$$

where $\boldsymbol{\Sigma}_\theta(\mathbf{e}_n^t, \mathbf{c}, t)$ is fixed to $\sigma^2(t) = \frac{1-\bar{\alpha}_{t-1}}{1-\bar{\alpha}_t}\beta_t$ following the common practice in previous work (Yang et al., 2023b; Wang et al., 2023d). $\boldsymbol{\mu}_\theta(\mathbf{e}_n^t, \mathbf{c}, t)$ is the predicted mean from a network $f_\theta(\cdot)$: $\boldsymbol{\mu}_\theta(\mathbf{e}_n^t, \mathbf{c}, t) = \frac{\sqrt{\alpha_t}(1-\bar{\alpha}_{t-1})}{\sqrt{1-\bar{\alpha}_t}}\mathbf{e}_n^t + \frac{\sqrt{\bar{\alpha}_{t-1}}(1-\alpha_t)}{1-\bar{\alpha}_t}f_\theta(\mathbf{e}_n^t, \mathbf{c}, t)$. In DM-based recommendation methods, $f_\theta$ is generally implemented as an MLP for efficiency.

The denoising process is guided by a preference condition $\mathbf{c}$ constructed from the context items $(\{\mathbf{e}_1, \cdots, \mathbf{e}_{n-1}\})$ using a sequence encoder (e.g., Transformer (Kang & McAuley, 2018), GRU (Hidasi et al., 2015)). This condition $\mathbf{c}$ represents users' overall preference.

***Optimization.*** The core of DM-based sequential recommendation is to optimize the conditional data generation distribution $p_\theta(\mathbf{e}_n^0|\mathbf{c})$, which is performed by optimizing the variational bound on negative log likelihood as follows:

$$\mathbb{E}\left[-\log p_\theta(\mathbf{e}_n^0|\mathbf{c})\right] \leq \mathbb{E}_q\left[-\log \frac{p_\theta(\mathbf{e}_n^{0:T}|\mathbf{c})}{q(\mathbf{e}_n^{1:T}|\mathbf{e}_n^0)}\right] := \mathcal{L}. \tag{3}$$

## 3 THE DISCO MODEL

In this section, we first introduce our disentangled diffusion model (Section 3.1) followed by the projection of diffusion targets into a credible subspace (Section 3.2). These two components jointly enable the learning of credible conditions and credible diffusion targets (i.e., two essential elements in diffusion models) to guide the model toward credible generation. Subsequently, to address the more realistic scenario where only a limited portion of content items are labeled with credibility information, we propose a progressive enhancement mechanism for the credible subspace (Section 3.3). Thereafter, we present the overall optimization objective of our proposed model, which integrates a content disentanglement term and a preference contrast term to simultaneously enhance recommendation credibility and accuracy (Section 3.4). Finally, we detail the credible generation and recommendation process after training (Section 3.5). All components are interlocked to construct a unified diffusion-based framework for accurate and credible content recommendation under limited credibility supervision. The pseudo-codes of our model are provided in Algorithms 1, 2, 3.

### 3.1 DISENTANGLED DIFFUSION MODEL

Our disentangled diffusion model is built upon two objectives: (1) generating item embeddings that reflect users' genuine preferences; and (2) reducing the negative influence of uncredible content on the item embedding generation process. To achieve these objectives, we guide DM to generate the item embeddings using the preference-related condition while discouraging the guidance by uncredible content-related condition.

To achieve this, we first introduce two content learners to extract user preference signals and uncredible content signals from items' embeddings. To ensure model simplicity and computational efficiency, both learners are implemented using MLP architectures. Formally, the preference-aware embedding $\mathbf{e}^{pre}$ and the uncredible content-aware embedding $\mathbf{e}^{unc}$ are obtained via $\mathbf{e}^{pre} = \text{MLP}_{pre}(\mathbf{e})$ and $\mathbf{e}^{unc} = \text{MLP}_{unc}(\mathbf{e})$, respectively. Accordingly, the context items in a user's interaction sequence can be transformed into two separate sequences: the preference-related sequence $s^{pre} = \{\mathbf{e}_1^{pre}, \cdots, \mathbf{e}_{n-1}^{pre}\}$ and the uncredible content-related sequence $s^{unc} = \{\mathbf{e}_1^{unc}, \cdots, \mathbf{e}_{n-1}^{unc}\}$.

Thereafter, we construct preference-related and uncredible content-related conditions from corresponding embedding sequences through a Transformer: $\mathbf{c}^{pre} = \text{Transformer}(\{\mathbf{e}_1^{pre}, \cdots, \mathbf{e}_{n-1}^{pre}\})$ and $\mathbf{c}^{unc} = \text{Transformer}(\{\mathbf{e}_1^{unc}, \cdots, \mathbf{e}_{n-1}^{unc}\})$. We employ the same Transformer architecture with (Kang & McAuley, 2018; Yang et al., 2023b). However, applying the Transformer twice is computationally expensive. Therefore, we replace the Transformer with mean pooling to construct un-

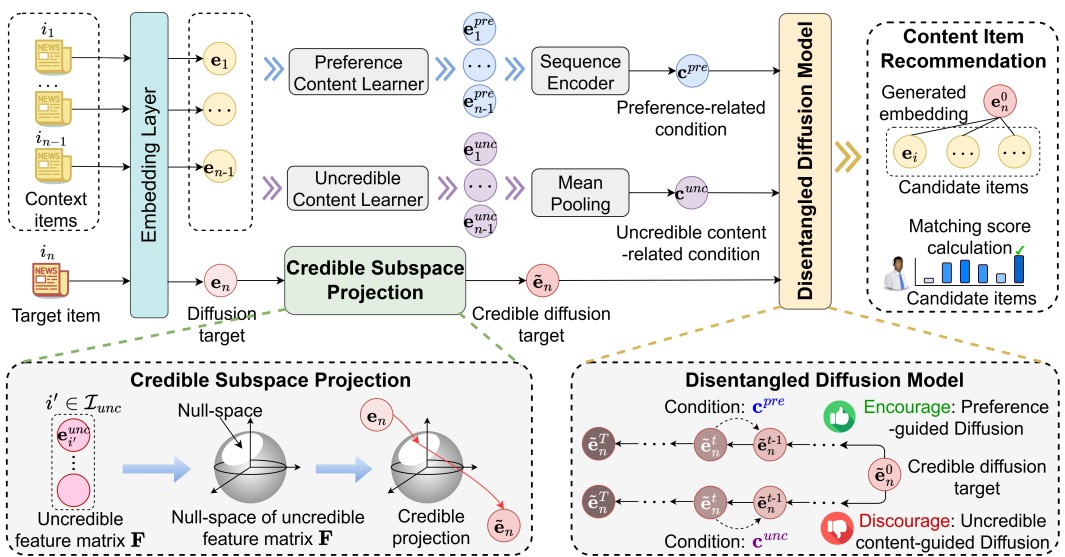

Figure 2: The overall framework of `Disco`. For simplicity and ease of understanding, the progressive enhancement of the credible subspace is not shown in the figure.

credible content-related condition (i.e., $\mathbf{c}^{unc} = \text{Mean}(\{\mathbf{e}_1^{unc}, \cdots, \mathbf{e}_{n-1}^{unc}\})$), since content credibility does not exhibit temporal dependencies.

After constructing the two conditions, `Disco` is optimized by jointly encouraging generation guided by preference-related condition $\mathbf{c}^{pre}$ and discouraging generation guided by uncredible content-related condition $\mathbf{c}^{unc}$. Specifically, it minimizes the variational bound on the target item $\mathbf{e}_n$ when conditioned on $\mathbf{c}^{pre}$, while maximizing the variational bound when conditioned on $\mathbf{c}^{unc}$:

$$\theta^* = \arg\min_\theta \ \mathbb{E}_q\left[-\log\frac{p_\theta(\mathbf{e}_n^{0:T}|\mathbf{c}^{pre})}{q(\mathbf{e}_n^{1:T}|\mathbf{e}_n^0)}\right] - \mathbb{E}_q\left[-\log\frac{p_\theta(\mathbf{e}_n^{0:T}|\mathbf{c}^{unc})}{q(\mathbf{e}_n^{1:T}|\mathbf{e}_n^0)}\right]. \tag{4}$$

By employing this training objective, DM can naturally disentangle the two types of information without requiring additional components to explicitly enforce the separation. Importantly, we do not use $\mathbf{e}_n^{pre}$ or $\mathbf{e}_n^{unc}$ as diffusion targets. Otherwise, disentanglement would be ineffective because the diffusion condition and target would lie in the same space, lacking a meaningful disentanglement direction. Our ablation study in Section 4.3 further confirms this, showing that replacing $\mathbf{e}_n$ with $\mathbf{e}_n^{pre}$ or $\mathbf{e}_n^{unc}$ significantly deteriorates recommendation performance.

The training objective in Equation 4 can be reformulated as the following loss:

$$\mathcal{L} = \mathbb{E}_{\mathbf{e}_n^0, \mathbf{c}^{pre}, t}\left[\|\mathbf{e}_n^0 - f_\theta(\mathbf{e}_n^t, \mathbf{c}^{pre}, t)\|_2^2\right] - \mathbb{E}_{\mathbf{e}_n^0, \mathbf{c}^{unc}, t}\left[\|\mathbf{e}_n^0 - f_\theta(\mathbf{e}_n^t, \mathbf{c}^{unc}, t)\|_2^2\right]. \tag{5}$$

The detailed derivation is provided in the Appendix D. However, directly training the model with this loss can lead to severe instability. Specifically, the second term may converge to an extremely small value, causing the model to predominantly optimize this term while neglecting the first term. To address this, inspired by (Liu et al., 2025a), we replace the MSE loss with a cosine loss:

$$\mathcal{L} = \mathbb{E}_{\mathbf{e}_n^0, \mathbf{c}^{pre}, t}\left[S\left(\mathbf{e}_n^0, f_\theta(\mathbf{e}_n^t, \mathbf{c}^{pre}, t)\right)\right] - \mathbb{E}_{\mathbf{e}_n^0, \mathbf{c}^{unc}, t}\left[S\left(\mathbf{e}_n^0, f_\theta(\mathbf{e}_n^t, \mathbf{c}^{unc}, t)\right)\right], \tag{6}$$

where $S(\cdot, \cdot) = (1 - \cos(\cdot, \cdot))^2$ and $\cos(\cdot, \cdot)$ is the cosine similarity of two embeddings. This loss preserves the same optimization direction as that in Equation 5, while its values remain within a stable range, thereby improving the stability of model training.

## 3.2 CREDIBLE SUBSPACE PROJECTION

The last item in a user's interaction sequence may also be an uncredible item, leading to an uncredible diffusion target. In such cases, optimizing Equation 6 might still be suboptimal for mitigating uncredible content. To address this, we design a credible subspace projection operation, which projects the diffusion target into the credible subspace to suppress uncredible content.

To achieve this, we first construct an uncredible feature matrix $\mathbf{F} \in \mathbb{R}^{|\mathcal{I}_{unc}| \times d}$ by stacking the uncredible content embeddings of all uncredible items, i.e., $\{\mathbf{e}_i^{unc} | i \in \mathcal{I}_{unc}\}$, where $\mathcal{I}_{unc}$ is the set of uncredible items and $d$ is the embedding size. The credible subspace projection is then performed by projecting the diffusion target into the null space of $\mathbf{F}$, which serves as a subspace that maximally excludes uncredible content. Following prior work on null-space projection (Fang et al., 2025; Wang et al., 2021a; Hu et al., 2025), we apply Singular Value Decomposition (SVD) on $\mathbf{F}^\top$:

$$\{\mathbf{U}, \mathbf{\Lambda}, \mathbf{V}\} = \text{SVD}(\mathbf{F}^\top), \tag{7}$$

where each column of left singular matrix $\mathbf{U}$ is an orthogonal basis of $\mathbf{F}^\top$. $\mathbf{V}$ denotes the right singular matrix. $\mathbf{\Lambda}$ contains the corresponding singular values, which indicate the magnitude of uncredible information encoded by the orthogonal basis in $\mathbf{U}$. A higher singular value corresponds to a orthogonal basis that is denser in uncredible information. Accordingly, we remove the submatrix $\mathbf{U}_1$ in $\mathbf{U}$ whose singular values exceed a predefined threshold. The remaining submatrix, $\mathbf{U}_2$, consists only of orthogonal basis containing sparse or no uncredible information. The diffusion target is subsequently projected into the null space of $\mathbf{F}$ using the following operation:

$$\tilde{\mathbf{e}}_n = \mathbf{e}_n \mathbf{U}_2 \mathbf{U}_2^\top, \tag{8}$$

where $\tilde{\mathbf{e}}_n$ is the credible diffusion target. To preserve the useful information contained in the original target item embedding $\mathbf{e}_n$, we adopt a residual connection to combine it with the projected embedding, yielding the responsible diffusion target as: $\tilde{\mathbf{e}}_n = (\tilde{\mathbf{e}}_n + \mathbf{e}_n)/2$. This credible diffusion target is then used to replace the original target in the diffusion loss, as defined in Equation 6:

$$\mathcal{L} = \mathbb{E}_{\tilde{\mathbf{e}}_n^0, \mathbf{c}^{pre}, t} \left[ S\left(\tilde{\mathbf{e}}_n^0, f_\theta(\tilde{\mathbf{e}}_n^t, \mathbf{c}^{pre}, t)\right) \right] - \mathbb{E}_{\tilde{\mathbf{e}}_n^0, \mathbf{c}^{unc}, t} \left[ S\left(\tilde{\mathbf{e}}_n^0, f_\theta(\tilde{\mathbf{e}}_n^t, \mathbf{c}^{unc}, t)\right) \right], \tag{9}$$

where $\tilde{\mathbf{e}}_n^0 = \tilde{\mathbf{e}}_n$. Training the model with this loss further enhances the credibility of recommendation generation by projecting the diffusion target into a more credible subspace.

## 3.3 Progressive Enhancement of Credible Projection

Owing to the second challenge mentioned in the introduction, the uncredible feature matrix $\mathbf{F}$ may capture only a limited set of uncredible features, leading to an incomplete credible subspace projection. To address this, we propose a progressive enhancement strategy for credible projection.

Let $\mathcal{I}_{unc}$ denote the set of items already labeled as uncredible content, and the remaining items in $\mathcal{I} \setminus \mathcal{I}_{unc}$ have uncertain labels. Actually, there is still a proportion of items in $\mathcal{I} \setminus \mathcal{I}_{unc}$ that are uncredible items but are not verified. In real-world scenarios, uncredible content often exhibits shared features. For instance, fake news articles tend to use emotionally charged or sensational headlines, such as those written in all capital letters[1]. In light of this, we try to detect the potential uncredible items by calculating the uncredible degree of items in $\mathcal{I} \setminus \mathcal{I}_{unc}$:

$$\text{UD}(i) = \frac{1}{|\mathcal{I}_{unc}|} \sum_{i' \in \mathcal{I}_{unc}} \cos(\mathbf{e}_i^{unc}, \mathbf{e}_{i'}^{unc}), \tag{10}$$

where $\mathbf{e}_i^{unc}$ and $\mathbf{e}_{i'}^{unc}$ are uncredible content embeddings of item $i$ in $\mathcal{I} \setminus \mathcal{I}_{unc}$ and item $i'$ in $\mathcal{I}_{unc}$. $\cos(\cdot, \cdot)$ calculates the cosine similarity between two embeddings. $\text{UD}(i)$ represents the uncredible degree of item $i$, quantifying the likelihood that item $i$ in $\mathcal{I} \setminus \mathcal{I}_{unc}$ is an uncredible item. Items with the highest uncredible degrees are selected as potential uncredible items.

At the early stages of training, the disentangled diffusion model is not fully trained, resulting in less accurate estimates of uncredible degrees. As training goes on, the model's capability improves. Therefore, instead of using a fixed selection ratio, we propose a progressive selection strategy. Specifically, we predefine a maximum selection ratio $\gamma$ and linearly increase the selection ratio from zero to $\gamma$ after $m$ training iterations. Consequently, the selection ratio at the $j$-th training iteration is given by $ratio(j) = \min(\gamma, \frac{j}{m}\gamma)$. After calculating the current selection ratio, the top $\lfloor |\mathcal{I} \setminus \mathcal{I}_{unc}| \cdot ratio(j) \rfloor$ items in $\mathcal{I} \setminus \mathcal{I}_{unc}$ with the highest uncredible degrees are selected as the potential uncredible items and added to the set $\mathcal{I}_{unc}$. Subsequently, the uncredible feature matrix $\mathbf{F}$ is updated based on the expanded set $\mathcal{I}_{unc}$. This update enhances the comprehensiveness of the null space of constructed uncredible content features, reduces residual uncredible features, and enables the diffusion target to be projected into a more credible subspace.

---

[1] https://techcrunch.com/2017/04/06/facebook-puts-link-to-10-tips-for-spotting-false-news-atop-feed/

## 3.4 Overall Optimization Objective of Disco

The optimization loss in Equation 9 primarily addresses two objectives: capturing a user's positive preference (i.e., the target item) and enforcing content disentanglement. However, in RSs, modeling a user's negative preference is also crucial, as it enables the model to understand which types of items users are not interested in. To incorporate this objective and further enhance recommendation accuracy, we formulate the final version of our diffusion loss with an additional preference contrast term by enlarging the distance between positive and negative preference:

$$
\mathcal{L}_{\texttt{Disco}} = \underbrace{S\left(\tilde{\mathbf{e}}_n^0, f_\theta(\tilde{\mathbf{e}}_n^t, \mathbf{c}^{pre}, t)\right) - S\left(\tilde{\mathbf{e}}_n^0, f_\theta(\tilde{\mathbf{e}}_n^t, \mathbf{c}^{unc}, t)\right)}_{\text{Content disentanglement}}
$$
$$
+ \underbrace{w\left(S\left(\tilde{\mathbf{e}}_n^0, f_\theta(\tilde{\mathbf{e}}_n^t, \mathbf{c}^{pre}, t)\right) - S\left(\mathbf{e}_{neg}^0, f_\theta(\mathbf{e}_{neg}^t, \mathbf{c}^{pre}, t)\right)\right)}_{\text{Preference contrast}},
\tag{11}
$$

where $\mathbf{e}_{neg}^0 = \mathbf{e}_{neg}$ is the embedding of a sampled negative preference item (i.e., an item that a user has not interacted with). $w$ is a hyperparameter controlling the contribution of each term, and $t \sim U(0, T)$. For simplicity, we omit the expectation notation. The second term encourages the diffusion model to generate items reflecting users' positive preferences rather than negative preferences. Although computing this loss requires multiple forward passes through $f_\theta$, the computational overhead remains minimal, as $f_\theta$ is implemented as an MLP, which is time-efficient. Moreover, since all components share a single $f_\theta$ network, no additional memory consumption is required.

## 3.5 Credible Generation and Content Recommendation

In this section, we describe the generation/inference process of `Disco`.

Following the generation paradigm of Denoising Diffusion Probabilistic Models (Ho et al., 2020), the one-step generation procedure is defined as follows:

$$
\mathbf{e}_n^{t-1} = \frac{\sqrt{\bar{\alpha}_{t-1}}(1-\alpha_t)}{1-\alpha_t} f_\theta(\mathbf{e}_n^t, \mathbf{c}^{pre}, t) + \frac{\sqrt{\alpha_t}(1-\bar{\alpha}_{t-1})}{1-\bar{\alpha}_t}\mathbf{e}_n^t + \sqrt{\frac{1-\bar{\alpha}_{t-1}}{1-\bar{\alpha}_t}(1-\alpha_t)}\boldsymbol{\epsilon}.
\tag{12}
$$

The generation step begins with $\mathbf{e}_n^T \sim \mathcal{N}(\mathbf{0}, \mathbf{I})$. We employ preference-related condition $\mathbf{c}^{pre}$ to guide the generation, ensuring the generated embeddings capture users' genuine preferences. This approach prevents the generated embedding from incorporating uncredible content features, even if users have previously interacted with uncredible items, thereby enhancing the credibility of the generation. To improve efficiency, we adopt the DDIM sampling strategy (Song et al., 2021).

The generated embedding $\mathbf{e}_n^0$ represents the user's predicted future preference. It is then used to compute matching scores with candidate items: $\hat{y}_i = \mathbf{e}_n^0 \cdot \mathbf{e}_i^\top$, where $\mathbf{e}_i$ is the embedding of candidate item $i$. The top-K items with the highest matching scores are subsequently recommended to the user.

**Discussion: Comparison between Disco and other DM-based methods.**

- **Model architecture:** DreamRecYang et al. (2023b), DiffuRec Li et al. (2023), and PreferDiff Liu et al. (2025a) all adopt a single-channel diffusion architecture, in which a single condition is used to guide the generation of the target item. In contrast, `Disco` employs a disentangled diffusion architecture with dual channels, leveraging two conditions to guide the generation. This design plays a crucial role in separating preference-related information from uncredible content signals.

- **Objective formulation:** DreamRec uses the standard ELBO objective for diffusion models, whereas PreferDiff adopts a variant ELBO combined with a Bayesian Personalized Ranking (BPR) loss. DiffuRec instead uses a cross-entropy (CE) objective, essentially turning a generative diffusion model into a discriminative one. By contrast, our model also belongs to a variant of the ELBO, but one specifically designed to achieve both accurate and credible generation—an ability that DreamRec, DiffuRec, and PreferDiff do not possess.

## 4 EXPERIMENTS

### 4.1 EXPERIMENTAL SETUP

**Datasets.** We evaluate our method on three datasets: PolitiFact, GossipCop and MHMisinfo. The PolitiFact and GossipCop datasets are derived from FakeNewsNet repository[2] (Shu et al., 2020). These datasets contain user-news interaction data, where fake news is treated as uncredible content items. Our task requires user–item interaction sequences together with credibility labels indicating whether the items are credible items or not. To the best of our knowledge, these three datasets are the only publicly available datasets that meet these requirements, which can be used in our experiments. MHMisinfo is collected from a video-based mental health misinformation dataset[3] (Nguyen et al., 2025). This dataset contains users' interaction sequence with videos and the videos containing misinformation are uncredible items. Since this dataset does not provide video metadata but only textual descriptions, we use the textual descriptions as the item content. A detailed description of these datasets is provided in the Appendix B.1.

**Baselines.** To evaluate the effectiveness of `Disco`, we compare it with four categories of sequential recommendation methods: **(1) Traditional methods**, including GRU4Rec (Hidasi et al., 2015), SASRec (Kang & McAuley, 2018), Bert4Rec (Sun et al., 2019), and LRURec (Yue et al., 2024); **(2) Contrastive learning-based methods**, including CL4SRec (Xie et al., 2022) and ContraRec (Wang et al., 2023a); **(3) Credible recommendation methods**, including Rec4Mit (Wang et al., 2022b), HDInt (Wang et al., 2024a), and PRISM (Ma et al., 2025); **(4) DM-based methods**, including DreamRec (Yang et al., 2023b), DiffuRec (Li et al., 2023), PRISM (Ma et al., 2025), PreferDiff (Liu et al., 2025a). The details of these methods are provided in the Appendix B.2.

**Evaluation Metrics.** We evaluate model performance using three types of metrics: accuracy-oriented metrics such as HR@K and NDCG@K, a credibility-oriented metric CR@K (i.e., credible rate), and a combined metric HC@K that integrates HR@K and CR@K. We follow the standard top-K evaluation protocol with $K = 5, 10$, as commonly adopted in sequential recommendation tasks (Kang & McAuley, 2018). Specifically, CR@K, proposed by (Wang et al., 2022b), measures the proportion of credible content items in the top-K recommendation list, where a higher value indicates a more credible output. The detailed definitions of these metrics are provided in Appendix B.3.

**Implementation Details.** During training, we assume that labels for 20% of randomly selected uncredible items are available, simulating the sparsity of labeled data in real-world scenarios. For fair comparison, we initialize each model's hyperparameters as suggested in the original papers and then fine-tune them on our datasets to ensure their best performances are reported. The hyperparameter $w$ is tuned within $\{0.5, 1, 1.5, 2, 5\}$, and $\gamma$ within $\{0.1, 0.2, 0.3, 0.4, 0.5\}$, while $m$ is fixed at 10,000. The threshold for constructing the null space is fixed at 3. Model parameters are optimized using AdamW (Loshchilov & Hutter, 2017). Each method is run five times, and we report the average performance along with the standard deviation. Additional implementation details and hyperparameter settings are provided in the Appendix B.4.

### 4.2 OVERALL PERFORMANCE COMPARISON

From the results reported in Table 1, we have the following observations:

**Our proposed method, `Disco`, consistently outperforms competitive methods in both accurate and credible content recommendation.** `Disco` achieves the best performance across all datasets and metrics. These results indicate that `Disco` can effectively reduce the recommendations of uncredible content while maintaining high recommendation accuracy. This is enabled by the disentangled diffusion model and the progressively enhanced credible subspace projection. Notably, `Disco` excels in recommendation accuracy due to the incorporation of negative preference modeling, thereby better modeling users' genuine preference.

**DM-based methods generally exhibit better recommendation accuracy than other approaches.** Thanks to their strong ability to model complex distributions of user behaviors and item content, as well as to capture the inherent uncertainty in user behaviors, DM-based methods consistently

---

[2]https://github.com/KaiDMML/FakeNewsNet
[3]https://zenodo.org/records/13191247

Table 1: Overall performance comparison. The best performances are in **bold**, and the second-best performances are underlined. The standard deviation is present in the form of percentage (%).

| Datasets | Methods | HR@5↑ | HR@10↑ | NDCG@5↑ | NDCG@10↑ | CR@5↑ | CR@10↑ | HC@5↑ | HC@10↑ |
|---|---|---|---|---|---|---|---|---|---|
| PolitiFact | GRU4Rec | 0.2142 ±0.51 | 0.3390 ±0.64 | 0.1463 ±0.37 | 0.1863 ±0.41 | 0.9266 ±0.84 | 0.9122 ±0.76 | 0.2929 ±0.51 | 0.3889 ±0.51 |
| | SASRec | 0.2158 ±0.10 | 0.3519 ±0.18 | 0.1386 ±0.05 | 0.1823 ±0.10 | 0.9059 ±0.41 | 0.9028 ±0.39 | 0.2929 ±0.12 | 0.3955 ±0.13 |
| | LRURec | 0.2168 ±0.14 | 0.3506 ±0.35 | 0.1443 ±0.06 | 0.1872 ±0.13 | 0.8976 ±0.20 | 0.8956 ±0.16 | 0.2924 ±0.14 | 0.3938 ±0.23 |
| | Bert4Rec | 0.2191 ±0.11 | 0.3473 ±0.16 | 0.1472 ±0.05 | 0.1883 ±0.08 | 0.9172 ±0.11 | 0.9045 ±0.15 | 0.2960 ±0.09 | 0.3929 ±0.13 |
| | CL4SRec | 0.2247 ±0.05 | 0.3527 ±0.17 | 0.1508 ±0.07 | 0.1919 ±0.05 | 0.9132 ±0.63 | 0.9027 ±0.67 | 0.3012 ±0.09 | 0.3960 ±0.14 |
| | ContraRec | 0.2241 ±0.25 | 0.3512 ±0.36 | 0.1508 ±0.11 | 0.1917 ±0.17 | 0.8803 ±2.67 | 0.8979 ±0.60 | 0.2969 ±0.35 | 0.3941 ±0.16 |
| | Rec4Mit | 0.2118 ±0.28 | 0.3449 ±0.40 | 0.1413 ±0.11 | 0.1840 ±0.15 | 0.8959 ±0.73 | 0.8925 ±0.59 | 0.2876 ±0.30 | 0.3891 ±0.25 |
| | HDInt | 0.2153 ±0.27 | 0.3594 ±0.34 | 0.1272 ±0.19 | 0.1734 ±0.21 | 0.8944 ±0.31 | 0.8946 ±0.36 | 0.2906 ±0.21 | 0.3985 ±0.18 |
| | PRISM | 0.1927 ±0.48 | 0.2758 ±0.27 | 0.1348 ±0.31 | 0.1615 ±0.25 | 0.9335 ±0.02 | 0.9172 ±1.21 | 0.2727 ±0.47 | 0.3446 ±0.26 |
| | DreamRec | 0.2416 ±1.88 | 0.3287 ±1.94 | 0.1767 ±1.70 | 0.2047 ±1.69 | 0.8620 ±3.24 | 0.8437 ±2.03 | 0.3054 ±1.42 | 0.3664 ±1.15 |
| | DiffuRec | 0.2606 ±1.21 | 0.3558 ±1.69 | 0.1894 ±0.92 | 0.2214 ±1.08 | 0.9265 ±1.60 | 0.9153 ±0.76 | 0.3334 ±0.81 | 0.4027 ±0.88 |
| | PreferDiff | 0.2531 ±1.02 | 0.3554 ±0.52 | 0.1818 ±1.04 | 0.2147 ±0.86 | 0.8925 ±2.08 | 0.8981 ±2.34 | 0.3228 ±0.90 | 0.3968 ±0.75 |
| | Disco | **0.2678 ±0.53** | **0.3775 ±0.70** | **0.1983 ±0.17** | **0.2336 ±0.19** | **0.9823 ±0.34** | **0.9425 ±1.72** | **0.3466 ±0.50** | **0.4192 ±0.78** |
| | $p$-values | $6.3e^{-2}$ | $8.1e^{-3}$ | $2.9e^{-2}$ | $2.4e^{-2}$ | $7.8e^{-4}$ | $1.8e^{-2}$ | $1.8e^{-2}$ | $2.4e^{-2}$ |
| GossipCop | GRU4Rec | 0.2226 ±2.44 | 0.3194 ±3.10 | 0.1466 ±1.83 | 0.1778 ±1.94 | 0.8864 ±1.80 | 0.8706 ±1.60 | 0.2957 ±2.61 | 0.3678 ±2.61 |
| | SASRec | 0.3078 ±0.19 | 0.4706 ±0.05 | 0.1607 ±0.19 | 0.2135 ±0.14 | 0.8743 ±1.87 | 0.8526 ±1.38 | 0.3612 ±0.49 | 0.4473 ±0.41 |
| | LRURec | 0.3316 ±0.18 | 0.5101 ±0.11 | 0.1697 ±0.12 | 0.2276 ±0.09 | 0.8544 ±2.39 | 0.8439 ±1.57 | 0.3732 ±0.52 | 0.4618 ±0.57 |
| | Bert4Rec | 0.2372 ±0.18 | 0.3711 ±0.18 | 0.1338 ±0.15 | 0.1770 ±0.14 | 0.8764 ±2.00 | 0.8587 ±0.74 | 0.3073 ±1.08 | 0.3984 ±0.12 |
| | CL4SRec | 0.2898 ±0.39 | 0.4100 ±0.45 | 0.1784 ±0.30 | 0.2174 ±0.30 | 0.8938 ±0.04 | 0.8932 ±1.50 | 0.3516 ±0.32 | 0.4275 ±0.39 |
| | ContraRec | 0.2848 ±0.14 | 0.4224 ±0.16 | 0.1574 ±0.15 | 0.2020 ±0.19 | 0.8754 ±2.06 | 0.8549 ±0.94 | 0.3450 ±0.39 | 0.4249 ±0.20 |
| | Rec4Mit | 0.2775 ±1.73 | 0.4403 ±1.98 | 0.1606 ±1.26 | 0.2133 ±1.28 | 0.8979 ±0.62 | 0.8649 ±1.39 | 0.3427 ±1.57 | 0.4360 ±0.89 |
| | HDInt | 0.3407 ±0.15 | 0.5249 ±0.27 | 0.1748 ±0.09 | 0.2345 ±0.09 | 0.8986 ±0.30 | 0.8694 ±0.91 | 0.3875 ±0.14 | 0.4755 ±0.25 |
| | PRISM | 0.2948 ±0.33 | 0.3447 ±0.25 | 0.2301 ±0.30 | 0.2463 ±0.29 | 0.8806 ±3.09 | 0.8738 ±1.63 | 0.3531 ±0.70 | 0.3852 ±0.42 |
| | DreamRec | 0.4619 ±0.08 | 0.5501 ±0.13 | 0.3415 ±0.05 | 0.3704 ±0.07 | 0.8464 ±3.77 | 0.8339 ±1.93 | 0.4415 ±1.04 | 0.4742 ±0.61 |
| | DiffuRec | 0.4571 ±0.43 | 0.5008 ±0.65 | 0.3887 ±0.23 | 0.4029 ±0.26 | 0.8313 ±0.58 | 0.8157 ±0.45 | 0.4354 ±0.19 | 0.4495 ±0.24 |
| | PreferDiff | 0.4969 ±0.05 | 0.6022 ±0.07 | 0.3655 ±0.01 | 0.3999 ±0.02 | 0.8307 ±3.36 | 0.8228 ±2.76 | 0.4523 ±1.14 | 0.4887 ±0.88 |
| | Disco | **0.5236 ±0.80** | **0.6143 ±0.66** | **0.3996 ±0.91** | **0.4292 ±0.81** | **0.9277 ±0.28** | **0.9039 ±1.53** | **0.4918 ±0.40** | **0.5207 ±0.60** |
| | $p$-values | $1.4e^{-3}$ | $9.5e^{-3}$ | $5.5e^{-2}$ | $2.5e^{-3}$ | $9.8e^{-5}$ | $2.4e^{-1}$ | $1.7e^{-3}$ | $3.0e^{-3}$ |
| MHMisinfo | GRU4Rec | 0.1151 ±4.47 | 0.1894 ±2.47 | 0.0760 ±4.43 | 0.0998 ±1.71 | 0.8380 ±2.68 | 0.8608 ±2.38 | 0.1803 ±2.18 | 0.2624 ±2.86 |
| | SASRec | 0.1485 ±0.39 | 0.2592 ±1.31 | 0.0826 ±0.26 | 0.1179 ±0.26 | 0.8839 ±1.24 | 0.8915 ±0.58 | 0.2190 ±0.56 | 0.3276 ±1.21 |
| | LRURec | 0.1571 ±0.77 | 0.2704 ±0.39 | 0.0877 ±0.42 | 0.1268 ±0.39 | 0.8359 ±0.78 | 0.8818 ±0.76 | 0.2283 ±0.93 | 0.3350 ±0.23 |
| | Bert4Rec | 0.1391 ±0.70 | 0.2299 ±0.03 | 0.0847 ±0.44 | 0.1138 ±0.53 | 0.8162 ±0.75 | 0.8786 ±0.60 | 0.2074 ±0.92 | 0.3017 ±1.08 |
| | CL4SRec | 0.1734 ±0.50 | 0.2621 ±0.35 | 0.1101 ±0.30 | 0.1387 ±0.19 | 0.8577 ±0.43 | 0.9081 ±0.21 | 0.2469 ±0.57 | 0.3323 ±0.33 |
| | ContraRec | 0.1357 ±0.62 | 0.2258 ±0.58 | 0.0832 ±0.27 | 0.1122 ±0.28 | 0.8275 ±3.79 | 0.8760 ±2.45 | 0.2043 ±0.85 | 0.2980 ±0.74 |
| | Rec4Mit | 0.1460 ±1.00 | 0.2659 ±0.90 | 0.0886 ±0.60 | 0.1269 ±0.34 | 0.8424 ±0.76 | 0.9006 ±0.48 | 0.2166 ±1.24 | 0.3343 ±0.83 |
| | HDInt | 0.1471 ±0.40 | 0.2654 ±1.20 | 0.0852 ±0.25 | 0.1230 ±0.24 | 0.8306 ±0.38 | 0.8881 ±0.55 | 0.2168 ±0.61 | 0.3301 ±0.83 |
| | PRISM | 0.1700 ±1.31 | 0.2339 ±1.56 | 0.1181 ±0.62 | 0.1388 ±0.69 | 0.8398 ±3.91 | 0.8919 ±1.94 | 0.2418 ±1.39 | 0.3065 ±1.28 |
| | DreamRec | 0.1819 ±0.84 | 0.2426 ±0.67 | 0.1313 ±0.60 | 0.1509 ±0.54 | 0.9002 ±2.94 | 0.8952 ±3.06 | 0.2633 ±1.10 | 0.3176 ±0.87 |
| | DiffuRec | 0.1402 ±2.34 | 0.2095 ±2.77 | 0.0919 ±2.05 | 0.1142 ±1.97 | 0.8976 ±3.18 | 0.9114 ±1.75 | 0.2128 ±2.57 | 0.2861 ±2.49 |
| | PreferDiff | 0.1974 ±0.72 | 0.2620 ±0.59 | 0.1389 ±0.54 | 0.1598 ±0.51 | 0.8693 ±3.73 | 0.8874 ±2.31 | 0.2713 ±0.85 | 0.3294 ±0.80 |
| | Disco | **0.2215 ±0.89** | **0.2822 ±0.89** | **0.1580 ±0.69** | **0.1778 ±0.64** | **0.9305 ±0.82** | **0.9264 ±1.01** | **0.3000 ±0.97** | **0.3507 ±0.86** |
| | $p$-values | $7.2e^{-3}$ | $6.0e^{-2}$ | $4.8e^{-3}$ | $6.6e^{-3}$ | $1.7e^{-2}$ | $4.3e^{-3}$ | $3.1e^{-3}$ | $1.5e^{-2}$ |

outperform traditional recommendation approaches. Specifically, Disco consistently achieves the best performance, while DiffuRec and PreferDiff generally rank second across most cases. This observation aligns with prior works (Yang et al., 2023b; Liu et al., 2025a).

**Other methods focusing on credible content recommendation perform poorly under limited labeled data.** Although methods such as Rec4Mit, HDInt, and PRISM aim to mitigate the recommendation of uncredible content, they assume full access to labels for all uncredible items. This assumption does not hold in real-world scenarios, where a large portion of content items remain unverified. Consequently, these methods cannot achieve satisfactory uncredible content mitigation, as they rely on accurate and complete labeled data to train classifiers for detecting uncredible items. This limitation motivates our design of the progressively enhanced credible subspace projection, which has been empirically shown to effectively mitigate uncredible content.

### 4.3 ABLATION STUDY AND HYPERPARAMETER ANALYSIS

**Ablation Study.** In this section, we evaluate the effectiveness of each key component of Disco. We design six variants: (1) **w/o Dis**, which removes the disentanglement module (i.e., using original item embeddings for subsequent modeling); (2) **w/o CSP**, which removes the credible subspace projection; **(3) w/o PERS**, which removes the progressive enhancement of credible subspace; **(4)**

Table 2: Ablation study of `Disco`.

| Methods | PolitiFact | | | | GossipCop | | | | MHMisinfo | | | |
|---|---|---|---|---|---|---|---|---|---|---|---|---|
| | HR@5 | NDCG@5 | CR@5 | HC@5 | HR@5 | NDCG@5 | CR@5 | HC@5 | HR@5 | NDCG@5 | CR@5 | HC@5 |
| `Disco` | **0.2664** | **0.1975** | **0.9835** | **0.3455** | **0.5236** | **0.3996** | 0.9272 | **0.4918** | **0.2215** | **0.1580** | **0.9305** | **0.3000** |
| w/o Dis | 0.2273 | 0.1605 | 0.9121 | 0.3033 | 0.4984 | 0.3678 | 0.9193 | 0.4783 | 0.1910 | 0.1342 | 0.8654 | 0.2650 |
| w/o CSP | 0.2575 | 0.1809 | 0.9431 | 0.3331 | 0.5183 | 0.3898 | 0.9155 | 0.4860 | 0.2178 | 0.1548 | 0.9066 | 0.2942 |
| w/o PERS | 0.2651 | 0.1919 | 0.9423 | 0.3393 | 0.5147 | 0.3891 | 0.9267 | 0.4876 | 0.2103 | 0.1498 | 0.9113 | 0.2877 |
| w/o PC | 0.2637 | 0.1934 | 0.9677 | 0.3413 | 0.4643 | 0.3531 | **0.9316** | 0.4651 | 0.2006 | 0.1419 | 0.8708 | 0.2747 |
| w/o CE | 0.1034 | 0.0622 | 0.7600 | 0.1626 | 0.0005 | 0.0003 | 0.7600 | 0.0010 | 0.1613 | 0.1011 | 0.8000 | 0.2299 |
| w/ DDT | 0.2609 | 0.1899 | 0.9499 | 0.3368 | 0.4025 | 0.3190 | 0.9069 | 0.4265 | 0.2089 | 0.1458 | 0.8466 | 0.2797 |

**w/o PC**, which removes the preference contrast term in Equation 11; **(5) w/o CE**, which replaces cosine error with MSE in Equation 11; **(6) w/ DDT**, which utilizes disentangled embedding of target item as the diffusion target in Equation 4. As shown in Table 2, each component contributes positively. Specifically, removing the disentanglement module significantly harms model performance, highlighting that this module can effectively separate uncredible content and preserve users' preference-related information. Figure 3 also verifies that our designed disentangled diffusion model can effectively separate preference-related content and uncredible content. Both removing credible subspace projection and progressive enhancement of credible subspace significantly degrade `Disco`'s recommendation credibility (i.e., CR@5). In addition, without the preference contrast term, the recommendation accuracy will deteriorate. In particular, if not replacing MSE loss with cosine error, `Disco`'s performance will be degraded to a great extent, due to unstable training (this issue is discussed in Appendix B.10 in detail). Meanwhile, if using the disentangled item embedding as the diffusion target, `Disco` cannot obtain satisfactory recommendation performance, due to ineffective disentanglement.

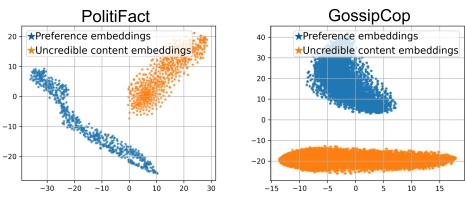

Figure 3: Disentanglement visualization.

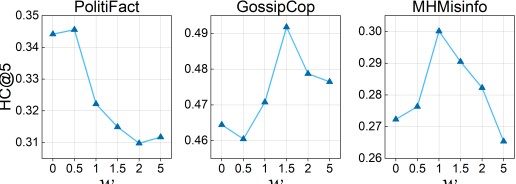

Figure 4: Effect of $w$ on `Disco`.

**Hyperparameter Analysis.** The hyperparameter $w$ controls the contribution of negative preference diffusion. As shown in Figure 4, `Disco` achieves the best performance when $w = 0.5$ on PolitiFact, $w = 1.5$ on GossipCop, and $w = 1$ on MHMisinfo. These results demonstrate the effectiveness of incorporating negative preference diffusion. However, continuously increasing the value of $w$ leads to an imbalance between different training objectives, thereby deteriorating model performance. More analysis of hyperparameters are provided in Appendix B.5. After analyzing the relationship between dataset statistics and the hyperparameter $w$, we found that the optimal value of $w$ is proportional to the number of items in a dataset. The hyperparameter $w$ controls the contribution of the preference-contrast term, which involves sampling negative items from the set of un-interacted items. When the number of items is larger, the sampled negative items represent a smaller portion of users' negative preferences. Therefore, a relatively larger $w$ is needed to adequately learn users' negative preferences, thereby improving recommendation accuracy.

## 5 CONCLUSIONS AND LIMITATIONS

In this paper, we proposed `Disco`, a model designed to steer DMs towards credible content recommendation. To this end, we first designed a disentangled diffusion model to separate uncredible content from the generation process. Considering the limited labeled data, a progressively enhanced credible subspace projection is proposed to make the diffusion training process more credible. However, similar to previous work (Liu et al., 2025a; Yang et al., 2023b), `Disco` also requires a relatively large embedding dimension to achieve strong performance. This inevitably leads to increased training time, which is a common limitation of current DM-based recommendation methods (Liu et al., 2025a). Future work could focus on designing DM-based models that maintain strong performance even under low-dimensional embeddings.

**Ethics Statement.** This paper aims to develop a diffusion model (DM)-based method for credible content recommendation. The goal of our approach is to serve societal good by mitigating the spread of uncredible content through recommender systems. We confirm that we do not anticipate any negative impacts and our work does not violate the ICLR code of ethics.

**Reproducibility Statement.** All results reported in this paper are fully reproducible. The pseudo codes of our model are provided in Algorithms 1, 2, and 3. The hyperparameter search space and experimental environment are discussed in Section B.4 and Table 4. We provide the code and data of our method at https://github.com/iamZhuoCai/Disco.

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

# A   RELATED WORK

Our content recommendation task follows the paradigm of sequential recommendation (SRec) (Kang & McAuley, 2018; Wang et al., 2019a). Accordingly, our work is closely aligned with the research on sequential recommendation and diffusion model (DM)-based sequential recommendation. In this section, we review the studies on these two topics in detail.

## A.1   SEQUENTIAL RECOMMENDATION

Sequential recommendation (SRec) has been widely studied in RSs Wang et al. (2023c), owing to the natural temporal order of users' behaviors (Wang et al., 2019a; 2021b; 2018; 2019b; Shi et al., 2023). SRec can be technically divided into two categories: traditional sequential models and deep learning-based models. Traditional sequential models generally leverage sequential pattern mining (Yap et al., 2012) or Markov chain models (He & McAuley, 2016) to model the item dependencies in users' interaction sequences. Traditional sequential methods can only capture simple interaction patterns or short-term dependencies, thereby cannot achieve satisfactory recommendation performance. To overcome these limitations, deep learning-based sequential recommendation methods are proposed to model complex and long-term dependencies in users' behaviors. Among this category, one research line focuses on designing the effective sequence encoders and backbone networks to encode users' interaction sequence, including GRU (Hidasi et al., 2015), CNNs (Tang & Wang, 2018), Transformer (Kang & McAuley, 2018), and Mamba (Liu et al., 2025b). Building upon these, another research line further introduces advanced models, such as Graph Neural Networks (GNNs) (Chang et al., 2021; Cai et al., 2023; 2022) and generative models (Deldjoo et al., 2024). Among them, generative models have recently attracted significant attention. In particular, DMs Liu et al. (2025a) and large language models (LLMs) (Sheng et al., 2025) have emerged as the two most prominent approaches. DM-based methods will be discussed in detail in Section A.2. With the widespread adoption of large language models (LLMs) across diverse application domains Yang et al. (2025); Li et al. (2025c); Zhang et al. (2024a), they have attracted considerable attention in RSs. LLM-based methods focus on leveraging the open-world knowledge encoded in LLMs to enhance sequential recommendation performance (Harte et al., 2023).

## A.2   DIFFUSION MODELS FOR SEQUENTIAL RECOMMENDATION

In recent years, owing to the strong capability to model complex distributions of user behaviors and item content, diffusion models (DMs) have been widely applied in recommendation scenarios (Wei & Fang, 2025; Lin et al., 2024; Li et al., 2025a), including top-K recommendation (Wang et al., 2023d; Zhao et al., 2024) and multimodal recommendation (Ma et al., 2024c; Li et al., 2025a). In SRec, DM-based recommendation methods can be broadly categorized into two types: next item generation-based methods, and data augmentation-based methods. The former generally employ sequence encoders (e.g., GRU and Transformer) to encode users' context items into condition embeddings, which then guide the generation of next items (Yang et al., 2023b; Liu et al., 2025a; Li et al., 2025b; Cai et al., 2025; Hu et al., 2024; Li et al., 2025b; Ma et al., 2024b; Wang et al., 2024c; Xie et al., 2024). For example, (Yang et al., 2023b; Liu et al., 2025a) utilize Transformer to learn condition embeddings from users' historical interactions, which are then utilized to guide the next-item generation process. The latter category leverages DMs to generate additional interaction data in order to enrich users' interaction sequences and alleviate sequence sparsity. For instance, (Liu et al., 2023; Ma et al., 2024a; Wu et al., 2023b) propose generating pseudo interaction sequences with DMs to mitigate the sequence sparsity problem. Additionally, several methods integrate contrastive learning with diffusion models to generate augmented views, thereby enhancing the training of DM-based recommendation methods (Cui et al., 2024b;a; Qu & Nobuhara, 2025).

Although these methods have achieved remarkable success, they pose a significant risk of generating uncredible content recommendations (e.g., fake news (Wang et al., 2022b; 2024a; Ma et al., 2025), misinformation (Pathak et al., 2023; Fernandez et al., 2024)), which can severely harm both user experience and societal well-being. While (Ma et al., 2025) attempts to leverage DMs to mitigate fake news, its effectiveness is limited under the challenge of scarce labeled data. This limitation motivates us to steer DMs towards credible content recommendation while simultaneously addressing the challenge of learning from only limited annotated data.

## B    More Experimental Details

### B.1    Datasets

In our task setting, we require users' chronological interaction sequences with content items, together with labels indicating whether each item contains uncredible content. However, only a limited number of public datasets fulfill these requirements. In this paper, we utilize three datasets: PolitiFact, GossipCop, and MHMisinfo.

The PolitiFact and GossipCop datasets are derived from the FakeNewsNet repository[4] , which collects data from two well-known fact-checking websites: PolitiFact and GossipCop. These datasets provide user–news interaction sequences along with labels that indicate whether each news article is fake or true. The MHMisinfo dataset is collected from a video-based mental health misinformation dataset[5] , containing users' interaction sequences with videos annotated by whether the videos contain mental health misinformation. Although this dataset records user–video interactions, the original video and image contents are not provided. Therefore, we represent the items using their video descriptions instead of visual features.

Given the high sparsity of these datasets, we adopt a data augmentation strategy following common practice (Yang et al., 2023b;a). Specifically, for each user, we transform their interaction sequence into multiple sub-sequences by treating each item as the target item and the items preceding it as historical context. This transformation increases the number of user–item interaction sequences and enriches the training data. The statistics of these datasets are reported in Table 3. After augmentation, the datasets have more sequences, thereby the recommendation performances of Rec4Mit and HDInt are different from the results reported in (Wang et al., 2022b) and (Wang et al., 2024a).

Table 3: The statistics of the three used datasets after preprocessing.

| Datasets | PolitiFact | GossipCop | MHMisinfo |
|---|---|---|---|
| # Content items | 616 | 9,529 | 3,160 |
| # Credible content items | 306 | 6,792 | 2,815 |
| # Uncredible content items | 310 | 2,737 | 345 |
| # Training sequences | 103,335 | 510,149 | 38,083 |
| # Test sequences | 21,490 | 68,002 | 8,060 |

### B.2    Baseline Descriptions

In this section, we introduce the baseline methods used in our comparison.

**Traditional sequential recommendation methods:**

- **GRU4Rec** (Hidasi et al., 2015) utilizes the Gated Recurrent Unit (GRU) to model the temporal dependencies of items in users' interaction sequences.

- **SASRec** (Kang & McAuley, 2018) employs the Transformer architecture to model the item dependencies in users' interaction sequences. This is one of the most representative sequential recommendation methods.

- **Bert4Rec** (Sun et al., 2019) replaces SASRec's unidirectional Transformer with a bidirectional Transformer architecture to model complex item dependencies. It also introduces a cloze task paradigm for sequential recommendation.

- **LRU4Rec** (Yue et al., 2024) designs linear recurrent units for sequential recommendation. It decomposes linear recurrence operations and proposes recursive parallelization, reducing model size and enabling efficient parallel training.

**Contrastive learning-based sequential recommendation methods:**

- **CL4SRec** (Xie et al., 2022) uses contrastive learning to address the data sparsity problem in sequential recommendation. It designs three sequence augmentation operations for contrastive

---

[4]https://github.com/KaiDMML/FakeNewsNet
[5]https://zenodo.org/records/13191247

learning: item cropping, item masking, and item reordering. Transformer is used as the sequential encoder of CL4SRec.

- **ContraRec** (Wang et al., 2023a) proposes two types of contrastive perspectives to enhance the performance of contrastive learning-based sequential recommendation: context-target contrast and context-context contrast. Transformer is used as the sequential encoder of ContraRec.

**Sequential recommendation methods for mitigating uncredible content:**

- **Rec4Mit** (Wang et al., 2022b) first utilizes a disentangler to extract event- and veracity-aware information, respectively. Thereafter, the event embeddings are utilized to derive users' genuine preferences and predict the next items users may be interested in.

- **HDInt** (Wang et al., 2024a). Similar to Rec4Mit, HDInt is also dedicated to mitigating fake news in recommender systems. HDInt also considers the political bias. We omit this part, since it requires additional data and the political bias is not considered in our task.

- **PRISM** (Ma et al., 2025) proposes a protection-enhanced news recommendation method based on interest-aware sequential modeling. It utilizes DMs' controllable ability to learn user interest and mitigate fake news. However, it assumes all the labels of fake news are fully available, which does not hold in the real world. It is also a DM-based sequential recommendation method.

**DM-based recommendation methods:**

- **DreamRec** (Yang et al., 2023b) assumes that each user has an "oracle" item in mind and selects items that match his ideal item. It uses a Transformer to learn users' preferences, which then serve as the condition for generating the oracle item for each user.

- **DiffuRec** (Li et al., 2023) employs a diffusion model to represent item embeddings in a distribution space and then feeds the embeddings into an approximator to generate target item representations. It argues that the standard objective function of DMs is unsuitable for recommendation tasks and uses cross-entropy loss to optimize model parameters.

- **PreferDiff** (Liu et al., 2025a) proposes a surrogate optimization objective which extend BPR recommendation loss (Rendle et al., 2009) to variational format. Meanwhile, this surrogate optimization objective can also be extended to multiple negative items.

### B.3 EVALUATION METRICS

HR@K and NDCG@K are two commonly used metrics to evaluate the recommendation accuracy, thereby we do not make further introduction for them. Credible Rate (CR@K) is a metric to measure the credibility of a recommendation model. Specifically, it calculates the average rate of the credible content items in the recommendation lists:

$$\text{CR@K} = \frac{1}{|\mathcal{S}_{test}|} \sum_{s \in \mathcal{S}_{test}} \frac{K - |\mathcal{R}_s \cap \mathcal{I}_{unc}^{Ground-truth}|}{K}, \tag{13}$$

where $\mathcal{S}_{test}$ is the test set of sequences. $\mathcal{R}_s$ is the recommendation list for sequence $s$. $\mathcal{I}_{unc}^{Ground-truth}$ denotes the ground-truth set of uncredible items. $|\mathcal{R}_s \cap \mathcal{I}_{neg}^{Ground-truth}|$ calculates the number of uncredible items in the recommendation list. The higher value of CR@K means the better performance in delivering credible recommendations.

In addition, we test how our methods perform in terms of both accurate and credible recommendations, we design a combined metric HC@K (i.e., combining HR@K and CR@K). Formally, HC@K is calculated as follows:

$$\text{HC@K} = \frac{2 \times \text{HR@K} \times (\text{CR@K}/2)}{\text{HR@K} + (\text{CR@K}/2)}. \tag{14}$$

This combined metric is inspired by the F1-score, which combines precision rate and recall rate. To note that, since the values of HR@K and CR@K are not on the same scale, we divide CR@K with a factor of 2 to rescale it into a similar value level with HR@K. This adjustment ensures a fair combination; otherwise, the metric with a much smaller magnitude would disproportionately dominate the combined score.

## B.4 Implementation Details

In this paper, we consider a more challenging and realistic scenario in which only a small proportion of uncredible items are verified. To simulate this setting, we randomly select 20% of the uncredible items with available labels during the training process, while the labels of the remaining items are treated as unknown. It is similar to the semi-supervised setting. In contrast, during the testing stage, all content labels are provided to enable an accurate evaluation.

The items in PolitiFact and GossipCop are news articles, and we use their textual descriptions as item content. In MHMisinfo, although the items are videos, only textual descriptions are available; thus, we can only rely on the textual descriptions for content representation. We encode these textual descriptions into language embeddings using LLaMA2-7B (Touvron et al., 2023), and further project them into a lower dimension through an MLP. Following (Liu et al., 2025a), we fix the transformed embedding dimension at 3072 for all DM-based methods, as they exhibit strong performance only with higher embedding sizes. For other methods, the embedding size is set to 64. We also experimented with larger embedding sizes for these methods, but observed little or no performance gain, and even performance drops for some methods, consistent with the findings in (Liu et al., 2025a).

In our implementation, we select Transformer as our sequence encoder. Following the standard configuration (Vaswani et al., 2017), the Transformer architecture in our implementation includes multi-head attention, position-wise feed-forward network, layer normalization, and dropout.

For our method Disco, the hyperparameter $w$ is tuned within $\{0.5, 1, 1.5, 2, 5\}$. We fix $m$ at 10,000 and tune $\gamma$ within $\{0.1, 0.2, 0.3, 0.4, 0.5\}$ to control the maximum selection ratio as well as the growth rate of the current selection ratio. The maximum number of diffusion steps is fixed at 2,000 and the DDIM step is set to 100, following the settings of (Liu et al., 2025a). For all DM-based methods, we utilize a linear schedule for $\beta_t$ in range [0.0001, 0.02]. In our implementation, we do not use a classifier-free guidance (Ho & Salimans), since we found it does not influence much to the performance of Disco. In our implementation, we found that the singular values in $\Lambda$ are relatively large; therefore, the threshold for constructing the null space of uncredible features is fixed at 3 for all datasets in our experiments. We search learning rate in range $\{$1e-5, 5e-5, 1e-4, 5e-4, 1e-3$\}$. The batch size is searched in $\{2048 \times 2^i\}_{i=0,1,2,3}$. The model parameters are initialized using normal initialization and optimized by AdamW (Loshchilov & Hutter, 2017). The hyperparameter settings of baseline methods are reported in Table 4. All experiments are conducted on an NVIDIA A40 GPU with 48 GB of memory. Each method is run five times, and we report the average performance along with the standard deviation.

Table 4: The hyperparameter settings of baseline methods.

| Methods | Hyperparameter searching space |
|---|---|
| GRU4Rec | lr$\sim\{$1e-2, 5e-2, 1e-3, 5e-3, 1e-4$\}$, weight decay=0 |
| SASRec | lr$\sim\{$1e-2, 5e-2, 1e-3, 5e-3, 1e-4$\}$, weight decay=0 |
| Bert4Rec | lr$\sim\{$1e-2, 5e-2, 1e-3, 5e-3, 1e-4$\}$, weight decay=0, mask probability$\sim\{$0.2, 0.4, 0.6, 0.8$\}$ |
| LRURec | lr$\sim\{$1e-2, 5e-2, 1e-3, 5e-3, 1e-4$\}$, weight decay=0, dropout rate$\sim\{$0.2, 0.4, 0.6, 0.8$\}$ |
| CL4SRec | lr$\sim\{$1e-2, 5e-2, 1e-3, 5e-3, 1e-4$\}$, weight decay=0, mask/reorder/crop proportion$\sim\{$0.2, 0.4, 0.6, 0.8$\}$, $\lambda \sim\{$0.1, 0.3, ..., 0.9$\}$ |
| ContraRec | lr$\sim\{$1e-2, 5e-2, 1e-3, 5e-3, 1e-4$\}$, weight decay=0, mask/reorder/crop proportion$\sim\{$0.2, 0.4, 0.6, 0.8$\}$, $\tau_1, \tau_2 \sim\{$0.1, 0.2, ..., 1$\}$, $\gamma \sim\{$0, 0.01, 0.1, 1, 5, 10$\}$ |
| Rec4Mit | lr$\sim\{$1e-2, 5e-2, 1e-3, 5e-3, 1e-4$\}$, weight decay=0, $k \sim\{$2, 4, ..., 20$\}$ |
| HDInt | lr$\sim\{$1e-2, 5e-2, 1e-3, 5e-3, 1e-4$\}$, weight decay=0, $\lambda \sim\{$1, 2, ..., 10$\}$, $\gamma \sim\{$2, 4, 6, 8, 10$\}$ |
| PRISM | lr$\sim\{$1e-2, 5e-2, 1e-3, 5e-3, 1e-4$\}$, weight decay=0, $T \sim\{$500, 1000, 1500, 2000$\}$, $w \sim\{$0, 2, 4, 6, 8$\}$, $\lambda_{OT}, \lambda_c, \lambda_r, \lambda_{rec} \sim\{$0.2, 0.4, 0.6, 0.8, 1$\}$, embedding size$\sim\{$64, 128, 256, 512, 1024, 2048, 3072$\}$ |
| DreamRec | lr$\sim\{$1e-2, 5e-2, 1e-3, 5e-3, 1e-4$\}$, weight decay=0, $T \sim\{$500, 1000, 1500, 2000$\}$, $w \sim\{$0, 2, 4, 6, 8$\}$, embedding size$\sim\{$64, 128, 256, 512, 1024, 2048, 3072$\}$ |
| DiffuRec | lr$\sim\{$1e-2, 5e-2, 1e-3, 5e-3, 1e-4$\}$, weight decay=0, $T \sim\{$16, 32, 64, 128$\}$, $\delta$=0.001, embedding size$\sim\{$64, 128, 256, 512, 1024, 2048, 3072$\}$ |
| PreferDiff | lr$\sim\{$1e-2, 5e-2, 1e-3, 5e-3, 1e-4$\}$, weight decay=0, $T \sim\{$500, 1000, 1500, 2000$\}$, $w \sim\{$0, 2, 4, 6, 8$\}$, $\lambda \sim\{$0.2, 0.4, 0.6, 0.8$\}$, embedding size$\sim\{$64, 128, 256, 512, 1024, 2048, 3072$\}$ |

## B.5 More Hyperparameter Experiments

The hyperparameter $\gamma$ controls the selection ratio of potential uncredible items. We evaluate the performance of Disco (using combined metric HC@5) under different values of $\gamma$ in range $\{$0.1, 0.2, 0.3, 0.4, 0.5$\}$. As shown in Figure 5, Disco achieves the best performance when fixing $w = 0.1$ on PolitiFact and GossipCop, and $w = 0.4$ on MHMisinfo. Lower values prevent the model

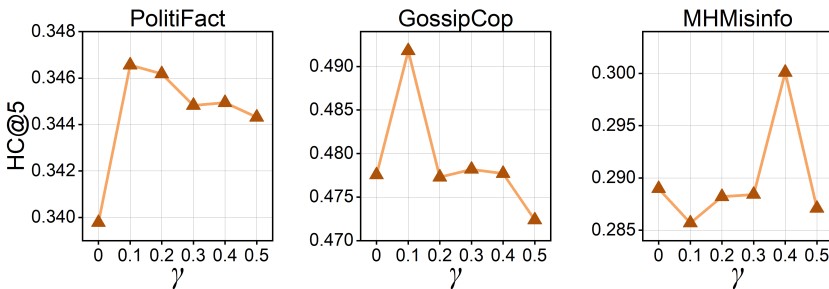

Figure 5: Effect of $\gamma$ on `Disco`.

from effectively capturing potentially uncredible items, while higher values may introduce excessive noise, both of which degrade model performance.

### B.6 TIME EFFICIENCY ANALYSIS

We conduct experiments to evaluate the training and inference cost (in seconds) of our model `Disco` and four DM-based methods under the same batch size. As shown in Table 5, the training cost of `Disco` is relatively higher than DreamRec and PreferDiff, mainly due to our additional designs for credible content recommendation, including content disentanglement and credible subspace projection. This is acceptable due to the higher recommendation accuracy and credibility of our proposed method. Our training cost is much lower than that of DiffuRec an PRISM. As for inference cost, our proposed method `Disco` demonstrates the highest efficiency. This is because we adopt DDIM (Song et al., 2021) as our generation strategy, which is more efficient than the DDPM (Ho et al., 2020) paradigm employed by DreamRec, DiffuRec and PRISM. Even compared with PreferDiff, which also adopts DDIM, `Disco` also exhibits higher efficiency. It is because we do not employ classifier-free guidance in our implementation, since it has limited influence on our model while incurring additional time consumption. Although `Disco` requires disentangling item embeddings first in the inference stage, it's inference cost remains comparable to PreferDiff on GossipCop dataset, which contain large number of items.

Table 5: Time cost (s) of different models on PolitiFact, GossipCop, and MHMisinfo.

| Datasets | Time cost (s) | DreamRec | DiffuRec | PRISM | PreferDiff | Disco |
|---|---|---|---|---|---|---|
| PolitiFact | Training/epoch | 9.4 | 74.8 | 25.3 | 11.3 | 12.6 |
| | Inference | 278.2 | 224.3 | 994.3 | 15.2 | 10.5 |
| GossipCop | Training/epoch | 43.3 | 376.9 | 137.6 | 58.8 | 73.1 |
| | Inference | 956.8 | 783.1 | 3223.6 | 127.9 | 133.5 |
| MHMisinfo | Training/epoch | 3.3 | 26.9 | 8.9 | 3.9 | 4.2 |
| | Inference | 107.6 | 87.1 | 372.9 | 8.7 | 7.7 |

### B.7 DISCUSSION ON EMBEDDING DIMENSION

As pointed out in our paper, DM-based methods can only achieve strong performance when the embedding dimension is high. To demonstrate the necessity of using high embedding dimension (i.e., 3072) for all DM-based methods, we conducted experiments on DM-based methods when fixing embedding dimension to 64. The results are shown in Table 6:

As shown in Table 6, not only `Disco` but also other DM-based recommendation methods (PRISM, DreamRec, and PreferDiff) experience a substantial performance drop when the embedding size is reduced to 64. This observation is consistent with the findings reported in [1] and further validates the necessity of using high-dimensional embeddings in DM-based recommender systems.

Although a higher embedding dimension increases the per-epoch training time, it also provides the benefit of significantly faster convergence. In our revised manuscript, we include a figure illustrating the convergence curves of Disco and SASRec. As presented in Figure 6, Disco converges much

Table 6: Performance comparison under embedding dimension 64 for DM-based methods.

| Methods | HR@5 | HR@10 | NDCG@5 | NDCG@10 | CR@5 | CR@10 | HC@5 | HC@10 |
|---|---|---|---|---|---|---|---|---|
| **PolitiFact** | | | | | | | | |
| GRU4Rec | 0.2142 | 0.3390 | 0.1463 | 0.1863 | 0.9266 | 0.9122 | 0.2929 | 0.3889 |
| SASRec | 0.2158 | 0.3519 | 0.1386 | 0.1823 | 0.9059 | 0.9028 | 0.2923 | 0.3955 |
| Bert4Rec | 0.2191 | 0.3473 | 0.1472 | 0.1883 | 0.9127 | 0.9045 | 0.2960 | 0.3929 |
| CL4SRec | 0.2247 | 0.3527 | 0.1508 | 0.1919 | 0.9132 | 0.9027 | 0.3012 | 0.3960 |
| PRISM (emb_size=3072) | 0.1927 | 0.2758 | 0.1348 | 0.1615 | 0.9325 | 0.9178 | 0.2727 | 0.3446 |
| PRISM (emb_size=64) | 0.0806 | 0.1261 | 0.0569 | 0.0715 | 0.7807 | 0.7222 | 0.1336 | 0.1869 |
| DreamRec (emb_size=3072) | 0.2416 | 0.3287 | 0.1767 | 0.2047 | 0.8620 | 0.8437 | 0.3054 | 0.3661 |
| DreamRec (emb_size=64) | 0.0814 | 0.1074 | 0.0651 | 0.0734 | 0.5744 | 0.5605 | 0.1268 | 0.1553 |
| PreferDiff (emb_size=3072) | 0.2531 | 0.3554 | 0.1818 | 0.2147 | 0.8925 | 0.8981 | 0.3228 | 0.3968 |
| PreferDiff (emb_size=64) | 0.1227 | 0.1841 | 0.0882 | 0.1078 | 0.8934 | 0.8788 | 0.1925 | 0.2595 |
| Disco (emb_size=3072) | 0.2678 | 0.3775 | 0.1983 | 0.2336 | 0.9823 | 0.9425 | 0.3466 | 0.4192 |
| Disco (emb_size=64) | 0.1171 | 0.1962 | 0.1107 | 0.1422 | 0.9916 | 0.9665 | 0.1895 | 0.2791 |
| **GossipCop** | | | | | | | | |
| GRU4Rec | 0.2226 | 0.3194 | 0.1466 | 0.1778 | 0.8864 | 0.8706 | 0.2957 | 0.3678 |
| SASRec | 0.3078 | 0.4706 | 0.1607 | 0.2135 | 0.8743 | 0.8526 | 0.3612 | 0.4473 |
| Bert4Rec | 0.2372 | 0.3711 | 0.1338 | 0.1770 | 0.8764 | 0.8587 | 0.3078 | 0.3981 |
| CL4SRec | 0.2898 | 0.4100 | 0.1784 | 0.2174 | 0.8938 | 0.8932 | 0.3516 | 0.4275 |
| PRISM (emb_size=3072) | 0.2948 | 0.3447 | 0.2301 | 0.2463 | 0.8806 | 0.8733 | 0.3531 | 0.3852 |
| PRISM (emb_size=64) | 0.0023 | 0.0034 | 0.0015 | 0.0018 | 0.6570 | 0.6940 | 0.0046 | 0.0067 |
| DreamRec (emb_size=3072) | 0.4619 | 0.5501 | 0.3415 | 0.3704 | 0.8464 | 0.8336 | 0.4415 | 0.4742 |
| DreamRec (emb_size=64) | 0.0036 | 0.0049 | 0.0027 | 0.0031 | 0.5791 | 0.5903 | 0.0071 | 0.0096 |
| PreferDiff (emb_size=3072) | 0.4969 | 0.6022 | 0.3655 | 0.3999 | 0.8307 | 0.8228 | 0.4523 | 0.4887 |
| PreferDiff (emb_size=64) | 0.0084 | 0.0139 | 0.0053 | 0.0070 | 0.6542 | 0.7400 | 0.0164 | 0.0268 |
| Disco (emb_size=3072) | 0.5236 | 0.6143 | 0.3996 | 0.4292 | 0.9272 | 0.9039 | 0.4918 | 0.5207 |
| Disco (emb_size=64) | 0.0087 | 0.0162 | 0.0053 | 0.0077 | 0.7993 | 0.7993 | 0.0170 | 0.0311 |
| **MHMisinfo** | | | | | | | | |
| GRU4Rec | 0.1151 | 0.1894 | 0.0760 | 0.0998 | 0.8380 | 0.8608 | 0.1803 | 0.2624 |
| SASRec | 0.1485 | 0.2592 | 0.0826 | 0.1179 | 0.8339 | 0.8915 | 0.2190 | 0.3276 |
| Bert4Rec | 0.1391 | 0.2299 | 0.0847 | 0.1138 | 0.8162 | 0.8786 | 0.2074 | 0.3017 |
| CL4SRec | 0.1734 | 0.2621 | 0.1101 | 0.1387 | 0.8577 | 0.9081 | 0.2469 | 0.3323 |
| PRISM (emb_size=3072) | 0.1700 | 0.2339 | 0.1181 | 0.1388 | 0.8398 | 0.8919 | 0.2418 | 0.3065 |
| PRISM (emb_size=64) | 0.0239 | 0.0295 | 0.0190 | 0.0208 | 0.7095 | 0.7317 | 0.0448 | 0.0546 |
| DreamRec (emb_size=3072) | 0.1819 | 0.2426 | 0.1313 | 0.1509 | 0.9002 | 0.8952 | 0.2633 | 0.3176 |
| DreamRec (emb_size=64) | 0.0282 | 0.0347 | 0.0233 | 0.0254 | 0.8281 | 0.8901 | 0.0528 | 0.0644 |
| PreferDiff (emb_size=3072) | 0.1974 | 0.2620 | 0.1389 | 0.1598 | 0.8693 | 0.8874 | 0.2713 | 0.3294 |
| PreferDiff (emb_size=64) | 0.0325 | 0.0380 | 0.0290 | 0.0308 | 0.8290 | 0.8989 | 0.0603 | 0.0701 |
| Disco (emb_size=3072) | 0.2215 | 0.2822 | 0.1580 | 0.1778 | 0.9305 | 0.9264 | 0.3000 | 0.3507 |
| Disco (emb_size=64) | 0.0123 | 0.0572 | 0.0074 | 0.0213 | 0.9968 | 0.9873 | 0.0240 | 0.1025 |

more rapidly than the non-DM-based method SASRec (embedding size = 64). Specifically, Disco reaches its best performance at approximately the 40-th epoch, whereas SASRec requires around 400 epochs. This fast convergence rate of Disco partially offsets the additional computational cost introduced by high-dimensional embeddings.

In addition, we conducted further experiments to demonstrate that Disco can still achieve superior performance compared with non-DM-based methods when the embedding dimension is restricted to 64, as long as a minor modification is applied to the overall optimization objective. In particular, we augment Disco's loss function with an additional Cross-Entropy term:

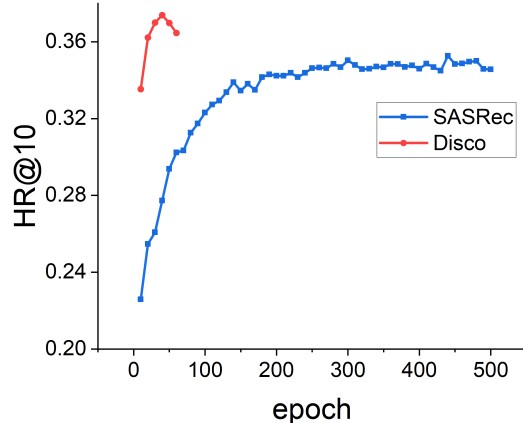

Figure 6: The performance convergence curve of SASRec and Disco on PolitiFact dataset.

Table 7: Performance comparison of Disco (optimized by $\mathcal{L}_{\texttt{Disco}^*}$) and non-DM recommendation methods on PolitiFact, GossipCop, and MHMisinfo when embedding dimension is se to 64.

| Methods | HR@5 | HR@10 | NDCG@5 | NDCG@10 | CR@5 | CR@10 | HC@5 | HC@10 |
|---|---|---|---|---|---|---|---|---|
| | | | | PolitiFact | | | | |
| GRU4Rec | 0.2142 | 0.3390 | 0.1463 | 0.1863 | 0.9266 | 0.9122 | 0.2929 | 0.3889 |
| SASRec | 0.2158 | 0.3519 | 0.1386 | 0.1823 | 0.9059 | 0.9028 | 0.2923 | 0.3955 |
| Bert4Rec | 0.2191 | 0.3473 | 0.1472 | 0.1883 | 0.9127 | 0.9045 | 0.2960 | 0.3929 |
| CL4SRec | 0.2247 | 0.3527 | 0.1508 | 0.1919 | 0.9132 | 0.9027 | 0.3012 | 0.3960 |
| **Disco (emb_size=64, $\mathcal{L}_{\texttt{Disco}^*}$)** | **0.2335** | **0.3555** | **0.1642** | **0.2034** | **0.9316** | **0.9213** | **0.3111** | **0.4013** |
| | | | | GossipCop | | | | |
| GRU4Rec | 0.2226 | 0.3194 | 0.1466 | 0.1778 | 0.8864 | 0.8706 | 0.2957 | 0.3678 |
| SASRec | 0.3078 | 0.4706 | 0.1607 | 0.2135 | 0.8743 | 0.8526 | 0.3612 | 0.4473 |
| Bert4Rec | 0.2372 | 0.3711 | 0.1338 | 0.1770 | 0.8764 | 0.8587 | 0.3078 | 0.3981 |
| CL4SRec | 0.2898 | 0.4100 | 0.1784 | 0.2174 | 0.8938 | 0.8932 | 0.3516 | 0.4275 |
| **Disco (emb_size=64, $\mathcal{L}_{\texttt{Disco}^*}$)** | **0.4250** | **0.5060** | **0.3320** | **0.3583** | **0.9151** | **0.9080** | **0.4407** | **0.4786** |
| | | | | MHMisinfo | | | | |
| GRU4Rec | 0.1151 | 0.1894 | 0.0760 | 0.0998 | 0.8380 | 0.8608 | 0.1803 | 0.2624 |
| SASRec | 0.1485 | 0.2592 | 0.0826 | 0.1179 | 0.8339 | 0.8915 | 0.2190 | 0.3276 |
| Bert4Rec | 0.1391 | 0.2299 | 0.0847 | 0.1138 | 0.8162 | 0.8786 | 0.2074 | 0.3017 |
| CL4SRec | 0.1734 | 0.2621 | 0.1101 | 0.1387 | 0.8577 | 0.9081 | 0.2469 | 0.3323 |
| **Disco (emb_size=64, $\mathcal{L}_{\texttt{Disco}^*}$)** | **0.1856** | **0.2660** | **0.1214** | **0.1475** | **0.8669** | **0.9129** | **0.2599** | **0.3361** |

$$\mathcal{L}_{\texttt{Disco}^*} = \mathcal{L}_{\texttt{Disco}} - \log \left( \frac{\exp(f_\theta(\tilde{\mathbf{e}}_n^t, \mathbf{c}^{pre}, t) \cdot \mathbf{e}_n^\top)}{\sum_{i \in \mathcal{I}} \exp(f_\theta(\tilde{\mathbf{e}}_n^t, \mathbf{c}^{pre}, t) \cdot \mathbf{e}_i^\top)} \right). \tag{15}$$

The added term encourages the diffusion network $f_\theta$ to align its predictions more closely to the target items than with other items. Using this enhanced loss function $\mathcal{L}_{\texttt{Disco}^*}$, Disco can achieve better performance than non-DM based methods. The comparison results are reported in Table 7.

As shown in the Table 7, Disco can achieve better performance than non-DM recommendation methods with only a minor adjustment to its overall loss. This improvement arises because adding a discriminative loss (i.e., Cross Entropy loss) to the generative loss ($\mathcal{L}_{\texttt{Disco}}$) can partially mitigate the dimensionality limitations inherent to diffusion models.

However, this practice will transform a purely generative model into a discriminative one. Our work does not make such a compromise. Nevertheless, the results clearly suggest that our model retains strong potential to surpass non-DM-based recommenders even when operating with substantially reduced embedding dimensions. This highlights the potential of `Disco` to achieve an effective trade-off between efficiency and performance.

### B.8 Discussion on Diffusion Step $T$

In this section, we conducted experiments to investigate whether `Disco` still strong performance when the diffusion step $T$ is much smaller. The experimental results are reported in Table 8.

From Table 8, we can have the following observations:

- As the diffusion step $T$ increases, the performance of our proposed model, `Disco`, improves.

- `Disco` maintains superior performance compared with non-DM methods even at much smaller diffusion step ($T$=100 for PolitiFact and GossiCop, and $T$=500 for MHMisinfo).

Table 8: Performance comparison of Disco and non-DM recommendation methods under different diffusion steps $T$.

| Methods | HR@5 | HR@10 | NDCG@5 | NDCG@10 | CR@5 | CR@10 | HC@5 | HC@10 |
|---|---|---|---|---|---|---|---|---|
| | | | **PolitiFact** | | | | | |
| GRU4Rec | 0.2142 | 0.3390 | 0.1463 | 0.1863 | 0.9266 | 0.9122 | 0.2929 | 0.3889 |
| SASRec | 0.2158 | 0.3519 | 0.1386 | 0.1823 | 0.9059 | 0.9028 | 0.2923 | 0.3955 |
| Bert4Rec | 0.2191 | 0.3473 | 0.1472 | 0.1883 | 0.9127 | 0.9045 | 0.2960 | 0.3929 |
| CL4SRec | 0.2247 | 0.3527 | 0.1508 | 0.1919 | 0.9132 | 0.9027 | 0.3012 | 0.3960 |
| `Disco` (Diffusion step $T$=100) | 0.2494 | 0.3724 | 0.1751 | 0.2146 | 0.9488 | 0.9352 | 0.3269 | 0.4146 |
| `Disco` (Diffusion step $T$=200) | 0.2602 | 0.3752 | 0.1842 | 0.2211 | 0.9427 | 0.9340 | 0.3353 | 0.4161 |
| `Disco` (Diffusion step $T$=500) | 0.2591 | 0.3828 | 0.1811 | 0.2208 | 0.9434 | 0.9272 | 0.3345 | 0.4193 |
| `Disco` (Diffusion step $T$=1000) | 0.2525 | 0.3784 | 0.1752 | 0.2156 | 0.9488 | 0.9369 | 0.3296 | 0.4186 |
| `Disco` (Diffusion step $T$=2000) | 0.2678 | 0.3775 | 0.1983 | 0.2336 | 0.9823 | 0.9425 | 0.3466 | 0.4192 |
| | | | **GossipCop** | | | | | |
| GRU4Rec | 0.2226 | 0.3194 | 0.1466 | 0.1778 | 0.8864 | 0.8706 | 0.2957 | 0.3678 |
| SASRec | 0.3078 | 0.4706 | 0.1607 | 0.2135 | 0.8743 | 0.8526 | 0.3612 | 0.4473 |
| Bert4Rec | 0.2372 | 0.3711 | 0.1338 | 0.1770 | 0.8764 | 0.8587 | 0.3078 | 0.3981 |
| CL4SRec | 0.2898 | 0.4100 | 0.1784 | 0.2174 | 0.8938 | 0.8932 | 0.3516 | 0.4275 |
| `Disco` (Diffusion step $T$=100) | 0.4603 | 0.5479 | 0.3419 | 0.3705 | 0.9304 | 0.9252 | 0.4627 | 0.5016 |
| `Disco` (Diffusion step $T$=200) | 0.4759 | 0.5659 | 0.3537 | 0.3831 | 0.9242 | 0.9199 | 0.4689 | 0.5075 |
| `Disco` (Diffusion step $T$=500) | 0.4867 | 0.5798 | 0.3621 | 0.3925 | 0.9258 | 0.9169 | 0.4745 | 0.5120 |
| `Disco` (Diffusion step $T$=1000) | 0.4936 | 0.5956 | 0.3651 | 0.3984 | 0.9202 | 0.9039 | 0.4763 | 0.5139 |
| `Disco` (Diffusion step $T$=2000) | 0.5236 | 0.6143 | 0.3996 | 0.4292 | 0.9272 | 0.9039 | 0.4918 | 0.5207 |
| | | | **MHMisinfo** | | | | | |
| GRU4Rec | 0.1151 | 0.1894 | 0.0760 | 0.0998 | 0.8380 | 0.8608 | 0.1803 | 0.2624 |
| SASRec | 0.1485 | 0.2592 | 0.0826 | 0.1179 | 0.8339 | 0.8915 | 0.2190 | 0.3276 |
| Bert4Rec | 0.1391 | 0.2299 | 0.0847 | 0.1138 | 0.8162 | 0.8786 | 0.2074 | 0.3017 |
| CL4SRec | 0.1734 | 0.2621 | 0.1101 | 0.1387 | 0.8577 | 0.9081 | 0.2469 | 0.3323 |
| `Disco` (Diffusion step $T$=100) | 0.1378 | 0.1998 | 0.0914 | 0.1111 | 0.8526 | 0.8783 | 0.2083 | 0.2746 |
| `Disco` (Diffusion step $T$=200) | 0.1393 | 0.2191 | 0.0921 | 0.1178 | 0.9161 | 0.9209 | 0.2136 | 0.2969 |
| `Disco` (Diffusion step $T$=500) | 0.1819 | 0.2610 | 0.1299 | 0.1553 | 0.9076 | 0.9144 | 0.2597 | 0.3323 |
| `Disco` (Diffusion step $T$=1000) | 0.2144 | 0.2638 | 0.1547 | 0.1708 | 0.9379 | 0.9311 | 0.2943 | 0.3358 |
| `Disco` (Diffusion step $T$=2000) | 0.2215 | 0.2822 | 0.1580 | 0.1778 | 0.9305 | 0.9264 | 0.3000 | 0.3507 |

## B.9 Discussion on Various Ratios of Available Credibility Labels

we conduct additional experiments under different credibility label ratio of uncredible content (i.e., 5%, 10%, 20%, 30%, 50%). The experimental results are reported in Table 9. From the results reported in the Table, we can have the following findings:

- **Finding 1:** As the credibility label ratio increases, the recommendation credibility (CR) improves steadily. This is because a larger number of credibility labels enables the construction of a more comprehensive and accurate credible subspace, allowing uncredible content to be mitigated more effectively.

- **Finding 2:** The recommendation accuracy fluctuates only slightly within a narrow range across different label ratios. This stability is attributed to our proposed disentangled diffusion model, which effectively mitigates uncredible content while preserving users' preference-related information, thereby maintaining high recommendation accuracy.

Table 9: Performance comparison under different credibility label ratios.

| Label ratios | HR@5 | HR@10 | NDCG@5 | NDCG@10 | CR@5 | CR@10 | HC@5 | HC@10 |
|---|---|---|---|---|---|---|---|---|
| | | | | **PolitiFact** | | | | |
| 5% | 0.2617 | 0.3869 | 0.1819 | 0.2222 | 0.9422 | 0.9279 | 0.3365 | 0.4219 |
| 10% | 0.2541 | 0.3836 | 0.1768 | 0.2184 | 0.9476 | 0.9357 | 0.3308 | 0.4216 |
| 20% | 0.2678 | 0.3775 | 0.1983 | 0.2336 | 0.9823 | 0.9425 | 0.3466 | 0.4192 |
| 30% | 0.2704 | 0.3838 | 0.1980 | 0.2345 | 0.9829 | 0.9518 | 0.3489 | 0.4249 |
| 50% | 0.2678 | 0.3832 | 0.1923 | 0.2294 | 0.9842 | 0.9486 | 0.3468 | 0.4239 |
| | | | | **GossipCop** | | | | |
| 5% | 0.5179 | 0.6021 | 0.4003 | 0.4279 | 0.9261 | 0.8726 | 0.4889 | 0.5060 |
| 10% | 0.5290 | 0.6115 | 0.4089 | 0.4359 | 0.9266 | 0.8764 | 0.4940 | 0.5105 |
| 20% | 0.5236 | 0.6143 | 0.3996 | 0.4292 | 0.9272 | 0.9039 | 0.4918 | 0.5207 |
| 30% | 0.5196 | 0.6141 | 0.3947 | 0.4255 | 0.9278 | 0.9101 | 0.4902 | 0.5227 |
| 50% | 0.5151 | 0.6069 | 0.3953 | 0.4253 | 0.9284 | 0.9176 | 0.4883 | 0.5226 |
| | | | | **MHMisinfo** | | | | |
| 5% | 0.2127 | 0.2762 | 0.1500 | 0.1705 | 0.9149 | 0.9114 | 0.2904 | 0.3439 |
| 10% | 0.2112 | 0.2798 | 0.1506 | 0.1728 | 0.9217 | 0.9152 | 0.2897 | 0.3473 |
| 20% | 0.2215 | 0.2822 | 0.1580 | 0.1778 | 0.9305 | 0.9264 | 0.3001 | 0.3507 |
| 30% | 0.2207 | 0.2836 | 0.1551 | 0.1755 | 0.9303 | 0.9244 | 0.2994 | 0.3515 |
| 50% | 0.2134 | 0.2715 | 0.1533 | 0.1721 | 0.9331 | 0.9279 | 0.2929 | 0.3425 |

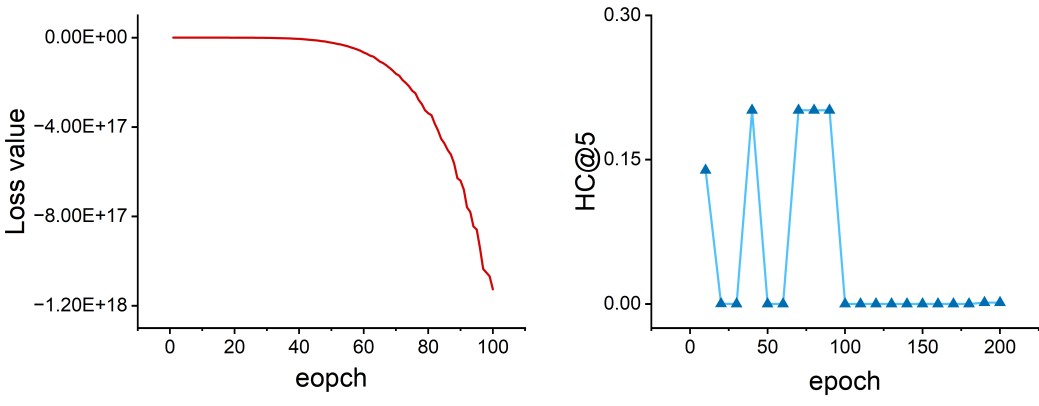

Figure 7: The loss and performance (HC@5) curves of variant w/o CE on PolitiFact dataset.

### B.10 Discussion on the Instability of Ablation Variant w/o CE

In this section, we conduct an empirical study on the convergence of the ablation variant "w/o CE". As show in Figure 7, we observe that the variant w/o CE (i.e., not replacing the MSE error with cosine error) leads to extremely unstable training and performance. Specifically, the loss rapidly collapses to an extremely small value (around $-1.2 \times 10^{18}$), and the performance (HR@5) exhibits severe fluctuations. These results verify the necessity of replacing the MSE error with the cosine error to ensure stable optimization.

## C Why DMs pose a danger of generating uncredible content recommendation?

In this section, we empirically and theoretically analyze why existing DM-based recommendation methods risk generating uncredible recommendations.

### C.1 Empirical Findings

In DM-based recommendation methods, the condition and diffusion target are two critical factors. In this section, we conduct experiments to examine how they influence the recommendation credibility of DM-based methods. Specifically, we divide the training dataset into four subsets based on whether the context items or the diffusion target (i.e., target items) contain uncredible content. We use ✓ to denote that context items or target items contain uncredible content, and ✗ to denote the opposite. After this dataset division, we train two representative DM-based recommendation methods (DreamRec (Yang et al., 2023b) and PreferDiff (Liu et al., 2025a)) on each subset. From the results reported in Table 10, we can find that these two factors indeed affect the recommendation credibility of DM-based methods. We refer to these two factors as uncredible condition and uncredible diffusion target.

- **Uncredible condition.** When controlling the diffusion target, if the context items contain uncredible content that leads to an uncredible condition, the credibility metric CR@10 (i.e., credible Rate) decreases to some extent for both DreamRec and PreferDiff across the PolitiFact and GossipCop datasets. This finding indicates that the uncredible condition is a factor contributing to the risk of DMs generating uncredible recommendation results.
- **Uncredible diffusion target.** When controlling the condition, if the diffusion target is an uncredible item (i.e., an uncredible diffusion target), CR@10 drops to an extremely low level. This further emphasizes that the uncredible diffusion target is another key contributing factor.

Apart from these two findings, we also observe that training with the complete datasets yields worse recommendation credibility compared to the subset where neither the condition nor the diffusion target contains uncredible items. This further validates that the uncredible condition and the uncredible diffusion target are indeed the key contributing factors that place DM-based recommendation methods at risk of generating uncredible recommendation results.

Moreover, **although simply removing uncredible items from the datasets can improve recommendation credibility, it significantly deteriorates recommendation accuracy**. This is because uncredible items may also reflect users' genuine preferences, thereby discarding them restricts the model's ability to accurately learn users' true interests. Therefore, it is crucial to design advanced models that can mitigate the recommendation of uncredible content while simultaneously preserving high recommendation accuracy. This is the motivation and research significance of our proposed model, `Disco`.

### C.2 Theoretical Analysis

**Proof: Uncredible condition can enhance DM's generation of uncredible results**

The training of a conditional DM is to maximize $\mathbb{E}_{p_{data}(\mathbf{e}_n, \mathbf{c})} [\log p_\theta(\mathbf{e}_n | \mathbf{c})]$, where $\mathbf{e}_n$ is the diffusion target (i.e., the last item in a user's interaction sequence) and $\mathbf{c}$ is the condition. This training objective pushes the generation toward the real data distribution.

Table 10: Performance comparison of DreamRec and PreferDiff under different settings of uncredible content items in condition and diffusion target on PolitiFact and GossipCop datasets. Best results are highlighted in bold.

| Methods | Whether contain uncredible content items? | | Politi | | | | Gossip | | | |
|---|---|---|---|---|---|---|---|---|---|---|
| | condition | diffusion target | HR@10 | NDCG@10 | CR@10 | HC@10 | HR@10 | NDCG@10 | CR@10 | HC@10 |
| DreamRec | Training with complete dataset | | **0.3287** | **0.2047** | 0.8437 | **0.3661** | **0.5501** | **0.3704** | 0.8336 | 0.4742 |
| | ✗ | ✗ | 0.2674 | 0.1571 | **0.9935** | 0.3477 | 0.4658 | 0.3160 | **0.9771** | **0.4769** |
| | ✗ | ✓ | 0.0577 | 0.0409 | 0.1888 | 0.0716 | 0.0372 | 0.0284 | 0.0522 | 0.0307 |
| | ✓ | ✗ | 0.2671 | 0.1541 | 0.9875 | 0.3467 | 0.1927 | 0.1368 | 0.9340 | 0.2728 |
| | ✓ | ✓ | 0.0684 | 0.0413 | 0.0806 | 0.0507 | 0.0539 | 0.0404 | 0.0450 | 0.0317 |
| PreferDiff | Training with complete dataset | | **0.3554** | **0.2147** | 0.8981 | **0.3968** | **0.6022** | **0.3999** | 0.8228 | **0.4887** |
| | ✗ | ✗ | 0.3035 | 0.1915 | **0.9591** | 0.3717 | 0.5036 | 0.3657 | **0.9315** | 0.4839 |
| | ✗ | ✓ | 0.0557 | 0.0410 | 0.1073 | 0.0547 | 0.0407 | 0.0304 | 0.0833 | 0.0412 |
| | ✓ | ✗ | 0.2625 | 0.1553 | 0.8561 | 0.3254 | 0.2074 | 0.1454 | 0.9076 | 0.2847 |
| | ✓ | ✓ | 0.0568 | 0.0385 | 0.0837 | 0.0482 | 0.0573 | 0.0421 | 0.0254 | 0.0208 |

When an uncredible content-related condition $\mathbf{c}^{unc}$ is utilized to guide the generation process, the model aims to maximize $\mathbb{E}_{p_{data}(\mathbf{e}_n,\mathbf{c}^{unc})}[\log p_\theta(\mathbf{e}_n|\mathbf{c}^{unc})]$. Then, we have:

$$
\begin{aligned}
\mathbb{E}_{p_{data}(\mathbf{e}_n,\mathbf{c}^{unc})}[\log p_\theta(\mathbf{e}_n|\mathbf{c}^{unc})] =& \mathbb{E}_{p_{data}(\mathbf{c}^{unc})}\mathbb{E}_{p_{data}(\mathbf{e}_n|\mathbf{c}^{unc})}[\log p_\theta(\mathbf{e}_n|\mathbf{c}^{unc})] \\
=& \mathbb{E}_{p_{data}(\mathbf{c}^{unc})}\left[\int_{\mathbf{e}_n} p_{data}(\mathbf{e}_n|\mathbf{c}^{unc})\log p_\theta(\mathbf{e}_n|\mathbf{c}^{unc})d\mathbf{e}_n\right] \\
=& \mathbb{E}_{p_{data}(\mathbf{c}^{unc})}\left[-H(p_{data}(\mathbf{e}_n^*|\mathbf{c}^{unc}), p_\theta(\mathbf{e}_n^*|\mathbf{c}^{unc}))\right] \\
=& \mathbb{E}_{p_{data}(\mathbf{c}^{unc})}\left[-H(p_{data}(\mathbf{e}_n^*|\mathbf{c}^{unc}))\right] \\
& - \mathbb{E}_{p_{data}(\mathbf{c}^{unc})}\left[D_{\mathrm{KL}}(p_{data}(\mathbf{e}_n^*|\mathbf{c}^{unc})\|p_\theta(\mathbf{e}_n^*|\mathbf{c}^{unc}))\right] \\
=& -H_{p_{data}}(\mathcal{E}|\mathcal{C}^{unc}) \\
& - \mathbb{E}_{p_{data}(\mathbf{c}^{unc})}\left[D_{\mathrm{KL}}(p_{data}(\mathbf{e}_n^*|\mathbf{c}^{unc})\|p_\theta(\mathbf{e}_n^*|\mathbf{c}^{unc}))\right],
\end{aligned}
\tag{16}
$$

where $\mathcal{E}$ represents the whole generation space and $\mathbf{e}_n^* \in \mathcal{E}$. $\mathcal{C}^{unc}$ is the whole space of uncredible condition $\mathbf{c}^{unc}$. $H(\cdot,\cdot)$ is the cross entropy between two variables or distributions. According to the above derivation, we have:

$$
\begin{aligned}
H_{p_{data}}(\mathcal{E}|\mathcal{C}^{unc}) =& -\mathbb{E}_{p_{data}(\mathbf{e}_n,\mathbf{c}^{unc})}[\log p_\theta(\mathbf{e}_n|\mathbf{c}^{unc})] \\
& - \mathbb{E}_{p_{data}(\mathbf{c}^{unc})}\left[D_{\mathrm{KL}}(p_{data}(\mathbf{e}_n^*|\mathbf{c}^{unc})\|p_\theta(\mathbf{e}_n^*|\mathbf{c}^{unc}))\right] \\
\leq& -\mathbb{E}_{p_{data}(\mathbf{e}_n,\mathbf{c}^{unc})}[\log p_\theta(\mathbf{e}_n|\mathbf{c}^{unc})].
\end{aligned}
\tag{17}
$$

Ideally, when the model is optimally trained, the $D_{\mathrm{KL}}$ term will approach zero, indicating that the conditional generation distribution approaches the real data distribution. Therefore, the mutual information between the whole conditional generation space $\mathcal{E}$ and the whole uncredible condition space $\mathcal{C}^{unc}$ can be calculated as:

$$
\begin{aligned}
I_{p_\theta}(\mathcal{E}, \mathcal{C}^{unc}) =& I_{p_{data}}(\mathcal{E}, \mathcal{C}^{unc}) \\
=& H_{p_{data}}(\mathcal{E}) - H_{p_{data}}(\mathcal{E}|\mathcal{C}^{unc}) \\
\geq& H_{p_{data}}(\mathcal{E}) + \mathbb{E}_{p_{data}(\mathbf{e}_n,\mathbf{c}^{unc})}[\log p_\theta(\mathbf{e}_n|\mathbf{c}^{unc})].
\end{aligned}
\tag{18}
$$

The second equation is derived according to the property of mutual information: $I(X,Y) = H(X) - H(X|Y)$. As the training goes on, the second term becomes larger. At the same time, $H_{p_{data}}(\mathcal{E})$ is a constant based on the real data distribution $p_{data}$. Hence, the lower bound of $I_{p_\theta}(\mathcal{E}, \mathcal{C}^{unc})$ also becomes larger. Based on this, we can conclude that training the diffusion model with uncredible conditions increases the mutual information between the generation space and the uncredible condition space. This indicates that the generation space increasingly contains uncredible features reflected in the uncredible conditions.

**Proof: Uncredible diffusion target can enhance DM's generation of uncredible results**

The optimization loss of existing DM-based recommendation methods can be formulated as:

$$
\mathcal{L} = \mathbb{E}_{t\sim U(0,T)}[\|\mathbf{e}_n^0 - f_\theta(\mathbf{e}_n^t, \mathbf{c}, t)\|_2^2].
\tag{19}
$$

When an uncredible item embedding $\mathbf{e}_j$ ($j \in \mathcal{I}_{unc}$) is used as the diffusion target (i.e., uncredible diffusion target) during training, the diffusion loss encourages the prediction direction of the diffusion network $f_\theta$ to move closer to $\mathbf{e}_j$. Specifically, the MSE distance between the diffusion target and the output of $f_\theta$ will be smaller, indicating higher similarity.

In the inference stage, the generation process of diffusion recommenders can be expressed as:

$$\mathbf{e}_n^{t-1} = w_1 f_\theta(\mathbf{e}_n^t, \mathbf{c}, t) + w_2 \mathbf{e}_n^t + w_3 \boldsymbol{\epsilon}, \quad \boldsymbol{\epsilon} \sim \mathcal{N}(\mathbf{0}, \mathbf{I}), \tag{20}$$

where $w_1 = \frac{\sqrt{\bar{\alpha}_{t-1}}\beta_t}{1-\bar{\alpha}_t}$, $w_2 = \frac{\sqrt{\alpha_t}(1-\bar{\alpha}_{t-1})}{1-\bar{\alpha}_t}$, and $w_3 = \sqrt{\frac{1-\bar{\alpha}_{t-1}}{1-\bar{\alpha}_t}(1-\alpha_t)}$. This generation process is performed step by step, and the final embedding $\mathbf{e}_n^0$ is taken as the generation result, which then serves as the reference for item prediction and recommendations.

Let $\mathbf{e}_n^{t-1}$ denote the generated embedding at step $t-1$ without using uncredible diffusion target $\mathbf{e}_j$ during training. In such case, the parameters of the diffusion network are denoted as $\theta$. Similarly, let $\hat{\mathbf{e}}_n^{t-1}$ denote the generated embedding at step $t-1$ with $\mathbf{e}_j$ as the uncredible diffusion target during training. In this case, the diffusion parameters are denoted as $\hat{\theta}$. We then calculate the difference in similarity between the normalized $\mathbf{e}_j$ and the normalized generated embeddings $\hat{\mathbf{e}}_n^{t-1}$ and $\mathbf{e}_n^{t-1}$ at step $t-1$ as follows:

$$\begin{aligned}
\Delta^{t-1} &= \text{sim}(\mathbf{e}_j, \hat{\mathbf{e}}_n^{t-1}) - \text{sim}(\mathbf{e}_j, \mathbf{e}_n^{t-1}) \\
&= \left[ w_1 \left( f_{\hat{\theta}}(\hat{\mathbf{e}}_n^t, \mathbf{c}, t) - f_\theta(\mathbf{e}_n^t, \mathbf{c}, t) \right) + w_2 \left( \hat{\mathbf{e}}_n^t - \mathbf{e}_n^t \right) + w_3 (\boldsymbol{\epsilon}^t - \boldsymbol{\epsilon}^t) \right] \cdot \mathbf{e}_j^\top \\
&= w_1 \left( f_{\hat{\theta}}(\hat{\mathbf{e}}_n^t, \mathbf{c}, t) - f_\theta(\mathbf{e}_n^t, \mathbf{c}, t) \right) \cdot \mathbf{e}_j^\top + w_2 \left( \hat{\mathbf{e}}_n^t - \mathbf{e}_n^t \right) \cdot \mathbf{e}_j^\top.
\end{aligned} \tag{21}$$

Here, we utilize the dot product to calculate the similarity. To avoid the interference from sampled noise, we use $\boldsymbol{\epsilon}^t$ to denote the sample noise $\boldsymbol{\epsilon}$ in step $t-1$, and use it for both generation processes to control this variable.

When $t = T$, we have:

$$\begin{aligned}
\Delta^{T-1} &= \text{sim}(\mathbf{e}_j, \hat{\mathbf{e}}_n^{T-1}) - \text{sim}(\mathbf{e}_j, \mathbf{e}_n^{T-1}) \\
&= w_1 \left( f_{\hat{\theta}}(\hat{\mathbf{e}}_n^T, \mathbf{c}, T) - f_\theta(\mathbf{e}_n^T, \mathbf{c}, T) \right) \cdot \mathbf{e}_j^\top + w_2 \left( \hat{\mathbf{e}}_n^T - \mathbf{e}_n^T \right) \cdot \mathbf{e}_j^\top \\
&= w_1 \left( f_{\hat{\theta}}(\boldsymbol{\epsilon}^T, \mathbf{c}, T) - f_\theta(\boldsymbol{\epsilon}^T, \mathbf{c}, T) \right) \cdot \mathbf{e}_j^\top + w_2 \left( \boldsymbol{\epsilon}^T - \boldsymbol{\epsilon}^T \right) \cdot \mathbf{e}_j^\top \\
&= w_1 \left( f_{\hat{\theta}}(\boldsymbol{\epsilon}^T, \mathbf{c}, T) \cdot \mathbf{e}_j^\top - f_\theta(\boldsymbol{\epsilon}^T, \mathbf{c}, T) \cdot \mathbf{e}_j^\top \right) \\
&= \underbrace{w_1}_{>0} \underbrace{\left( \text{sim}\left( \mathbf{e}_j, f_{\hat{\theta}}(\boldsymbol{\epsilon}^T, \mathbf{c}, T) \right) - \text{sim}\left( \mathbf{e}_j, f_\theta\left( \boldsymbol{\epsilon}^T, \mathbf{c}, T \right) \right) \right)}_{>0} > 0.
\end{aligned} \tag{22}$$

We control the process of two generations start from the same point $\hat{\mathbf{e}}_n^T = \mathbf{e}_n^T = \boldsymbol{\epsilon}^T$ for fair comparison. As mentioned earlier, the prediction direction of $f_{\hat{\theta}}$ is closer to $\mathbf{e}_j$ than that of $f_\theta$. Hence, the MSE distance between the output of $f_{\hat{\theta}}$ and to $\mathbf{e}_j$ is smaller than that between the output of $f_\theta$ and $\mathbf{e}_j$. When the embeddings are normalized, a smaller MSE distance corresponds to a higher dot product similarity. Consequently, $\text{sim}\left( \mathbf{e}_j, f_{\hat{\theta}}(\boldsymbol{\epsilon}^T, \mathbf{c}, T) \right) - \text{sim}(\mathbf{e}_j, f_\theta\left( \boldsymbol{\epsilon}^T, \mathbf{c}, T \right)) > 0$. At the same time, $w_1 > 0$, therefore $\Delta^{T-1} > 0$. This indicates that, when starting from the same initial point, the generation result at step $T-1$ produced by model $f_{\hat{\theta}}$, which has been trained with an uncredible diffusion target, will be more similar to this uncredible diffusion target.

When $t = T-1$, we have:

$$\begin{aligned}
\Delta^{T-2} &= w_1 \left( f_{\hat{\theta}}(\hat{\mathbf{e}}_n^{T-1}, \mathbf{c}, T-1) - f_\theta(\mathbf{e}_n^{T-1}, \mathbf{c}, T-1) \right) \cdot \mathbf{e}_j^\top + w_2 \left( \hat{\mathbf{e}}_n^{T-1} - \mathbf{e}_n^{T-1} \right) \cdot \mathbf{e}_j^\top \\
&= w_1 \mathrm{C}_{T-1}^+ + w_2 \left( \text{sim}(\mathbf{e}_j, \hat{\mathbf{e}}_n^{T-1}) - \text{sim}(\mathbf{e}_j, \mathbf{e}_n^{T-1}) \right) \\
&= w_1 \mathrm{C}_{T-1}^+ + w_2 \Delta^{T-1}.
\end{aligned} \tag{23}$$

As mentioned before, when a uncredible item embedding $\mathbf{e}_j$ is taken for training, the diffusion loss will encourage the prediction direction of $f_{\hat{\theta}}$ closer to $\mathbf{e}_j$. At the same time, $\hat{\mathbf{e}}_n^{T-1}$ is closer to $\mathbf{e}_j$, as compared to that of $\mathbf{e}_n^{T-1}$. This further enforces $f_{\hat{\theta}}(\hat{\mathbf{e}}_n^{T-1}, \mathbf{c}, T-1)$ more similar to $\mathbf{e}_j$, than that of $f_\theta(\mathbf{e}_n^{T-1}, \mathbf{c}, T-1)$. Hence, the first term is a positive constant, and we denote it by $\mathrm{C}_{T-1}^+$.

Similarly, when $t < T-1$, we have:

$$\begin{aligned}
\Delta^{T-3} &= w_1 \mathrm{C}_{T-2}^+ + w_2 \Delta^{T-2} \\
&= w_1 \mathrm{C}_{T-2}^+ + w_2 (w_1 \mathrm{C}_{T-1}^+ + w_2 \Delta^{T-1}) \\
&= w_1 \mathrm{C}_{T-2}^+ + w_1 w_2 \mathrm{C}_{T-1}^+ + w_2^2 \Delta^{T-1}.
\end{aligned} \tag{24}$$

$$
\begin{aligned}
\Delta^{T-4} \quad &= w_1 \mathrm{C}^+_{T-3} + w_2 \Delta^{T-3} \\
&= w_1 \mathrm{C}^+_{T-3} + w_2(w_1 \mathrm{C}^+_{T-2} + w_1 w_2 \mathrm{C}^+_{T-1} + w_2^2 \Delta^{T-1}) \\
&= w_1 \mathrm{C}^+_{T-3} + w_1 w_2 \mathrm{C}^+_{T-2} + w_1 w_2^2 \mathrm{C}^+_{T-1} + w_2^3 \Delta^{T-1}.
\end{aligned}
\tag{25}
$$

$$
\cdots
$$

$$
\begin{aligned}
\Delta^0 \quad &= w_1 \mathrm{C}^+_1 + w_1 w_2 \mathrm{C}^+_2 + \cdots + w_1 w_2^{T-2} \mathrm{C}^+_{T-1} + w_2^{T-1} \Delta^{T-1} \\
&= \underbrace{\sum_m^{T-1} w_1 w_2^{m-1} \mathrm{C}^+_m}_{>0} + \underbrace{w_2^{T-1} \Delta^{T-1}}_{>0} \\
&= \mathrm{sim}(\mathbf{e}_j, \hat{\mathbf{e}}_n^0) - \mathrm{sim}(\mathbf{e}_j, \hat{\mathbf{e}}_n^0) \\
&> 0.
\end{aligned}
\tag{26}
$$

According to above analysis, the final generated result $\hat{\mathbf{e}}_n^0$ using diffusion network $f_{\hat{\theta}}$ is more similar with uncredible item embedding $\mathbf{e}_j$, as compared to the final generated result $\mathbf{e}_n^0$ using diffusion network $f_\theta$. This indicates that when uncredible items are used as the diffusion targets during training, the model tends to generate outputs that carry more uncredible features, i.e., embeddings that are more similar to uncredible items.

---

**Algorithm 1** Training of `Disco`

---

1: **Input:** Training dataset $\mathcal{S}_{train} = \{(\mathbf{e}_n, \mathbf{e}_{neg}, s^{pre}, s^{unc})\}_{s=1}^{|\mathcal{S}_{train}|}$, available uncredible item set $\mathcal{I}_{unc}$, trainable parameters $\Theta$, total diffusion steps $T$, learning rate $\eta$, variance schedules $\{\alpha_t\}_{t=0}^T$.
2: **Output:** Optimized parameters $\Theta$.
3: $\mathbf{F} = \mathrm{Stack}(\{\mathbf{e}_i^{unc}\}_{i \in \mathcal{I}_{unc}})$ ▷ Construct uncredible feature matrix
4: **repeat**
5: $\quad (\mathbf{e}_n, \mathbf{e}_{neg}, s^{pre}, s^{unc}) \sim \mathcal{S}_{train}$ ▷ Sample training data
6: $\quad \mathbf{c}^{pre} = \mathrm{Transmformer}(s^{pre})$ ▷ Obtain preference-related condition
7: $\quad \mathbf{c}^{unc} = \mathrm{Mean}(s^{unc})$ ▷ Obtain uncredible content-related condition
8: $\quad$ **Update F** by Algorithm 3 ▷ Progressive uncredible feature matrix enhancement
9: $\quad [\mathbf{U}_1; \mathbf{U}_2], \boldsymbol{\Lambda}, \mathbf{V} = \mathrm{SVD}(\mathbf{F}^\top)$ ▷ Construct null space of uncredible feature matrix
10: $\quad \tilde{\mathbf{e}}_n = \mathbf{e}_n \mathbf{U}_2 \mathbf{U}_2^\top$ ▷ Credible subspace projection for diffusion target $\mathbf{e}_n$
11: $\quad \tilde{\mathbf{e}}_n = (\tilde{\mathbf{e}}_n + \mathbf{e}_n)/2$ ▷ Residual connection
12: $\quad t \sim \mathrm{Uniform}(1, T)$ ▷ Sample diffusion step
13: $\quad \tilde{\mathbf{e}}_n^t = \sqrt{\bar{\alpha}_t} \tilde{\mathbf{e}}_n^0 + \sqrt{1 - \bar{\alpha}_t} \boldsymbol{\epsilon}$ ▷ Add noise to the embedding of diffusion target
14: $\quad \mathbf{e}_{neg}^t = \sqrt{\bar{\alpha}_t} \mathbf{e}_{neg}^0 + \sqrt{1 - \bar{\alpha}_t} \boldsymbol{\epsilon}$ ▷ Add noise to the embedding of negative preference item
15: $\quad \Theta \leftarrow \Theta - \eta \nabla_\Theta \mathcal{L}_{\texttt{Disco}}(\tilde{\mathbf{e}}_n, \mathbf{e}_{neg}, \mathbf{c}^{pre}, \mathbf{c}^{unc}, t, \Theta)$ ▷ Update parameters
16: **until** convergence
17: **return** $\Theta$

---

---

**Algorithm 2** Inference of `Disco`

---

1: **Input:** Test dataset $\mathcal{S}_{test} = \{s^{pre}\}_{s=1}^{|\mathcal{S}_{test}|}$, trained diffusion network parameters $\theta \in \Theta$, total reverse steps $T$, DDIM steps $T'$, variance schedules $\{\alpha_t\}_{t=0}^T$.
2: **Output:** A recommendation list for each user/sequence.
3: $s^{pre} \sim \mathcal{S}_{test}$ ▷ Sample test sequence
4: $\mathbf{c}^{pre} = \mathrm{Transformer}(s^{pre})$ ▷ Obtain preference-related condition
5: **for** $t' = T', \cdots, 1$ **do**
6: $\quad t = \lfloor t' \times (T/T') \rfloor$ ▷ Calculate DDIM denoising step
7: $\quad \mathbf{e}_n^T \sim \mathcal{N}(\mathbf{0}, \mathbf{I})$ ▷ Start from Gaussian noise
8: $\quad \mathbf{e}_n^{t-1} = \frac{\sqrt{\bar{\alpha}_{t-1}}(1-\alpha_t)}{1-\alpha_t} f_\theta(\mathbf{e}_n^t, \mathbf{c}^{pre}, t) + \frac{\sqrt{\alpha_t}(1-\bar{\alpha}_{t-1})}{1-\bar{\alpha}_t} \mathbf{e}_n^t + \sqrt{\frac{1-\bar{\alpha}_{t-1}}{1-\bar{\alpha}_t}(1-\alpha_t)} \boldsymbol{\epsilon}$ ▷ Step-by-step generation
9: **end for**
10: $\hat{y}_i = \mathbf{e}_n^0 \cdot \mathbf{e}_i^\top$ ▷ Calculate the matching score between a user/sequence and a candidate item $\mathbf{e}_i$
11: $\mathcal{R} = \{i | \mathrm{TopK}(\hat{y}_i), i \in \mathcal{I}\}$ ▷ Select top K items with highest matching scores
12: **return** $\mathcal{R}$

---

---

**Algorithm 3** Progressive enhancement of uncredible feature matrix

---

1: **Input:** Original uncredible feature matrix $\mathbf{F}$, available uncredible item set $\mathcal{I}_{unc}$, current iteration $j$, maximum selection ratio $\gamma$, maximum iteration $m$ to reach $\gamma$.
2: **Output:** Updated uncredible feature matrix $\mathbf{F}$.
3: $\text{UD}(i) = \frac{1}{|\mathcal{I}_{unc}|} \sum_{i' \in \mathcal{I}_{unc}} \cos(\mathbf{e}_i^{unc}, \mathbf{e}_{i'}^{unc})$  ▷ Calculate uncredible degree of items in $\mathcal{I} \setminus \mathcal{I}_{unc}$
4: $ratio(j) = \min(\gamma, \frac{j}{m}\gamma)$  ▷ Calculate the selection ratio at current iteration
5: select $\lfloor |\mathcal{I} \setminus \mathcal{I}_{unc}| \cdot ratio(j) \rfloor$ items with highest uncredible degree ▷ Select potential uncredible items
6: Add potential uncredible items into $\mathcal{I}_{unc}$  ▷ Extension of uncredible item set
7: $\mathbf{F} = \text{Stack}(\{\mathbf{e}_i^{unc}\}_{i \in \mathcal{I}_{unc}})$  ▷ Enhancement of uncredible feature matrix
8: **return F**

---

## D  DERIVATION OF EQUATION 5

In this section, we provide the derivation of Equation 5. For simplicity, we only need to derive the first term, since the derivation of the second term follows the same procedure. The detailed derivation is as follows:

$$
\begin{aligned}
-\mathbb{E}_q \left[ \log \frac{p_\theta\left(\mathbf{e}_n^{0:T}|\mathbf{c}^{pre}\right)}{q(\mathbf{e}_n^{1:T}|\mathbf{e}_n^0)} \right] &\overset{①}{=} -\mathbb{E}_q \left[ \log \frac{p(\mathbf{e}_n^T|\mathbf{c}^{pre})p_\theta(\mathbf{e}_n^0|\mathbf{e}_n^1,\mathbf{c}^{pre})\prod_{t>1}^T p_\theta\left(\mathbf{e}_n^{t-1}|\mathbf{e}_n^t,\mathbf{c}^{pre}\right)}{q(\mathbf{e}_n^1|\mathbf{e}_n^0)\prod_{t>1}^T q(\mathbf{e}_n^t|\mathbf{e}_n^{t-1},\mathbf{e}_n^0)} \right] \\[4pt]
&\overset{②}{=} -\mathbb{E}_q \left[ \log \frac{p(\mathbf{e}_n^T|\mathbf{c}^{pre})p_\theta(\mathbf{e}_n^0|\mathbf{e}_n^1,\mathbf{c}^{pre})\prod_{t>1}^T p_\theta\left(\mathbf{e}_n^{t-1}|\mathbf{e}_n^t,\mathbf{c}^{pre}\right)}{q(\mathbf{e}_n^1|\mathbf{e}_n^0)\prod_{t>1}^T \frac{q(\mathbf{e}_n^{t-1}|\mathbf{e}_n^t,\mathbf{e}_n^0)q(\mathbf{e}_n^t|\mathbf{e}_n^0)}{q(\mathbf{e}_n^{t-1}|\mathbf{e}_n^0)}} \right] \\[4pt]
&\overset{③}{=} -\mathbb{E}_q \left[ \log \frac{p(\mathbf{e}_n^T|\mathbf{c}^{pre})p_\theta(\mathbf{e}_n^0|\mathbf{e}_n^1,\mathbf{c}^{pre})\prod_{t>1}^T p_\theta\left(\mathbf{e}_n^{t-1}|\mathbf{e}_n^t,\mathbf{c}^{pre}\right)}{q(\mathbf{e}_n^1|\mathbf{e}_n^0)\frac{q(\mathbf{e}_n^2|\mathbf{e}_n^0)}{q(\mathbf{e}_n^1|\mathbf{e}_n^0)}\cdots\frac{q(\mathbf{e}_n^T|\mathbf{e}_n^0)}{q(\mathbf{e}_n^{T-1}|\mathbf{e}_n^0)}\prod_{t>1}^T q(\mathbf{e}_n^{t-1}|\mathbf{e}_n^t,\mathbf{e}_n^0)} \right] \\[4pt]
&\overset{④}{=} -\mathbb{E}_q \left[ \log p_\theta(\mathbf{e}_n^0|\mathbf{e}_n^1,\mathbf{c}^{pre}) + \log \frac{p_\theta(\mathbf{e}_n^T)}{q(\mathbf{e}_n^T|\mathbf{e}_n^0)} + \log \frac{\prod_{t>1}^T p_\theta\left(\mathbf{e}_n^{t-1}|\mathbf{e}_n^t,\mathbf{c}^{pre}\right)}{\prod_{t>1}^T q(\mathbf{e}_n^{t-1}|\mathbf{e}_n^t,\mathbf{e}_n^0)} \right] \\[4pt]
&\overset{⑤}{=} -\mathbb{E}_q \left[ \log p(\mathbf{e}_n^0|\mathbf{e}_n^1,\mathbf{c}^{pre}) \right] - \mathbb{E}_q \left[ \log \frac{p_\theta(\mathbf{e}_n^T)}{q(\mathbf{e}_n^T|\mathbf{e}_n^0)} \right] \\
&\qquad - \sum_{t>1}^T \mathbb{E}_q \left[ \log \frac{p_\theta\left(\mathbf{e}_n^{t-1}|\mathbf{e}_n^t,\mathbf{c}^{pre}\right)}{q(\mathbf{e}_n^{t-1}|\mathbf{e}_n^t,\mathbf{e}_n^0)} \right] \\[4pt]
&\overset{⑥}{=} -\underbrace{\mathbb{E}_q \left[ \log p(\mathbf{e}_n^0|\mathbf{e}_n^1,\mathbf{c}^{pre}) \right]}_{\text{reconstruction term}} + \underbrace{D_{\text{KL}}\left(q(\mathbf{e}_n^T|\mathbf{e}_n^0)\|p_\theta(\mathbf{e}_n^T)\right)}_{\text{prior matching term}} \\
&\qquad + \underbrace{\sum_{t>1}^T \mathbb{E}_q \left[ D_{\text{KL}}\left(q(\mathbf{e}_n^{t-1}|\mathbf{e}_n^t,\mathbf{e}_n^0)\|p_\theta\left(\mathbf{e}_n^{t-1}|\mathbf{e}_n^t,\mathbf{c}^{pre}\right)\right) \right]}_{\text{denoising matching term}}.
\end{aligned}
$$

$$(27)$$

Equation ② is derived through Bayes rule: $q(\mathbf{e}_n^t|\mathbf{e}_n^{t-1},\mathbf{e}_n^0) = \frac{q(\mathbf{e}_n^{t-1}|\mathbf{e}_n^t,\mathbf{e}_n^0)q(\mathbf{e}_n^t|\mathbf{e}_n^0)}{q(\mathbf{e}_n^{t-1}|\mathbf{e}_n^0)}$. Equation ④ is obtained since $p(\mathbf{e}_n^T|\mathbf{c}^{pre}) = p(\mathbf{e}_n^T)$ given $\mathbf{e}_n^T \sim \mathcal{N}(\mathbf{0},\mathbf{I})$, which is independent with condition $\mathbf{c}^{pre}$.

DMs generally optimize the denoising matching term $D_{\text{KL}}\left(q(\mathbf{e}_n^{t-1}|\mathbf{e}_n^t,\mathbf{e}_n^0)\|p_\theta\left(\mathbf{e}_n^{t-1}|\mathbf{e}_n^t,\mathbf{c}^{pre}\right)\right)$ instead of the whole variational bound. Then, this denoising matching term can be derived into the optimization loss $\mathcal{L} = \mathbb{E}_{\mathbf{e}_n^0,\mathbf{c}^{pre},t} \left[ \frac{1}{2\sigma_t^2} \left\| \boldsymbol{\mu}_q(\mathbf{e}_n^t,\mathbf{e}_n^0) - \boldsymbol{\mu}_\theta(\mathbf{e}_n^t,\mathbf{c}^{pre},t) \right\|_2^2 \right]$, by adding the condition $\mathbf{c}^{pre}$ into $\boldsymbol{\mu}_\theta(\mathbf{e}_n^t,t)$ in (Ho et al., 2020). Similar with (Pathak et al., 2023), $\boldsymbol{\mu}_q(\mathbf{e}_n^t,\mathbf{e}_n^0)$ is defined as

(Pathak et al., 2023):

$$\boldsymbol{\mu}_q(\mathbf{e}_n^t, \mathbf{e}_n^0) = \frac{\sqrt{\alpha_t}(1 - \bar{\alpha}_{t-1})\mathbf{e}_n^t + \sqrt{\bar{\alpha}_{t-1}}(1 - \alpha_t)\mathbf{e}_n^0}{1 - \bar{\alpha}_t}. \tag{28}$$

In our model, $\boldsymbol{\mu}_\theta(\mathbf{e}_n^t, \mathbf{c}^{pre}, t)$ is defined as:

$$\boldsymbol{\mu}_\theta(\mathbf{e}_n^t, \mathbf{c}^{pre}, t) = \frac{\sqrt{\alpha_t}(1 - \bar{\alpha}_{t-1})\mathbf{e}_n^t + \sqrt{\bar{\alpha}_{t-1}}(1 - \alpha_t)f_\theta(\mathbf{e}_n^t, \mathbf{c}^{pre}, t)}{1 - \bar{\alpha}_t}, \tag{29}$$

where $f_\theta(\mathbf{e}_n^t, \mathbf{c}^{pre}, t)$ is the predicted $\mathbf{e}_n^0$ using the diffusion network $f_\theta$.

Then, the optimization term can be rewritten as:

$$
\begin{aligned}
\mathcal{L} =& \mathbb{E}_{\mathbf{e}_n^0, \mathbf{c}^{pre}, t}\left[\frac{1}{2\sigma_t^2}\left\|\boldsymbol{\mu}_q(\mathbf{e}_n^t, \mathbf{e}_n^0) - \boldsymbol{\mu}_\theta(\mathbf{e}_n^t, t)\right\|_2^2\right] \\
=& \mathbb{E}_{\mathbf{e}_n^0, \mathbf{c}^{pre}, t}\left[\frac{1}{2\sigma_t^2}\left\|\frac{\sqrt{\alpha_t}(1 - \bar{\alpha}_{t-1})\mathbf{e}_n^t + \sqrt{\bar{\alpha}_{t-1}}(1 - \alpha_t)\mathbf{e}_n^0}{1 - \bar{\alpha}_t}\right.\right. \\
& \left.\left. - \frac{\sqrt{\alpha_t}(1 - \bar{\alpha}_{t-1})\mathbf{e}_n^t + \sqrt{\bar{\alpha}_{t-1}}(1 - \alpha_t)f_\theta(\mathbf{e}_n^t, \mathbf{c}^{pre}, t)}{1 - \bar{\alpha}_t}\right\|_2^2\right] \\
=& \mathbb{E}_{\mathbf{e}_n^0, \mathbf{c}^{pre}, t}\left[\frac{1}{2\sigma_q^2(t)}\left\|\frac{\sqrt{\bar{\alpha}_{t-1}}(1 - \alpha_t)}{1 - \bar{\alpha}_t}\mathbf{e}_n^0 - \frac{\sqrt{\bar{\alpha}_{t-1}}(1 - \alpha_t)}{1 - \bar{\alpha}_t}f_\theta(\mathbf{e}_n^t, \mathbf{c}^{pre}, t)\right\|_2^2\right] \\
=& \mathbb{E}_{\mathbf{e}_n^0, \mathbf{c}^{pre}, t}\left[\frac{1}{2\sigma_q^2(t)}\left(\frac{\sqrt{\bar{\alpha}_{t-1}}(1 - \alpha_t)}{1 - \bar{\alpha}_t}\right)^2\left\|\mathbf{e}_n^0 - f_\theta(\mathbf{e}_n^t, \mathbf{c}^{pre}, t)\right\|_2^2\right].
\end{aligned}
\tag{30}
$$

In practice, the coefficient $\frac{1}{2\sigma_q^2(t)}\left(\frac{\sqrt{\bar{\alpha}_{t-1}}(1-\alpha_t)}{1-\bar{\alpha}_t}\right)^2$ is generally omitted (Ho et al., 2020). Hence, the optimization loss of our preference-related condition guided generation can be rewritten as $\mathcal{L}_{pre} = \mathbb{E}_{\mathbf{e}_n^0, \mathbf{c}^{pre}, t}\left[\left\|\mathbf{e}_n^0 - f_\theta(\mathbf{e}_n^t, \mathbf{c}^{pre}, t)\right\|_2^2\right]$. Similarly, the optimization loss of uncredible content-related condition guided generation is: $\mathcal{L}_{unc} = \mathbb{E}_{\mathbf{e}_n^0, \mathbf{c}^{unc}, t}\left[\left\|\mathbf{e}_n^0 - f_\theta(\mathbf{e}_n^t, \mathbf{c}^{unc}, t)\right\|_2^2\right]$. Our `Disco` model aims to encourage the generation guided by preference-related condition and discourage the generation guided by uncredible content-related condition. To achieve this goal, the optimization objective is formulated as shown in Equation 5:

$$\mathcal{L} = \mathcal{L}_{pre} - \mathcal{L}_{unc} = \mathbb{E}_{\mathbf{e}_n^0, \mathbf{c}^{pre}, t}\left[\left\|\mathbf{e}_n^0 - f_\theta(\mathbf{e}_n^t, \mathbf{c}^{pre}, t)\right\|_2^2\right] - \mathbb{E}_{\mathbf{e}_n^0, \mathbf{c}^{unc}, t}\left[\left\|\mathbf{e}_n^0 - f_\theta(\mathbf{e}_n^t, \mathbf{c}^{unc}, t)\right\|_2^2\right]. \tag{31}$$

## E    THEORETICAL JUSTIFICATION OF CREDIBLE SUBSPACE PROJECTION

Our credible subspace projection is constructed using SVD. Applying SVD to the uncredible feature matrix $\mathbf{F}^\top$ yields the eigenvector matrix $\mathbf{U}$ and the diagonal eigenvalue matrix $\boldsymbol{\Lambda}$. $\mathbf{U}$ and $\boldsymbol{\Lambda}$ can be expressed as $\mathbf{U} = [\mathbf{U}_1; \mathbf{U}_2]$ and $\boldsymbol{\Lambda} = \begin{bmatrix} \boldsymbol{\Lambda}_1 & 0 \\ 0 & \boldsymbol{\Lambda}_2 \end{bmatrix}$. Correspondingly, $\mathbf{V}$ can be expressed as $\mathbf{V} = [\mathbf{V}_1; \mathbf{V}_2]$. All zero or near-zero singular values are contained in $\boldsymbol{\Lambda}_2$, and the corresponding eigenvectors are given by $\mathbf{U}_2$ and $\mathbf{V}_2$.

According to the principles of SVD, the following equation holds:

$$\mathbf{U}_2^\top \mathbf{F}^\top = \mathbf{U}_2^\top \mathbf{U}_1 \boldsymbol{\Lambda}_1 \mathbf{V}_1^\top. \tag{32}$$

Since the matrix $\mathbf{U}$ obtained from the SVD is an orthogonal matrix, we have:

$$\mathbf{U}_2^\top \mathbf{F}^\top = \underbrace{\mathbf{U}_2^\top \mathbf{U}_1}_{=\mathbf{0}} \boldsymbol{\Lambda}_1 \mathbf{V}_1^\top = \mathbf{0}. \tag{33}$$

Our credible diffusion target is derived through $\tilde{\mathbf{e}}_n = \mathbf{e}_n \mathbf{U}_2 \mathbf{U}_2^\top$. Hence, we have:

$$\tilde{\mathbf{e}}_n \mathbf{F}^\top = \mathbf{e}_n \mathbf{U}_2 \underbrace{\mathbf{U}_2^\top \mathbf{F}^\top}_{=\mathbf{0}} = \mathbf{0}. \tag{34}$$

This indicates that the derived credible diffusion target $\tilde{\mathbf{e}}_n$ is orthogonal to the uncredible feature matrix $\mathbf{F}$. In summary, our proposed credible subspace projection effectively removes uncredible content from the diffusion targets by ensuring that the projected targets lie orthogonally to the uncredible content.

## F    CASE STUDY

In this section, we conduct a case study using the GossipCop dataset to evaluate the effectiveness of `Disco`. The GossipCop dataset contains users' interaction sequences with news articles, including both true news (i.e., credible items) and fake news (i.e., uncredible items). Specifically, we present the historical interaction sequences and recommendation lists for five users. The credible content items are marked in green, while uncredible items are marked in red. In addition, to illustrate the semantic relevance between content items, we utilize the same background color to highlight content with similar or related topics. From Table 11, we have the following observations:

- `Disco` demonstrates strong capability in delivering credible recommendations. Specifically, although all these users have interacted with uncredible items in their historical interaction sequences, the recommendation lists generated by `Disco` contain no uncredible content.
- `Disco` is capable of mitigating uncredible content while still preserving high recommendation accuracy. This is achieved by removing uncredible features while retaining users' genuine preference-related information. For example, taking User4 as an example, this user had historically interacted with some news (including fake news) about the death of celebrities (highlighted in yellow). `Disco` can effectively capture this user's genuine preference and recommend some content also in such topics. It is worth noting that User4 had interacted with fake news about the death of "Tom Petty", and `Disco` recommends this user with a credible news article about the same event. This plays an important role in countering misinformation, as it helps users correct false impressions formed through prior exposure to uncredible content.

## G    USAGE CLAIM OF LARGE LANGUAGE MODELS

We only utilize ChatGPT for polishing the academic writing, with the prompt "Proofread the grammar and polish the writing of the given sentences".

Table 11: Five cases showcasing the historical interaction sequences and the recommendation lists of five users sampled from GossipCop dataset. Credible refers to credible content (i.e., true news) and Uncredible refers to uncredible content (i.e., fake news). In a user sample, the texts marked by the same-color background refer to similar topics. "**ground truth**" means the corresponding recommended content items have been actually read by the user in the test set.

| | | | | | | |
|---|---|---|---|---|---|---|
| User1 | Historical sequence | Credible: Justin Timberlake, Chris Stapleton release 'Say Something' song, video. | Uncredible: Nicole Kidman, Keith Urban: Secrets to a Successful Relationship. | Uncredible:Kendall Jenner Shades Scott Disick Over Photo With Sofia Richie and His Kids. | Uncredible: Grammy winners 2018: the complete list. | |
| | Recommendations | Credible 2018 Latin GRAMMY Awards Complete Winners List. | Credible: Weinstein Company Files for Bankruptcy and Revokes Nondisclosure Agreements. | Credible: Oscars: The Complete Winners List. | Credible: Pop superstar Lady Gaga has officially landed her first Las Vegas residency. | Credible (**ground truth**): TV News Roundup: Netflix Reveals Fuller House Season 4 Premiere Date |
| User2 | Historical sequence | Credible: 13 Nights Of Halloween 2017 Schedule: Full List of Movies. | Uncredible: Taylor Swift will reportedly keep her new album off streaming services like Spotify and Apple Music for a week. | Uncredible: Former NBC interviewer lashes out at Trump in an NYT op-ed for reportedly casting doubt on the authenticity of the infamous tape. | Credible: 'Big Little Lies' Season 2 News, Premiere Date & Cast. | |
| | Recommendations | Credible (**ground truth**): Justin Timberlake Announces New Album Man of the Woods. | Credible: Seven-time and defending champion says she isn't quite ready to return after giving birth to daughter in September. | Credible: Pop superstar Lady Gaga has officially landed her first Las Vegas residency. | Credible: Jamie Lynn Spears' second child on the way will join big sister Maddie Briann. | Credible: "Good morning baby of mine, John Stamos' fiance Caitlin McHugh wrote as she debuted her baby bump... |
| User3 | Historical sequence | Credible: Hugh Grant and Anna Eberstein's baby on the way joins their daughter. | Uncredible: The cancellation of the third Sex and the City film came with headline-making fallout something Sarah Jessica Parker struggled with | Uncredible: Selena Gomez has completed her treatment for depression and anxiety and is reported feeling | Credible: Congratulations are in order for Rachel McAdams the 39-year-old actress is reportedly going to be a first-time mom! Though she has not personally confirmed the baby news | |
| | Recommendations | Credible: All Chicago West Baby Photos Timeline. | Credible: Demi Lovato Says She Contemplated Suicide at Age 7. | Credible: 'Black Panther' is the most tweeted about movie ever. | Credible (**ground truth**): His wife Faith Hill said the country star had been suffering from dehydration. | Credible: Tisha Campbell-Martin Files For Divorce From Husband of 21 Years |
| User4 | Historical sequence | Uncredible: Caitlyn Jenner told Diane Sawyer that she had undergone the final surgery in her gender reassignment procedures on Friday night's 20/20 special. | Credible: Indiana police found the actress unresponsive after responding to a 911 call Saturday. | Credible: Roger Ailes, Former Fox News CEO, Dies At 77. | Uncredible: Tom Petty Dead: Celebrities React on Social Media Variety. | |
| | Recommendations | Credible: An emotional Celine Dion returned to the stage in Las Vegas on Tuesday night. | Credible (**ground truth**): Rocker Tom Petty died Monday after being rushed to a Los Angeles hospital. | Credible: Hugh Hefner's death certificate from the Los Angeles County Department of Public Health. | Credible: The final season of Netflix's "House of Cards" keeps the secret of how Frank Underwood died until the very end. | Credible: Pauley Perrette announces she's leaving "NCIS" after 15 seasons. |
| User5 | Historical sequence | Credible: Benjamin Glaze had never kissed a girl before Katy Perry tricked him during the ABC reboot of American Idol. | Uncredible: During her chat with Ryan Murphy Friday (March 16) for the opening night of PaleyFest in Los Angeles. | Credible: A longtime aerialist for the famed Cirque Du Soleil plummeted to his death in front of a horrified crowd in Florida on Saturday night while trying out a new act... | Uncredible: Justin Bieber's struggling with his split from Selena Gomez as she's all smiles on her Australian vacation. Here's how the Biebs is coping with his... | |
| | Recommendations | Credible (**ground truth**): Justin Bieber Wants to Be With Selena Gomez But Is Hanging With Baskin Champion. | Credible: The singer covered Ariana Grande's 'Just a Little Bit of Your Heart' in the arena where her concert was attacked... | Credible: Trevorrow helmed the rebooted franchise's first installment. | Credible: Voting closes at 5pm PT today (June 29) for this year's News' TV Scoop Awards... | Credible: Blake Shelton Gets His Palms Read With Jimmy Fallon, Jokes About Having Too Much Sex. |

