# OpenReview forum: "Steering Diffusion Models Towards Credible Content Recommendation"
_ICLR.cc/2026/Conference — ICLR 2026 Poster_

### Official Review · Reviewer_fb2b · 2025-10-31

**Soundness:** 2
**Presentation:** 3
**Contribution:** 3
**Rating:** 4
**Confidence:** 3

**Summary:**

This paper proposes Disco, a diffusion model-based recommender system designed to mitigate uncredible content recommendations such as fake news and misinformation. The authors identify two factors causing existing diffusion models to generate uncredible recommendations: uncredible conditions from users' historical interactions with uncredible items, and uncredible diffusion targets when the target item itself is uncredible. Disco addresses these through three main contributions: a disentangled diffusion model that separates preference signals from uncredible content signals by jointly encouraging generation conditioned on preference-related embeddings while discouraging generation conditioned on uncredible content embeddings; a credible subspace projection module using SVD-based null-space projection to remove uncredible features from diffusion targets; and a progressive enhancement strategy to handle limited labeled data by iteratively detecting and incorporating potential uncredible items. Experiments on PolitiFact, GossipCop, and MHMisinfo datasets with only 20% labeled uncredible items demonstrate improvements in both recommendation accuracy and credibility metrics.

**Strengths:**

1. **Addresses a Critical Real-World Problem**. The paper tackles the important societal concern of recommender systems amplifying uncredible content, with well-motivated examples including COVID-19 misinformation spread (lines 59-62). The problem formulation is realistic, acknowledging that only partial credibility labels are available in practice (lines 135-138), which distinguishes this work from prior methods like Rec4Mit, HDInt, and PRISM that assume complete label availability. This practical constraint makes the research more applicable to real-world deployment scenarios.

2. **Strong Theoretical Foundations for the design**. The disentangled diffusion model uses the diffusion objective itself as a disentangler without auxiliary networks (lines 211-214), reducing computational overhead. The reformulation in Equation 4 that minimizes variational bound conditioned on preference while maximizing it conditioned on uncredible content is theoretically well-motivated. The credible subspace projection using SVD null-space decomposition (Equations 7-8) is grounded in prior null-space projection work, and Appendix C provides both empirical evidence (Table 6) and theoretical proofs demonstrating how uncredible conditions and targets enhance uncredible generation, strengthening the paper's technical contributions.

**Weaknesses:**

1. **Limited Experimental Scope and Insufficient Baseline Analysis**. The evaluation is restricted to three datasets, all in news/video domains (lines 337-343), limiting generalizability to other content types like e-commerce or music. More critically, the paper fixes the labeled data ratio at 20% (line 360) without ablation studies on different ratios (e.g., 10%, 15%, 30%, 50%), making it unclear how sensitive Disco is to label availability.

2. **Unclear Evaluation Protocol**. While the paper states "complete labels are provided during testing" (lines 137-138), it doesn't clarify whether uncredible items are included in the candidate pool during evaluation or how this impacts metrics like CR@K. The data augmentation strategy in Appendix B.1 (lines 772-776) transforms each user sequence into multiple sub-sequences, but doesn't specify how train/test splits prevent leakage when subsequences from the same user appear in both sets.

3. **Inadequate Justification and Analysis of results**. Table 2's ablation reveals that replacing cosine loss with MSE ("w/o CE") causes severe performance degradation (HR@5 drops from 0.2664 to 0.1034 on PolitiFact, line 439), attributed to training instability (lines 227-229), yet the paper provides no learning curves, gradient analysis, or convergence studies to characterize this instability. The weight parameter w in Equation 11 varies across datasets (w=0.5 for PolitiFact, w=1.5 for GossipCop, w=1 for MHMisinfo in Figure 4), suggesting dataset-dependent tuning requirements that are not adequately discussed.

**Questions:**

1. What characteristics of each dataset (e.g., proportion of uncredible items, interaction sparsity, content diversity) determine the optimal w?
2. Can you provide guidelines or a principled approach for setting w on new datasets without exhaustive grid search?
3. Given the data augmentation strategy that creates multiple sub-sequences from each user's interaction history (lines 772-776), how do you ensure no data leakage occurs when sub-sequences from the same user might span the train/test boundary?

---

> ### Author Response · Authors · 2025-11-22
> **Response to Reviewer fb2b -- Part 1**
>
> Thank you very much for highlighting our strengths and for providing such valuable feedback. We sincerely appreciate your thoughtful insights and suggestions. These points are both essential and constructive, and they have prompted us to carefully address each issue, which has significantly improved the quality of our paper. Below, we present our detailed responses. We would be delighted to further engage in discussion if you have any additional questions.
>
> > **Weakness 1: Limited experimental scope and insufficient baseline analysis** — “The evaluation is restricted to three datasets, all in news/video domains (lines 337-343), limiting generalizability to other content types like e-commerce or music. More critically, the paper fixes the labeled data ratio at 20\% (line 360) without ablation studies on different ratios (e.g., 10\%, 15\%, 30\%, 50\%), making it unclear how sensitive Disco is to label availability.”
>
> ### **(1). Concern about the datasets.**
>
> ### **Our used three datasets are the only publicly available datasets that satisfy the requirements of our task.**
>
> Thank you for your valuable insight. We acknowledge your concern about more datasets. However, **these three datasets are the only publicly available dataset which can be used in our task.** The detailed reasons and explanations are as follows:
>
> Our work focuses on mitigating the recommendations of uncredible content to users, with the goal of enhancing user trust and addressing broader societal concerns. Therefore, **credibility labels are required for evaluating the effectiveness of both our method and baseline models** in the context of credible content recommendation.
>
> However, **widely used datasets such as e-commerce dataset Amazon (e.g., Beauty, Sports) and other music datasets do not contain credibility labels for items and thus cannot be used for our task.** Since credible content recommendation is still in its early stages, the three datasets we employ—PolitiFact, GossipCop, and MHMisinfo—are, to the best of our knowledge, the only publicly available datasets that include both user–item interaction data and items’ credibility labels.
>
> **We have made extensive efforts to search across GitHub, HuggingFace, and other public sources, but were unable to identify any additional datasets that satisfy these requirements.**
>
> We have added a clarification regarding this constraint in our updated version (refer to Lines 385-387).
>
> Looking ahead, we hope that as this research area evolves, more suitable datasets will be released to support advancements in credible content recommendation.

---

> ### Author Response · Authors · 2025-11-22
> **Continue on Part 1**
>
> ### **(2). Experiments under various credibility label ratios.**
>
> Thank you for your valuable suggestion. We acknowledge your concern on the performance of Disco under different label ratio (not just 20%). In response, we conduct additional experiments under different credibility label ratio of uncredible content (i.e., 5%, 10%, 20%, 30%, 50%). The experimental results are reported in Table 1. From the results reported in the Table, we can have the following findings:
>
> ### **Key Finding: Even when only 5% of credibility labels are available, our method still achieves superior performance.**
>
> - **Finding 1:** **As the credibility label ratio increases, the recommendation credibility (CR) improves steadily.** This is because a larger number of credibility labels enables the construction of a more comprehensive and accurate credible subspace, allowing uncredible content to be mitigated more effectively.
> - **Finding 2: Even when only 5% of credibility labels are available, Disco still achieves superior performance** compared to competitive methods in terms of both recommendation credibility and recommendation.
> - **Finding 3: The recommendation accuracy fluctuates only slightly within a narrow range across different label ratios.** This stability is attributed to our proposed disentangled diffusion model, which effectively mitigates uncredible content while preserving users’ preference-related information, thereby maintaining high recommendation accuracy.
>
> **Table 1: Performance comparison under different label ratios. Politi, Gossip, MHMis are the abbreviations for PolitiFact, GossipCop and MHMisinfo.**
>
> | Methods (Label ratio) | Politi |  |  &nbsp; &nbsp; &nbsp; &nbsp; &nbsp; &nbsp; &nbsp; &nbsp; &nbsp;| Gossip |  |  &nbsp; &nbsp; &nbsp; &nbsp; &nbsp; &nbsp; &nbsp; &nbsp; &nbsp;| MHMis |  |  |
> | --- | --- | --- | --- | --- | --- | --- | --- | --- | --- |
> |  | HR@5 | CR@5 | HC@5 | HR@5 | CR@5 | HC@5 | HR@5 | CR@5 | HC@5 |
> | GRU4Rec | 0.2142 | 0.9266 | 0.2929 | 0.2226 | 0.8864 | 0.2957 | 0.1151 | 0.8380 | 0.1803 |
> | SASRec | 0.2158 | 0.9059 | 0.2923 | 0.3078 | 0.8743 | 0.3612 | 0.1485 | 0.8339 | 0.2190 |
> | Bert4Rec | 0.2191 | 0.9127 | 0.2960 | 0.2372 | 0.8764 | 0.3078 | 0.1391 | 0.8162 | 0.2074 |
> | CL4SRec | 0.2247 | 0.9132 | 0.3012 | 0.2898 | 0.8938 | 0.3516 | 0.1734 | 0.8577 | 0.2469 |
> | Rec4Mit (20%) | 0.2185 | 0.8959 | 0.2876 | 0.2775 | 0.8979 | 0.3427 | 0.1460 | 0.8424 | 0.2166 |
> | HDInt (20%) | 0.2153 | 0.8946 | 0.2906 | 0.3407 | 0.8986 | 0.3875 | 0.1471 | 0.8306 | 0.2168 |
> | PRISM (20%) | 0.1927 | 0.9325 | 0.2727 | 0.2948 | 0.8806 | 0.3531 | 0.1700 | 0.8398 | 0.2418 |
> | Disco (5%) | 0.2617 | 0.9422 | 0.3365 | 0.5179 | 0.9261 | 0.4889 | 0.2127 | 0.9149 | 0.2904 |
> | Disco (10%) | 0.2541 | 0.9476 | 0.3308 | 0.5290 | 0.9266 | 0.4940 | 0.2112 | 0.9217 | 0.2897 |
> | Disco (20%) | 0.2678 | 0.9823 | 0.3466 | 0.5236 | 0.9272 | 0.4918 | 0.2215 | 0.9305 | 0.3001 |
> | Disco (30%) | 0.2704 | 0.9829 | 0.3489 | 0.5196 | 0.9278 | 0.4902 | 0.2207 | 0.9303 | 0.2994 |
> | Disco (50%) | 0.2678 | 0.9842 | 0.3468 | 0.5151 | 0.9284 | 0.4883 | 0.2134 | 0.9331 | 0.2929 |

---

> ### Author Response · Authors · 2025-11-22
> **Response to Reviewer fb2b -- Part 2**
>
> > **Weakness 2: Unclear Evaluation Protocol** — “While the paper states "complete labels are provided during testing" (lines 137-138), it doesn't clarify whether uncredible items are included in the candidate pool during evaluation or how this impacts metrics like CR@K. The data augmentation strategy in Appendix B.1 (lines 772-776) transforms each user sequence into multiple sub-sequences, but doesn't specify how train/test splits prevent leakage when subsequences from the same user appear in both sets.”
>
> Thank you for raising this concern. Our responses are as follows:
>
> ### **(1). Concern about whether uncredible items are included in the candidate pool.**
>
> Following most existing work, the **candidate pool in our model includes all items, encompassing both credible and uncredible items. Complete labels are used solely to evaluate whether our model effectively mitigates the recommendation of uncredible items, rather than to select the candidate pool.** This approach ensures a fair comparison and better reflects real-world scenarios, where the credibility labels of most content items are not available in advance.
>
> ### **(2). There is no data leakage in our datasets.**
>
> The original datasets have already been divided into training and test sets, with completely distinct user sequences and no user overlap between training and test sets. **Data augmentation is performed separately on the training and test sets**, ensuring that the two sets remain independent. Consequently, this procedure **does not introduce data leakage**, as subsequences from the same user never appear in both sets.

---

> ### Author Response · Authors · 2025-11-22
> **Response to Reviewer fb2b -- Part 3**
>
> > **Weakness 3 & Question 3: Inadequate Justification and Analysis of results** — “Table 2's ablation reveals that replacing cosine loss with MSE ("w/o CE") causes severe performance degradation (HR@5 drops from 0.2664 to 0.1034 on PolitiFact, line 439), attributed to training instability (lines 227-229), yet the paper provides no learning curves, gradient analysis, or convergence studies to characterize this instability.”}
>
> Thank you so much for your valuable suggestion. We have **added an analysis in Appendix B.10** (refer to Lines 1242-1249 in our updated version) to analyze the convergence of Disco w/o CE. We have **added two figures (refer to Figures 7 in Page 23 in our updated version), showing the convergence curves of loss value and metric HC@5.**
>
> ### **w/o CE leads to extremely unstable training and performance** ###
>
> From the figure, we observe that the variant **w/o CE** (i.e., not replacing the MSE error with cosine error) leads to **extremely unstable training and performance**. Specifically, the **loss rapidly collapses to an extremely small value (around $-1.2\times 10^{18}$)**, and the performance **(HR@5) exhibits severe fluctuations**. These results verify the necessity of replacing the MSE error with the cosine error to ensure stable optimization.

---

> ### Author Response · Authors · 2025-11-22
> **Response to Reviewer fb2b -- Part 4**
>
> > **Weakness 3 & Question 1 & Question 2: Setting guidelines of w** — “The weight parameter w in Equation 11 varies across datasets (w=0.5 for PolitiFact, w=1.5 for GossipCop, w=1 for MHMisinfo in Figure 4), suggesting dataset-dependent tuning requirements that are not adequately discussed. What characteristics of each dataset (e.g., proportion of uncredible items, interaction sparsity, content diversity) determine the optimal w? Can you provide guidelines or a principled approach for setting w on new datasets without exhaustive grid search?”
>
>
>
> Thank you for your insightful feedback. Below are our detailed responses.
>
> ### **(1). Relationship between dataset statistics and the hyperparameter $w$.**
>
> After analyzing the relationship between dataset statistics and the hyperparameter $w$, we found that the **optimal value of $w$ is roughly proportional to the number of items** in a dataset.
>
> As shown in Table 2, datasets with more items require a relatively larger $w$ to achieve the best performance. We have included a detailed discussion of this observation in the updated version.
>
> **Table 2: Dataset statistics (number of items) and optimal parameter $w$ values.**
>
> | **Datasets** | **PolitiFact** | **MHMisinfo** | **GossipCop** |
> | --- | --- | --- | --- |
> | **Number of items** | 616 | 3,160 | 9,529 |
> | **Best $w$ value** | 0.5 | 1 | 1.5 |
>
> ### **(2). Deep analysis of this phenomenon.**
>
> The hyperparameter $w$ controls the contribution of the preference-contrast term, which involves sampling negative items from the set of un-interacted items. **When the number of items is larger, the sampled negative items represent a smaller portion of users’ negative preferences. Therefore, a relatively larger $w$ is needed to adequately learn users’ negative preferences**, thereby improving recommendation accuracy.
>
> ### **(3). Guidelines for setting $w$ in new datasets without exhaustive grid search.**
>
> When applying our model to new datasets, **the value of $w$ can be initially be set according to the number of items**. For datasets with a relatively larger number of items (e.g., around or above 10k), $w$ can start with a higher value, such as 1.5-5, and then be finetuned around this range to identify the optimal setting. Conversely, for datasets with fewer items, $w$ can start within a lower range, such as 0-1, and then be find-tuned accordingly.

---

> ### Author Response · Authors · 2025-11-26
> **A gentle reminder for discussion**
>
> Dear Reviewer fb2b,
>
> Thank you very much for your constructive comments and valuable suggestions.
>
> We have carefully addressed your concerns by (1) clarifying the **selection of datasets**; (2) conducting **additional experiments on credibility label ratio**; (3) clarifying **whether uncredible items are included in the candidate pool and the potential of data leakage**; (4) providing **more analysis of an ablation variant: w/o CE**; and (5) providing **more analysis of hyperparameter $w$**. Corresponding revisions have been incorporated into our updated version.
>
> We are kindly wondering if our responses have addressed your concerns. Your feedback is really important to us, and we are looking forward to your response and are happy to answer any further questions.
>
>
>
> Best regards,
>
> The Authors of Submission 11137

---

### Official Review · Reviewer_Vtvf · 2025-10-31

**Soundness:** 3
**Presentation:** 3
**Contribution:** 3
**Rating:** 4
**Confidence:** 4

**Summary:**

This work primarily addresses the critical problem of untrustworthy content generation in recommendation systems. To mitigate this issue, the authors propose Disco, a framework designed to guide diffusion models toward generating credible content recommendations. The core idea is to reformulate the diffusion objective by incorporating preference-related signals that encourage credible generation, while simultaneously suppressing signals associated with untrustworthy or low-credibility content. Extensive experiments conducted on three benchmark datasets demonstrate the effectiveness of Disco.

**Strengths:**

1. The critical problem of untrustworthy content generation in recommendation systems is highly relevant to both the broader societal context and the ICLR research community, given its implications for information integrity, and user trust.
2. The authors focus on addressing the limitations of diffusion models in content recommendation and propose an effective framework for credible content generation, which presents an interesting to the field.
3. The authors provide sufficient experimental validation of the proposed Disco framework, including comparative evaluations, ablation studies, and hyperparameter analyses.
4. The authors provide sufficient mathematical proofs to support the theoretical soundness of the proposed Disco framework.

**Weaknesses:**

1.  While this work aims to advance credible content recommendation, the experimental results primarily focus on traditional accuracy-based metrics, without providing dedicated evaluations or metrics to assess credibility. It remains unclear how the Disco framework identifies uncredible content or ensures the generation of credible recommendations. The authors should further elaborate on (1) how credibility is defined and operationalized within their framework, (2) whether the model explicitly detects or filters uncredible content, and (3) which metrics or benchmarks are used to quantify credibility in the evaluation.
2. Fake news is often defined based on human judgment, which inherently involves subjective and context-dependent factors. However, the Disco framework relies primarily on ID-based embeddings to identify uncredible content, which may lack the semantic richness and contextual understanding necessary to capture the nuanced nature of credibility. Consequently, the motivation for improving content credibility by adjusting the diffusion process solely based on ID-level signals is potentially limited and requires further theoretical justification and empirical validation.
3. The proposed improvement to the diffusion model appears to be incremental, primarily involving the addition of a disentanglement strategy. To better position the novelty of their work, the authors should provide a more detailed comparison with existing diffusion-based recommendation methods such as DreamRec and DiffuRec. Specifically, it is important to clarify how Disco differs in terms of model architecture, and objective formulation.
4. The motivation behind the use of null-space projection for filtering uncredible information is insufficiently explained and appears debatable. While the method is intended to suppress untrustworthy signals, the theoretical justification for why null-space projection effectively filters such content is not clearly articulated. Moreover, the authors do not provide any dedicated experiments to validate the effectiveness of this mechanism in isolating or removing uncredible information.
5. The three datasets used in the experiments are relatively small and may not be sufficient to robustly validate the effectiveness and generalizability of the proposed model. To strengthen the empirical evaluation, the authors are encouraged to consider larger and more diverse datasets that better reflect real-world recommendation scenarios.

**Questions:**

1.  Authors should explain how the Disco framework identifies uncredible content or ensures the generation of credible recommendations. The authors should further elaborate on (1) how credibility is defined and operationalized within their framework, (2) whether the model explicitly detects or filters uncredible content, and (3) which metrics or benchmarks are used to quantify credibility in the evaluation.
2. The authors should provide empirical evidence to support the claim that ID-based embeddings are capable of identifying uncredible content. Without such validation, it remains unclear whether these embeddings capture meaningful credibility-related signals. Additionally, a visualization demonstrating the consistency between the learned embeddings and the ground-truth credibility labels would help substantiate this claim.
3. The proposed improvement to the diffusion model appears to be incremental, primarily involving the addition of a disentanglement strategy. The authors should provide a more detailed comparison with existing diffusion-based recommendation methods such as DreamRec and DiffuRec.
4. The motivation behind the use of null-space projection for filtering uncredible information is insufficiently explained and appears debatable.
5. The authors should consider larger and more diverse datasets to validate the model's effectiveness.

---

> ### Author Response · Authors · 2025-11-22
> **Response to Reviewer Vtvf -- Part 1**
>
> Thank you for your insightful reviews and for recognition of the strengths of our work. We truly appreciate your thoughtful feedback and all the suggestions you raised. Below are our detailed responses. We would be delighted to engage further with you if you have additional questions and concerns.
>
> > **Weakness 1 & Question 1: Clarification on how credibility is defined, operationalized and evaluated within our model** — “While this work aims to advance credible content recommendation, the experimental results primarily focus on traditional accuracy-based metrics, without providing dedicated evaluations or metrics to assess credibility. It remains unclear how the Disco framework identifies uncredible content or ensures the generation of credible recommendations. The authors should further elaborate on (1) how credibility is defined and operationalized within their framework, (2) whether the model explicitly detects or filters uncredible content, and (3) which metrics or benchmarks are used to quantify credibility in the evaluation.”
>
>
>
> Thank you for raising this concern. However, we believe **you may have missed an important part of our evaluation metrics. In fact, we do employ a credibility-oriented metric, CR@K, to assess recommendation credibility.** Our detailed responses are as follows:
>
> ### **(1). Evaluation metric for quantify recommendation credibility.**
>
> We have **already adopted a credibility-oriented evaluation metric, CR@K**, which measures the proportion of credible items among the top-K recommended items.  This description of **this metric is provided in Lines 401-402 and the detailed formulation is provided in Lines 896–908**. This metric was first proposed in [1] (as true news rate in news recommendation) and has been widely used in subsequent studies [2][3].
>
> In addition, we design an accuracy–credibility balanced metric, HC@K, to jointly evaluate recommendation accuracy and credibility (Lines 402-403 and 909–917).
>
> ### **(2). How credibility is defined?**
>
> **Content credibility indicates whether an item contains uncredible information such as false, misleading, or inaccurate content.** Items containing such information are regarded uncredible (e.g., fake news, misinformation), whereas all others are regarded credible. To improve clarity, we have **added an definition of content credibility in the revised version (refer to Line 143-146).**
>
> ### **(3). How credibility is operationalized?**
>
> In our datasets, each item is annotated with a credibility label indicating whether it is uncredible. To simulate a more realistic scenario, where only a small amount of items are verified (Lines 82–84), we assume that only 20% of uncredible items have available labels during training (Lines 367–368).
>
> In contrast, during testing, all credibility labels are available to ensure accurate evaluation of recommendation credibility.
>
> ### **(4). Whether the model explicitly detects or filters uncredible content?**
>
> **Yes. Our model incorporates explicitly designed mechanisms to detect and filter uncredible content.** Specifically, the **progressive enhancement of credible projection module dynamically identifies potential uncredible items** by leveraging the learned uncredible content-related features derived from limited labeled uncredible items (refer to **Lines 103-105** and **Section 3.3** in the updated version). Together with the disentangled diffusion model and credible subspace projection, our framework **effectively mitigates the recommendation of both known and previously unseen uncredible items**.
>
> [1] Shoujin Wang, Xiaofei Xu, Xiuzhen Zhang, Yan Wang, and Wenzhuo Song. Veracity-aware and event-driven personalized news recommendation for fake news mitigation. In WWW, pp. 3673–3684, 2022.
>
> [2] Shoujin Wang, Wentao Wang, Xiuzhen Zhang, Yan Wang, Huan Liu, and Fang Chen. A hierarchical and disentangling interest learning framework for unbiased and true news recommendation. In SIGKDD, pp. 3200–3211, 2024a.
>
> [3] Zihan Ma, Minnan Luo, Yiran Hao, Zhi Zeng, Xiangzheng Kong, and Jiahao Wang. Bridging interests and truth: Towards mitigating fake news with personalized and truthful recommendations. In SIGIR, pp. 490–503, 2025.

---

> ### Author Response · Authors · 2025-11-22
> **Response to Reviewer Vtvf -- Part 2**
>
> > **Weakness 2 & Question 2: Concern on ID-based embedding** — “Fake news is often defined based on human judgment, which inherently involves subjective and context-dependent factors. However, the Disco framework relies primarily on ID-based embeddings to identify uncredible content, which may lack the semantic richness and contextual understanding necessary to capture the nuanced nature of credibility. Consequently, the motivation for improving content credibility by adjusting the diffusion process solely based on ID-level signals is potentially limited and requires further theoretical justification and empirical validation.”
>
>
>
> Thank you for raising this concern. **You may have overlooked the section describing how we obtain item content embeddings**.
>
> ### **Our model is based on Language embeddings, instead of ID embeddings.**
>
> Unlike traditional e-commerce recommender systems that rely on ID-based item embeddings, **our model does not rely on ID-based embeddings**. We have mentioned that item embeddings are **derived using modality-specific encoders** (refer to **Lines 128-130**). Since the content items in our work are text-based (e.g., news articles, misinformation), **we employ LLaMA2-7B [4] to extract language embeddings** for all content items (see **Lines 927–929**).
>
> Besides, for a fair comparison, we used the content embeddings extracted by LLaMA2-7B as the initial item embeddings for all baselines, after projecting them into the required dimensions with an MLP. This ensures that the improvement of our model is convincing.
>
> [4] Hugo Touvron, Louis Martin, Kevin Stone, Peter Albert, Amjad Almahairi, Yasmine Babaei, Nikolay Bashlykov, Soumya Batra, Prajjwal Bhargava, Shruti Bhosale, et al. Llama 2: Open foundation and fine-tuned chat models. arXiv preprint arXiv:2307.09288, 2023.

---

> ### Author Response · Authors · 2025-11-22
> **Response to Reviewer Vtvf -- Part 3**
>
> > **Weakness 3 & Question 3: Clarification of Novelty** — “The proposed improvement to the diffusion model appears to be incremental, primarily involving the addition of a disentanglement strategy. To better position the novelty of their work, the authors should provide a more detailed comparison with existing diffusion-based recommendation methods such as DreamRec and DiffuRec. Specifically, it is important to clarify how Disco differs in terms of model architecture, and objective formulation.”
>
>
>
> Thank you so much for your valuable suggestion. Below are our responses:
>
> ### **(1). The novelty of our work.**
>
> In addition to proposing a disentangled diffusion model, we introduce a novel credible subspace projection module (refer to Section 3.2). **These two modules work together to learn credible conditions and diffusion targets**—two essential components of diffusion-model-based recommenders. **To address the challenge posed by limited annotated data**, we further develop a progressive enhancement strategy for the credible projection module (refer to Section 3.3). Moreover, **considering the characteristics of recommendation tasks**, we incorporate a new preference-contrast term into the overall optimization objective (refer to Section 3.4). **All these components are tightly integrated to construct a reliable diffusion framework capable of generating both accurate and credible content recommendations.**
>
> ### **(2). Key contributions of our work.**
>
> This research represents one of the earliest efforts to enhance recommendation credibility. Moreover, **to the best of our knowledge, it is the first study to address this problem under conditions where only a limited portion of credibility labels is available, reflecting a more realistic scenario where only a small part of content items are verified.** Importantly, our key contribution is not the proposal of an entirely new diffusion paradigm. Rather, it lies in being **the first work to empirically and theoretically demonstrate why existing DM-based recommenders are at risk of generating uncredible recommendations**, which may have harmful social impacts, and in **proposing a novel solution to address this practical and important issue and gap.**
>
> ### **(3). Comparison with other DM-based methods.**
>
> We also agree that it is important to clarify how our model differs from existing DM-based recommenders. Thus, at the end of the Methodology section, we include a dedicated discussion comparing our model with DreamRec, DiffuRec, and PreferDiff from two perspectives: **(i) model architecture** and **(ii) objective formulation** (refer to Lines 365-377 in our updated version).
>
> - **Model architecture:** DreamRec, DiffuRec, and PreferDiff all adopt a single-channel diffusion architecture, in which a single condition is used to guide the generation of the target item. In contrast, Disco employs a disentangled diffusion architecture with dual channels, leveraging two conditions to guide the generation. This design plays a crucial role in separating preference-related information from uncredible content signals.
> - **Objective formulation:** DreamRec uses the standard ELBO objective for diffusion models, whereas PreferDiff adopts a variant ELBO combined with a Bayesian Personalized Ranking (BPR) loss. DiffuRec instead uses a cross-entropy (CE) objective, essentially turning a generative diffusion model into a discriminative one. By contrast, our model also belongs to a variant of the ELBO loss, but one specifically designed to achieve both accurate and credible generation—an ability that DreamRec, DiffuRec, and PreferDiff do not possess.

---

> ### Author Response · Authors · 2025-11-22
> **Response to Reviewer Vtvf -- Part 4**
>
> > **Weakness 4 & Question 4: Motivation of null-space projection** — “The motivation behind the use of null-space projection for filtering uncredible information is insufficiently explained and appears debatable. While the method is intended to suppress untrustworthy signals, the theoretical justification for why null-space projection effectively filters such content is not clearly articulated. Moreover, the authors do not provide any dedicated experiments to validate the effectiveness of this mechanism in isolating or removing uncredible information.”
>
> Thank you so much for your suggestions. Our response is as follows.
>
> ### **(1) Motivation of null-space projection.**
>
> Projecting the diffusion targets into the null space of uncredible content is **equivalent to projecting them into the orthogonal complement of uncredible content**. By doing so, the **diffusion targets after null-space projection become orthogonal to credible content, indicating that the uncredible content contained in the diffusion targets has been suppressed.** To better support this motivation, we provide a theoretical justification as below and our updated version.
>
> ### **(2) Theoretical justification.**
>
>  Moreover, we have **included a theoretical justification** of the credible subspace projection in the updated version of our paper (refer to Lines 1550-1574). The detailed derivation is presented as follows:
>
> We provide a theoretical justification of the credible subspace projection in the updated version of our paper (refer to Lines 1550-1574). The detailed derivation is as follows:
>
> Our credible subspace projection is constructed using SVD. Applying SVD to the uncredible feature matrix $\mathbf{F}^\top$ yields the eigenvector matrix $\mathbf{U}$ and the diagonal eigenvalue matrix $\mathbf{\Lambda}$. $\mathbf{U}$ and $\mathbf{\Lambda}$ can be expressed as $\mathbf{U}=[\mathbf{U}_1;\mathbf{U}_2]$ and $\mathbf{\Lambda} =[\mathbf{\Lambda}_1; \mathbf{\Lambda}_2]$. Correspondingly, $\mathbf{V}$ can be expressed as $\mathbf{V}=[\mathbf{V}_1;\mathbf{V}_2]$. All zero or near-zero singular values are contained in $\mathbf{\Lambda}_2$, and the corresponding eigenvectors are given by $\mathbf{U}_2$ and $\mathbf{V}_2$.
>
> According to the principles of SVD, the following equation holds:
>
> $$\mathbf{U}_2^\top\mathbf{F}^\top=\mathbf{U}_2^\top \mathbf{U}_1 \mathbf{\Lambda}_1 \mathbf{V}_1^\top.$$
>
> Since the matrix $\mathbf{U}$ obtained from the SVD is an orthogonal matrix, we have:
>
> $$\mathbf{U}_2^\top\mathbf{F}^\top=\underbrace{\mathbf{U}_2^\top \mathbf{U}_1}\_{=\mathbf{0}} \mathbf{\Lambda}_1 \mathbf{V}_1^\top=\mathbf{0}.$$
>
> Our credible diffusion target is derived through $\tilde{\mathbf{e}}_n=\mathbf{e}_n\mathbf{U}_2\mathbf{U}_2^\top$. Hence, we have:
>
> $$\tilde{\mathbf{e}}_n\mathbf{F}^\top=\mathbf{e}_n\mathbf{U}_2\underbrace{\mathbf{U}2^\top\mathbf{F}^\top}\_{=\mathbf{0}}=\mathbf{0}.$$
>
> This indicates that the derived credible diffusion target $\tilde{\mathbf{e}}_n$ is orthogonal to the uncredible feature matrix $\mathbf{F}$. **In summary, our proposed credible subspace projection effectively removes uncredible content from the diffusion targets by ensuring that the projected targets lie orthogonally to the uncredible content.**

---

> ### Author Response · Authors · 2025-11-22
> **Response to Reviewer Vtvf -- Part 5**
>
> > **Weakness 5 & Question 5: Datasets** — “The three datasets used in the experiments are relatively small and may not be sufficient to robustly validate the effectiveness and generalizability of the proposed model. To strengthen the empirical evaluation, the authors are encouraged to consider larger and more diverse datasets that better reflect real-world recommendation scenarios.”
>
> ### **Our used three datasets are the only publicly available datasets that satisfy the requirements of our task.**
>
> Thank you for your valuable insight. We acknowledge your concern about larger datasets. However, **these three datasets are the only publicly available datasets which can be used in our task.** The detailed reasons and explanations are as follows:
>
> Our work focuses on mitigating the recommendations of uncredible content to users, with the goal of enhancing user trust and addressing broader societal concerns. Therefore, **credibility labels are required for evaluating the effectiveness of both our method and baseline models** in the context of credible content recommendation.
>
> However, **widely used datasets such as Amazon (e.g., Beauty, Sports) do not contain credibility labels for items and thus cannot be used for our task.** Since credible content recommendation is still in its early stages, the three datasets we employ—PolitiFact, GossipCop, and MHMisinfo—are, to the best of our knowledge, the only publicly available datasets that include both user–item interaction data and items’ credibility labels.
>
> **We have made extensive efforts to search across GitHub, HuggingFace, and other public sources, but were unable to identify any additional datasets that satisfy these requirements.**
>
> We have added a clarification regarding this constraint in our updated version (refer to Lines 385-387).
>
> Looking ahead, we hope that as this research area evolves, more suitable datasets will be released to support advancements in credible content recommendation.
>
> ### **Justification of the dataset size.**
>
> We also acknowledge that larger datasets can improve the evaluation of our model. This is why we adopt a data augmentation strategy to construct significantly larger datasets, following prior work [5]. Table 1 reports the number of sequences for the datasets used in our experiments, including GossipCop, PolitFact, and the widely used Amazon datasets (Sports, Beauty, Toys, Automotive, Phones, and Tools).
>
> ### **Key Findings: Our used datasets have much more training sequences than other commonly used datasets.**
>
> As shown in Table 1, **the datasets we use (GossipCop and PolitiFact) contain significantly more sequences than three commonly used Amazon datasets (Sports, Beauty, and Toys).** Notably, the GossipCop dataset contains approximately **2–3 times more sequences than even the larger Amazon datasets (Automotive, Phones, and Tools).** Therefore, we argue that our used datasets provide a robust basis for validating both the effectiveness and generalizability of our model.
>
> **Table 1: User-item interaction sequence number of different datasets.**
>
> | Datasets | GossipCop | PolitiFact | Sports | Beauty | Toys | Automotive | Phones | Tools |
> | --- | --- | --- | --- | --- | --- | --- | --- | --- |
> | # sequences | 510,149 | 103,335 | 35,598 | 22,363 | 19,412 | 193,651 | 157,212 | 240,799 |
>
>
> [5] Zhengyi Yang, Jiancan Wu, Zhicai Wang, Xiang Wang, Yancheng Yuan, and Xiangnan He. Generate what you prefer: Reshaping sequential recommendation via guided diffusion. In NeurIPS, volume 36, 2023b.

---

> ### Author Response · Authors · 2025-11-26
> **A gentle reminder for discussion**
>
> Dear Reviewer Vtvf,
>
> Thank you very much for your constructive comments and valuable suggestions.
>
> We have carefully addressed your concerns by (1) clarifying the **evaluation metrics and the definition of credibility**; (2) clarifying **how item embeddings are derived**; (2) clarifying the **novelty of our approach and adding comparisons with existing DM-based methods**; (4) providing **theoretical evidence for the credible subspace projection**; and (5) clarifying the **selection of datasets**. Corresponding revisions have been incorporated into our updated version.
>
> We are kindly wondering if our responses have addressed your concerns. Your feedback is really important to us, and we are looking forward to your response and are happy to answer any further questions.
>
> &nbsp;
>
> Best regards,
>
> The Authors of Submission 11137

---

### Official Review · Reviewer_5evV · 2025-10-31

**Soundness:** 3
**Presentation:** 2
**Contribution:** 2
**Rating:** 4
**Confidence:** 4

**Summary:**

The paper proposes Disco, a novel framework aimed at steering diffusion models (DMs) towards credible content recommendations. While DMs have proven effective in recommendation systems, they risk generating uncredible content such as fake news or misinformation. Disco addresses this issue by introducing a disentangled diffusion model that separates credible user preferences from uncredible content. This separation ensures that content generation is guided by preferences while discouraging the inclusion of uncredible signals. Additionally, Disco incorporates a progressively enhanced credible subspace projection, which suppresses uncredible content by projecting the diffusion targets into a null space that excludes uncredible features. The effectiveness of Disco is demonstrated through experiments on real-world datasets, showing that it delivers both accurate and credible recommendations.

**Strengths:**

1. Innovative Approach to Credibility: Disco offers a pioneering solution to mitigate uncredible content generation in DMs, which is crucial for real-world recommendation systems, particularly in sensitive areas like news recommendations.

2. Disentanglement of Credible and Uncredible Content: The disentangled diffusion model efficiently separates preference-related content from uncredible content, preserving the user’s genuine preferences while filtering out harmful signals.

**Weaknesses:**

1. **Paper Writing**: The formatting in Section 2.2 is quite poor, which makes it difficult to follow the explanation. Furthermore, the terms preference-aware embedding and uncredible content-aware embedding are introduced, but their definitions and distinctions aren't clear until Section 3.1. This lack of clarity is problematic, especially in the context of recommendation tasks or datasets, as readers are left unsure about what these embeddings represent and how they differ from each other until much later in the paper. Providing clearer definitions and explanations earlier in the paper would greatly improve the readability and understanding of these concepts.

2. **Novelty**: The Credible Subspace Projection seems to be inspired by AlphaFuse and other heuristic methods. The formulation in Eq (4), where the two conditions are directly subtracted, also appears heuristic. It's worth questioning why this approach of direct subtraction was chosen—why not explore other possible forms?

3. **Experiments**: All the experiments are conducted on datasets related to Fake News or misinformation videos. Has the author tried applying the method to other domains, such as Beauty or Sports? Expanding the dataset to include more diverse domains would help verify the method's effectiveness and generalizability.

**Questions:**

See weaknesses.

---

> ### Author Response · Authors · 2025-11-22
> **Response to Reviewer 5evV -- Part 1**
>
> Thank you for your insightful comments and for highlighting the strengths of our work. We truly appreciate the thoughtful feedback and the three suggestions you raised. Below are our detailed responses. We would be delighted to engage further with you if you have additional questions and concerns.
>
> > **Weakness 1: Paper Writing** — “The formatting in Section 2.2 is quite poor, which makes it difficult to follow the explanation. Furthermore, the terms preference-aware embedding and uncredible content-aware embedding are introduced, but their definitions and distinctions aren't clear until Section 3.1. This lack of clarity is problematic, especially in the context of recommendation tasks or datasets, as readers are left unsure about what these embeddings represent and how they differ from each other until much later in the paper. Providing clearer definitions and explanations earlier in the paper would greatly improve the readability and understanding of these concepts.”
>
>
>
> Thank you for your valuable suggestion. Our responses are as follows:
>
> ### **(1). Revision of Section 2.2.**
>
> **To improve clarity and readability, we have revised Section 2.2** in the updated version of our paper by **reorganizing the structure and rewriting several descriptions to make the explanations more intuitive and accessible** (refer to Lines 151-172 in the updated version).
>
> ### **(2). Explanation of two concepts.**
>
> To improve the readability and understanding of these concepts (e.g., preference-aware embeddings and uncredible-aware embedding), we have **added detailed explanations** of these two concepts as well as related terms (e.g. preference-related signals and uncredible content-related signals) in Introduction in our updated version (refer to Lines 98-101).
>
> - **Preference-aware embeddings:** vector representations that encode item information related to users’ preferences, such as content topics.
> - **Uncredible content-aware embeddings:** vector representations that encode uncredible aspects of item content, such as inaccurate and misleading information.

---

> ### Author Response · Authors · 2025-11-22
> **Response to Reviewer 5evV -- Part 2**
>
> > **Weakness 2: Clarification on credible subspace projection**  — " The Credible Subspace Projection seems to be inspired by AlphaFuse and other heuristic methods.”
>
>
>
> Thank you so much for your valuable suggestion. We acknowledge your concern on credible subspace projection and the motivation of Equation (4). Below are our responses.
>
> ### **(1). Clarification of credible subspace projection.**
>
> Our proposed credible subspace is based on null space projection, which indeed also been applied in many other works and we have properly cited these works in the corresponding place of our paper. However, **our key contribution is that we are the first to design a credible subspace projection approach to remove uncredible content from item representations and the process of diffusion models**, so as to improve the credibility of content recommendation. The effectiveness of our credible subspace projection is also verified in our ablation study (w/o credible subspace projection (i.e., w/o CSP)), as shown in Table 1. **This part had already been included in our paper (refer to Lines 484, 490-491)**
>
> **Table 1: Ablation study of credible subspace projection. Politi, Gossip, MHMis are the abbreviations for PolitiFact, GossipCop and MHMisinfo.**
>
> | Model | Politi |  |  &nbsp; &nbsp; &nbsp; &nbsp; &nbsp; &nbsp; &nbsp; &nbsp; &nbsp;| Gossip |  |  &nbsp; &nbsp; &nbsp; &nbsp; &nbsp; &nbsp; &nbsp; &nbsp; &nbsp;| MHMis |  |  &nbsp; &nbsp; &nbsp; &nbsp; &nbsp; &nbsp; &nbsp; &nbsp; &nbsp;|
> | --- | --- | --- | --- | --- | --- | --- | --- | --- | --- |
> |  | HR@5 | CR@5 | HC@5 | HR@5 | CR@5 | HC@5 | HR@5 | CR@5 | HC@5 |
> | Disco | **0.2664** | **0.9835** | **0.3455** | **0.5236** | **0.9272** | **0.4918** | **0.2215** | **0.9305** | **0.3000** |
> | w/o Credible Subspace Projection | 0.2575 | 0.9431 | 0.3331 | 0.5183 | 0.9155 | 0.4860 | 0.2178 | 0.9066 | 0.2942 |
>
> ### **(2). Theoretical justification of credible subspace projection.**
>
> Moreover, we have **included a theoretical justification of the credible subspace projection in the updated version of our paper** (refer to Lines 1550-1574). The detailed derivation is presented as follows:
>
> We provide a theoretical justification of the credible subspace projection in the updated version of our paper (refer to Lines 1550-1574). The detailed derivation is as follows:
>
> Our credible subspace projection is constructed using SVD. Applying SVD to the uncredible feature matrix $\mathbf{F}^\top$ yields the eigenvector matrix $\mathbf{U}$ and the diagonal eigenvalue matrix $\mathbf{\Lambda}$. $\mathbf{U}$ and $\mathbf{\Lambda}$ can be expressed as $\mathbf{U}=[\mathbf{U}_1;\mathbf{U}_2]$ and $\mathbf{\Lambda} =[\mathbf{\Lambda}_1; \mathbf{\Lambda}_2]$. Correspondingly, $\mathbf{V}$ can be expressed as $\mathbf{V}=[\mathbf{V}_1;\mathbf{V}_2]$. All zero or near-zero singular values are contained in $\mathbf{\Lambda}_2$, and the corresponding eigenvectors are given by $\mathbf{U}_2$ and $\mathbf{V}_2$.
>
> According to the principles of SVD, the following equation holds:
>
> $$\mathbf{U}_2^\top\mathbf{F}^\top=\mathbf{U}_2^\top \mathbf{U}_1 \mathbf{\Lambda}_1 \mathbf{V}_1^\top.$$
>
> Since the matrix $\mathbf{U}$ obtained from the SVD is an orthogonal matrix, we have:
>
> $$\mathbf{U}_2^\top\mathbf{F}^\top=\underbrace{\mathbf{U}_2^\top \mathbf{U}_1}\_{=\mathbf{0}} \mathbf{\Lambda}_1 \mathbf{V}_1^\top=\mathbf{0}.$$
>
> Our credible diffusion target is derived through $\tilde{\mathbf{e}}_n=\mathbf{e}_n\mathbf{U}_2\mathbf{U}_2^\top$. Hence, we have:
>
> $$\tilde{\mathbf{e}}_n\mathbf{F}^\top=\mathbf{e}_n\mathbf{U}_2\underbrace{\mathbf{U}_2^\top\mathbf{F}^\top}\_{=\mathbf{0}}=\mathbf{0}.$$
>
> This indicates that the derived credible diffusion target $\tilde{\mathbf{e}}_n$ is orthogonal to the uncredible feature matrix $\mathbf{F}$. **In summary, our proposed credible subspace projection effectively removes uncredible content from the diffusion targets by ensuring that the projected targets lie orthogonally to the uncredible content.**

---

> ### Author Response · Authors · 2025-11-22
> **Response to Reviewer 5evV -- Part 3**
>
> > **Weakness 2: Clarification on Eq (4)**  — “The formulation in Eq (4), where the two conditions are directly subtracted, also appears heuristic. It's worth questioning why this approach of direct subtraction was chosen—why not explore other possible forms?”
>
> Thank you so much for your insightful comment. Our responses are as follows.
>
> Since our goal is to encourage the generation process conditioned on preference-related signals and discourage the generation process conditioned on uncredible content-related signals, thereby selecting the **subtraction operation is the most intuitive and straightforward practice**. Compared with other complex operations, subtraction has multiple advantages, such as **easy to implement and computational efficient**.
>
> However, we acknowledge your concern about whether other operations can have better performance. In response, we **design more variants and conduct experiments using these variants**.
>
> - **Variant 1:**
>
> $$
> \theta^*=\mathrm{arg}\ \underset{\theta}{\mathrm{min}}\ \ \frac{\mathbb{E}\_q\left[-\mathrm{log}\ \frac{p\_{\theta}(\mathbf{e}\_n^{0:T}|\mathbf{c}^{pre})}{q(\mathbf{e}\_n^{1:T}|\mathbf{e}\_n^0)}\right]}{\mathbb{E}\_q\left[-\mathrm{log}\ \frac{p\_{\theta}(\mathbf{e}\_n^{0:T}|\mathbf{c}^{unc})}{q(\mathbf{e}\_n^{1:T}|\mathbf{e}\_n^0)}\right]+\delta},
> $$
>
> where $\delta$ is a small constant introduced to avoid division by zero. In our implementation, we set $\delta=1\times10^{-8}$.
>
> - **Variant 2:**
>
> $$
> \theta^*=\mathrm{arg}\ \underset{\theta}{\mathrm{min}}\ \mathbb{E}\_q\left[-\mathrm{log}\ \frac{p\_{\theta}(\mathbf{e}_n^{0:T}|\mathbf{c}^{pre})}{q(\mathbf{e}_n^{1:T}|\mathbf{e}_n^0)}\right]+\frac{1}{\mathbb{E}\_q\left[-\mathrm{log}\ \frac{p\_{\theta}(\mathbf{e}_n^{0:T}|\mathbf{c}^{unc})}{q(\mathbf{e}_n^{1:T}|\mathbf{e}_n^0)}\right]+\delta}.
> $$
>
> - **Variant 3:**
>
> $$
> \theta^*=\mathrm{arg}\ \underset{\theta}{\mathrm{min}}\ \ - \text{log}\ \text{Sigmoid}\left(-\mathbb{E}\_q\left[-\mathrm{log}\ \frac{p\_{\theta}(\mathbf{e}_n^{0:T}|\mathbf{c}^{pre})}{q(\mathbf{e}_n^{1:T}|\mathbf{e}_n^0)}\right]+\mathbb{E}\_q\left[-\mathrm{log}\ \frac{p\_{\theta}(\mathbf{e}_n^{0:T}|\mathbf{c}^{unc})}{q(\mathbf{e}_n^{1:T}|\mathbf{e}_n^0)}\right]\right).
> $$
>
> - **Variant 4:**
>
> $$
> \theta^*=\mathrm{arg}\ \underset{\theta}{\mathrm{min}}\ \mathbb{E}\_q\left[-\mathrm{log}\ \frac{p\_{\theta}(\mathbf{e}_n^{0:T}|\mathbf{c}^{pre})}{q(\mathbf{e}_n^{1:T}|\mathbf{e}_n^0)}\right]\times \text{exp}\left(-\mathbb{E}\_q\left[-\mathrm{log}\ \frac{p\_{\theta}(\mathbf{e}_n^{0:T}|\mathbf{c}^{unc})}{q(\mathbf{e}_n^{1:T}|\mathbf{e}_n^0)}\right]\right).
> $$
>
> These four variants have the **same optimization direction with subtraction operation**. The comparison results between Disco and these four variants are reported in Table 2.
>
> ### **Finding: Subtraction operation used in Disco generally achieves the best performance than other variants.**
>
> From Table 2, we can see that the subtraction operation used in Disco generally achieves the best performance compared with other variants. This demonstrates that our choice of the subtraction operation is reasonable—straightforward yet effective.
>
> **Table 2. Performance comparison of Disco (using subtraction operation) with other variants.  Politi, Gossip, MHMis are the abbreviations for PolitiFact, GossipCop and MHMisinfo.**
>
> |  | **Politi** |  |  | **Gossip** |  |  | **MHMis** |  |  |
> | --- | --- | --- | --- | --- | --- | --- | --- | --- | --- |
> | **Variants** | HR@5 | CR@5 | HC@5 | HR@5 | CR@5 | HC@5 | HR@5 | CR@5 | HC@5 |
> | **Disco** | **0.2678** | **0.9823** | **0.3466** | **0.5236** | **0.9272** | **0.4918** | **0.2215** | 0.9305 | **0.3000** |
> | **Variant 1** | 0.2410 | 0.9147 | 0.3157 | 0.4493 | 0.9175 | 0.4540 | 0.1707 | 0.8921 | 0.2469 |
> | **Variant 2** | 0.2442 | 0.9491 | 0.3225 | 0.5005 | 0.9202 | 0.4795 | 0.1944 | 0.9247 | 0.2737 |
> | **Variant 3** | 0.2442 | 0.9491 | 0.3225 | 0.4802 | 0.9170 | 0.4691 | 0.2051 | **0.9352** | 0.2851 |
> | **Variant 4** | 0.2436 | 0.9446 | 0.3214 | 0.4511 | 0.9166 | 0.4547 | 0.1718 | 0.8926 | 0.2481 |

---

> ### Author Response · Authors · 2025-11-22
> **Response to Reviewer 5evV -- Part 4**
>
> > **Weakness 3: Datasets** — “All the experiments are conducted on datasets related to Fake News or misinformation videos. Has the author tried applying the method to other domains, such as Beauty or Sports? Expanding the dataset to include more diverse domains would help verify the method's effectiveness and generalizability.”
>
>
>
> ### **Our used three datasets are the only publicly available datasets that satisfy the requirements of our task.**
>
> Thank you for your valuable insight. We acknowledge your concern about more datasets. **However, these three datasets are the only publicly available dataset which can be used in our task.** The detailed reasons and explanations are as follows:
>
> Our work focuses on mitigating the recommendations of uncredible content to users, with the goal of enhancing user trust and addressing broader societal concerns. Therefore, **credibility labels are required for evaluating the effectiveness of both our method and baseline models** in the context of credible content recommendation.
>
> However, **widely used datasets such as Amazon (e.g., Beauty, Sports) do not contain credibility labels for items and thus cannot be used for our task.** Since credible content recommendation is still in its early stages, the three datasets we employ—PolitiFact, GossipCop, and MHMisinfo—are, to the best of our knowledge, the only publicly available datasets that include both user–item interaction data and items’ credibility labels.
>
> **We have made extensive efforts to search across GitHub, HuggingFace, and other public sources, but were unable to find any additional datasets that satisfy these requirements.**
>
> We have added a clarification regarding this constraint in our updated version (refer to Lines 385-387).
>
> Looking ahead, we hope that as this research area evolves, more suitable datasets will be released to support advancements in credible content recommendation.

---

> > ### Comment · Reviewer_5evV · 2025-11-23
> >
> > Thank you for your response. I believe the sections on preference-aware embeddings and uncredible content-aware embeddings—as well as the explanations of their corresponding tasks—are crucial for understanding the entire paper. I now fully understand why you chose not to use Amazon-related datasets. I appreciate your effort during the rebuttal stage, and I will raise my score accordingly.

---

> > > ### Author Response · Authors · 2025-11-24
> > >
> > > Thank you so much for your active feedback and engagement throughout the rebuttal process. We sincerely appreciate your support and endorsement.

---

### Official Review · Reviewer_o9ud · 2025-11-11

**Soundness:** 3
**Presentation:** 2
**Contribution:** 2
**Rating:** 4
**Confidence:** 2

**Summary:**

This paper proposes Disco, a diffusion model-based recommendation model designed to mitigate uncredible content while maintaining recommendation accuracy. It identifies two key factors that cause existing diffusion models to generate uncredible recommendations: (1) uncredible conditions when users have interacted with uncredible items, and (2) uncredible diffusion targets when the target item itself is uncredible. Disco addresses these through a disentangled diffusion model that separates preference-related and uncredible content signals, combined with a progressively enhanced credible subspace projection that projects diffusion targets into the null space of uncredible content features.

**Strengths:**

1. Credible content recommendation is a new but important problem. The paper addresses a significant real-world issue - recommender systems amplifying misinformation and fake news.
2. The paper considers a realistic scenario where only 20% of uncredible items are labeled during training, which can be more representative than full label availability.

**Weaknesses:**

1. The strong performance relies heavily on comparing against baselines (Traditional, Contrastive, and generic DM-based methods) that were not originally designed to address credible content recommendation under conditions of limited labels.
2. Achieving strong performance requires a large embedding dimension (e.g., 3072 for DM-based methods) and a large number of diffusion steps. How to deal with the tradeoff between efficiency and performance is a challenge.
3. The overall framework is intricate, integrating a disentangled diffusion model, a separate credible subspace projection, and a preference contrast term. The paper does not well explore the intrinsic connections or necessity of coupling all these components, potentially make the design of the model a bit too complex.

**Questions:**

1. If Disco uses a low-dimensional embedding (e.g., 64 or 128) or fewer time steps, how does it perform comparable to the non-DM baselines?
2. How does the method perform when the proportion of labeled uncredible items varies (not just 20%)?

---

> ### Author Response · Authors · 2025-11-22
> **Response to Reviewer o9ud -- Part 1**
>
> Thank you very much for highlighting our strengths and for providing such valuable feedback. We sincerely appreciate your thoughtful insights and suggestions. These points are both essential and constructive, and they have prompted us to carefully address each issue, which has significantly improved the quality of our paper. Below, we present our detailed responses. We would be delighted to further engage with you if you have any additional questions.
>
> > **Weakness 1: Lack of baselines addressing limited labels** — “The strong performance relies heavily on comparing against baselines (Traditional, Contrastive, and generic DM-based methods) that were not originally designed to address credible content recommendation under conditions of limited labels.
>
> Thank you for your comment. Our responses are as follows.
>
> ### **(1). Our work is the first to address credible content recommendation under the condition of limited credibility labels.**
>
> We acknowledge your concern on baselines designed to address credible content recommendation under conditions of limited labels. However, to the best of our knowledge, our paper is the first to address credible content recommendation under the condition of limited credibility labels.
>
> ### **(2). No existing methods are designed for this limited-label setting.**
>
> While several prior studies have explored credible recommendation, **they all require full access to complete credibility labels** for all items (as noted in **Lines 86–88** in the updated version of our paper). Therefore, no existing methods are designed for this limited-label setting, and thus no appropriate baselines are available for comparison.
>
> We emphasize that considering this more realistic scenario of limited credibility labels is one of our key contributions and a major factor that distinguishes our work from prior studies. To further highlight our contribution and novelty, we have added the following sentence to the summary of contributions: **“To the best of our knowledge, Disco is the first work designed for credible content recommendation under the condition of limited credibility labels.”**

---

> ### Author Response · Authors · 2025-11-22
> **Response to Reviewer o9ud -- Part 2**
>
> > **Weakness 2 & Question 1: High embedding dimension** — “Achieving strong performance requires a large embedding dimension (e.g., 3072 for DM-based methods). How to deal with the tradeoff between efficiency and performance is a challenge. If Disco uses a low-dimensional embedding (e.g., 64 or 128), how does it perform comparable to the non-DM baselines”
>
> Thank you for raising this question. We fully understand and appreciate this concern, **which we have also acknowledged as a limitation in our paper (see Lines 536-539).** Below, we first explain why a high embedding dimension is necessary for DM-based recommenders. Moreover, we also point out that **Disco can outperform non-DM based methods when embeddings size is set to 64 only with some small adjustment to the overall loss**.  Finally, we **provide some insights on how to deal with the tradeoff between efficiency and performance**. We have added this content in our updated version.
>
> ### **(1). Necessity of high embedding dimensions for DM-based recommenders.**
>
> **This issue is not unique to our model, Disco, but is inherent to the DM-based recommenders**, as previously pointed out in [1]. The ****root cause lies in the variance-preserving nature of diffusion models, which generally requires high-dimensional embeddings to maintain performance and training stability [1].
>
> ### **(2). Experimental results of Disco when embedding size is set to 64.**
>
> We have evaluated all DM-based methods with the embedding size fixed to 64, and the results are reported in Table 1. **More experiments results can be found in our updated version (refer to Lines 1026-1070 in Page 20).**
>
> As shown in Table 1, **not only Disco but also other DM-based recommendation methods (PRISM, DreamRec, and PreferDiff) experience a substantial performance drop when the embedding size is reduced to 64.** This observation is consistent with the findings reported in [1] and further validates the necessity of using high-dimensional embeddings in DM-based recommender systems.
>
> **Table 1: Performance comparison under embedding dimension  64 for DM-based methods. Politi, Gossip, MHMis are the abbreviations for PolitiFact, GossipCop and MHMisinfo.**
>
> | **Methods** | **Politi** |  |  &nbsp; &nbsp; &nbsp; &nbsp; &nbsp; &nbsp; &nbsp; &nbsp; &nbsp;| **Gossip** |  |  &nbsp; &nbsp; &nbsp; &nbsp; &nbsp; &nbsp; &nbsp; &nbsp; &nbsp;| **MHMis** |  |  |
> | --- | --- | --- | --- | --- | --- | --- | --- | --- | --- |
> |  | HR@5 | CR@5 | HC@5 | HR@5 | CR@5 | HC@5 | HR@5 | CR@5 | HC@5 |
> | GRU4Rec | 0.2142 | 0.9266 | 0.2929 | 0.2226 | 0.8864 | 0.2957 | 0.1151 | 0.8380 | 0.1803 |
> | SASRec | 0.2158 | 0.9059 | 0.2923 | 0.3078 | 0.8743 | 0.3612 | 0.1485 | 0.8339 | 0.2190 |
> | Bert4Rec | 0.2191 | 0.9127 | 0.2960 | 0.2372 | 0.8764 | 0.3078 | 0.1391 | 0.8162 | 0.2074 |
> | CL4SRec | 0.2247 | 0.9132 | 0.3012 | 0.2898 | 0.8938 | 0.3516 | 0.1734 | 0.8577 | 0.2469 |
> | PRISM (emb_size=3072) | 0.1927 | 0.9325 | 0.2727 | 0.2948 | 0.8806 | 0.3531 | 0.1700 | 0.8398 | 0.2418 |
> | PRISM (emb_size=64) | 0.0806 | 0.7807 | 0.1336 | 0.0023 | 0.6570 | 0.0046 | 0.0239 | 0.7095 | 0.0448 |
> | DreamRec (emb_size=3072) | 0.2416 | 0.8620 | 0.3054 | 0.4619 | 0.8464 | 0.4415 | 0.1819 | 0.9002 | 0.2633 |
> | DreamRec (emb_size=64) | 0.0814 | 0.5744 | 0.1268 | 0.0036 | 0.5791 | 0.0071 | 0.0282 | 0.8281 | 0.0528 |
> | PreferDiff (emb_size=3072) | 0.2531 | 0.8925 | 0.3228 | 0.4969 | 0.8307 | 0.4523 | 0.1974 | 0.8693 | 0.2713 |
> | PreferDiff (emb_size=64) | 0.1227 | 0.8934 | 0.1925 | 0.0084 | 0.6542 | 0.0164 | 0.0325 | 0.8290 | 0.0603 |
> | Disco (emb_size=3072) | 0.2678 | 0.9823 | 0.3466 | 0.5236 | 0.9272 | 0.4918 | 0.2215 | 0.9305 | 0.3000 |
> | Disco (emb_size=64) | 0.1171 | 0.9916 | 0.1895 | 0.0087 | 0.7993 | 0.0170 | 0.0123 | 0.9968 | 0.0240 |
>
> ### **Concern about effectiveness-efficiency trade-off.**
>
> Although a higher embedding dimension increases the per-epoch training time, it also provides the benefit of significantly faster convergence. In our updated version, we include a figure illustrating the convergence curves of Disco and a representative non-DM-based recommender SASRec. As presented in Figure 6 (Page 21), **Disco converges much more rapidly than SASRec** (embedding size = 64). Specifically, Disco reaches its best performance at approximately the 40-th epoch, whereas SASRec requires around 400 epochs. **This fast convergence rate of Disco partially offsets the additional computational cost introduced by high-dimensional embeddings.**
>
> [1] Shuo Liu, An Zhang, Guoqing Hu, Hong Qian, and Tat-seng Chua. Preference diffusion for recommendation. In ICLR, 2025a.

---

> ### Author Response · Authors · 2025-11-22
> **Continue on Part 2**
>
> ### **(3). Our method can still achieve superior performance compared with non-DM-based methods only with a minor modification.**
>
> In addition, we conducted further experiments to demonstrate that **Disco can still achieve superior performance compared with non-DM-based methods when the embedding dimension is restricted to 64, as long as a minor modification is applied to the overall optimization objective**. In particular, we augment Disco’s loss function with an additional Cross-Entropy term:
>
> $$\mathcal{L}\_{\texttt{Disco}^*}=\mathcal{L}\_{\texttt{Disco}}-\text{log}\left(\frac{\text{exp}(f\_{\theta}(\tilde{\mathbf{e}}\_n^{t},\mathbf{c}^{pre},t)\cdot\mathbf{e}\_n^\top)}{\sum\_{i\in\mathcal{I}}\text{exp}(f\_{\theta}(\tilde{\mathbf{e}}\_n^{t},\mathbf{c}^{pre},t)\cdot\mathbf{e}\_i^\top)}\right).$$
>
> The added term encourages the diffusion network $f_\theta$ to align its predictions more closely to the target items than with other items. Using this enhanced loss function $\mathcal{L}_{\texttt{Disco}^*}$, Disco can achieve better performance than non-DM based methods. The comparison results are reported in Table 2. **More experimental results can be found in our updated version (refer to Lines 1097-1120 in Page 21).** From Table 2, we have the following observation:
>
> - **Disco can achieve better performance than non-DM recommendation methods with only a minor adjustment to its overall loss.** This improvement arises because adding a discriminative loss (i.e., Cross Entropy loss) to the generative loss ($\mathcal{L}_{\texttt{Disco}}$) can partially mitigate the dimensionality limitations inherent to diffusion models.
>
> **However, this practice will transform a purely generative model into a discriminative one. Our work does not make such a compromise.**
>
> Nevertheless, the results clearly suggest that **our model retains strong potential to surpass non-DM-based recommenders even when operating with substantially reduced embedding dimensions**. This highlights the potential of Disco to achieve an effective trade-off between efficiency and performance.
>
> **Table 2: Performance comparison of Disco (optimized by $\mathcal{L}_{\texttt{Disco}^*}$) and non-DM recommendation methods when embedding dimension is se to 64. Politi, Gossip, MHMis are the abbreviations for PolitiFact, GossipCop and MHMisinfo.**
>
> | **Methods** | **Politi** |  |  &nbsp; &nbsp; &nbsp; &nbsp; &nbsp; &nbsp; &nbsp; &nbsp; &nbsp;| **Gossip** |  |  &nbsp; &nbsp; &nbsp; &nbsp; &nbsp; &nbsp; &nbsp; &nbsp; &nbsp;| **MHMis** |  |  &nbsp; &nbsp; &nbsp; &nbsp; &nbsp; &nbsp; &nbsp; &nbsp; &nbsp;|
> | --- | --- | --- | --- | --- | --- | --- | --- | --- | --- |
> |  | HR@5 | CR@5 | HC@5 | HR@5 | CR@5 | HC@5 | HR@5 | CR@5 | HC@5 |
> | GRU4Rec | 0.2142 | 0.9266 | 0.2929 | 0.2226 | 0.8864 | 0.2957 | 0.1151 | 0.8380 | 0.1803 |
> | SASRec | 0.2158 | 0.9059 | 0.2923 | 0.3078 | 0.8743 | 0.3612 | 0.1485 | 0.8339 | 0.2190 |
> | Bert4Rec | 0.2191 | 0.9127 | 0.2960 | 0.2372 | 0.8764 | 0.3078 | 0.1391 | 0.8162 | 0.2074 |
> | CL4SRec | 0.2247 | 0.9132 | 0.3012 | 0.2898 | 0.8938 | 0.3516 | 0.1734 | 0.8577 | 0.2469 |
> | **Disco (emb_size=64, $\mathcal{L}_{\texttt{Disco}^*}$)** | **0.2335** | **0.9316** | **0.3111** | **0.4250** | **0.9151** | **0.4407** | **0.1856** | **0.8669** | **0.2599** |
>
> ### **(3). Some insights on how to deal with the tradeoff between efficiency and performance.**
>
> Inspired by the above results, we argue that one possible way to preserve model performance when reducing the embedding dimension is to **incorporate a discriminative loss (e.g., BCE, CE, BPR) into the generative objective.** Another promising direction is to **develop embedding-dimension truncation strategies specifically tailored for diffusion models**, enabling them to maintain performance under lower-dimensional settings.

---

> ### Author Response · Authors · 2025-11-22
> **Response to Reviewer o9ud -- Part 3**
>
> > **Question 1: High diffusion steps** — “If Disco uses fewer times steps, how does it perform comparable to the non-DM baselines?”
>
>
>
> Thank you for your insightful feedback. We acknowledge your concern on whether our model can achieve comparable performance than non-DM baselines. In response, we **conducted additional experiments** to investigate the performance of our model under different values of maximum diffusion step $T$, the results are shown in Table 3. **More results can be found in our updated version (refer to Lines 1155-1187 in Page 22).**
>
> ### **Key findings: Our method maintains superior performance compared with non-DM methods even at much smaller diffusion step**
>
> - **Finding 1.** As the diffusion step $T$ increases, the performance of our proposed model, Disco, improves.
> - **Finding 2.** **Disco maintains superior performance compared with non-DM methods even at much smaller diffusion step** ($T$=100 for PolitiFact and GossiCop, and $T$=500 for MHMisinfo).
>
> **Table 3: Performance comparison of Disco and non-DM recommendation methods under different diffusion steps T. Politi, Gossip, MHMis are the abbreviations for PolitiFact, GossipCop and MHMisinfo.**
>
> | Methods | Politi |  |  &nbsp; &nbsp; &nbsp; &nbsp; &nbsp; &nbsp; &nbsp; &nbsp; &nbsp;| Gossip |  |  &nbsp; &nbsp; &nbsp; &nbsp; &nbsp; &nbsp; &nbsp; &nbsp; &nbsp;| MHMis |  |  &nbsp; &nbsp; &nbsp; &nbsp; &nbsp; &nbsp; &nbsp; &nbsp; &nbsp;|
> | --- | --- | --- | --- | --- | --- | --- | --- | --- | --- |
> |  | HR@5 | CR@5 | HC@5 | HR@5 | CR@5 | HC@5 | HR@5 | CR@5 | HC@5 |
> | GRU4Rec | 0.2142 | 0.9266 | 0.2929 | 0.2226 | 0.8864 | 0.2957 | 0.1151 | 0.8380 | 0.1803 |
> | SASRec | 0.2158 | 0.9059 | 0.2923 | 0.3078 | 0.8743 | 0.3612 | 0.1485 | 0.8339 | 0.2190 |
> | Bert4Rec | 0.2191 | 0.9127 | 0.2960 | 0.2372 | 0.8764 | 0.3078 | 0.1391 | 0.8162 | 0.2074 |
> | CL4SRec | 0.2247 | 0.9132 | 0.3012 | 0.2898 | 0.8938 | 0.3516 | 0.1734 | 0.8577 | 0.2469 |
> | Disco (Diffusion step $T$=100) | 0.2494 | 0.9488 | 0.3269 | 0.4603 | 0.9304 | 0.4627 | 0.1378 | 0.8526 | 0.2083 |
> | Disco (Diffusion step $T$=200) | 0.2602 | 0.9427 | 0.3353 | 0.4759 | 0.9242 | 0.4689 | 0.1393 | 0.9161 | 0.2136 |
> | Disco (Diffusion step $T$=500) | 0.2591 | 0.9434 | 0.3345 | 0.4867 | 0.9258 | 0.4745 | 0.1819 | 0.9076 | 0.2597 |
> | Disco (Diffusion step $T$=1000) | 0.2525 | 0.9488 | 0.3296 | 0.4936 | 0.9202 | 0.4763 | 0.2144 | 0.9379 | 0.2943 |
> | Disco (Diffusion step $T$=2000) | 0.2678 | 0.9823 | 0.3466 | 0.5236 | 0.9272 | 0.4918 | 0.2215 | 0.9305 | 0.3000 |
>
> ### **How our work deal with the efficiency of diffusion step under high diffusion steps?**
>
> We acknowledge that a larger diffusion step $T$ increases the computational cost of the generation process. Importantly, **we have proactively addressed this issue in our work by replacing the DDPM paradigm used in DreamRec and DiffuRec with the DDIM paradigm** (refer to Line 940 and Line 994-997 in the updated version), which provides substantially higher sampling efficiency. In our experiments, we set $T$ = 2000 and apply a DDIM skipping step of 100, meaning that **Disco requires only 2000/100=20 generation steps in total.** This is significantly more efficient than both DreamRec and DiffuRec, while maintaining strong recommendation performance.

---

> ### Author Response · Authors · 2025-11-22
> **Response to Reviewer o9ud -- Part 4**
>
> > **Question 2: Performance under different proportion of labeled data** — “How does the method perform when the proportion of labeled uncredible items varies (not just 20%)?”
>
> Thank you for your valuable suggestion. We acknowledge your concern on the performance of Disco under different label ratio (not just 20%). In response, we **conduct additional experiments** under different credibility label ratio of uncredible content (i.e., **5%, 10%, 20%, 30%, 50%**). The experimental results are reported in Table 4. **More results can be found in our updated version (refer to Lines 1104-1124 in Page 23).**
>
> ### **Key Findings: Even when only 5% of credibility labels are available, our method still achieves superior performance.**
>
> - **Finding 1:** **As the credibility label ratio increases, the recommendation credibility (CR) improves steadily.** This is because a larger number of credibility labels enables the construction of a more comprehensive and accurate credible subspace, allowing uncredible content to be mitigated more effectively.
> - **Finding 2: Even when only 5% of credibility labels are available, Disco still achieves superior performance** compared to competitive methods in terms of both recommendation credibility and recommendation.
> - **Finding 3: The recommendation accuracy fluctuates only slightly within a narrow range across different label ratios.** This stability is attributed to our proposed disentangled diffusion model, which effectively mitigates uncredible content while preserving users’ preference-related information, thereby maintaining high recommendation accuracy.
>
> **Table 4: Performance comparison under different label ratios. Politi, Gossip, MHMis are the abbreviations for PolitiFact, GossipCop and MHMisinfo.**
>
> | Methods (Label ratio) | Politi |  |  &nbsp; &nbsp; &nbsp; &nbsp; &nbsp; &nbsp; &nbsp; &nbsp; &nbsp;| Gossip |  |  &nbsp; &nbsp; &nbsp; &nbsp; &nbsp; &nbsp; &nbsp; &nbsp; &nbsp;| MHMis |  |  &nbsp; &nbsp; &nbsp; &nbsp; &nbsp; &nbsp; &nbsp; &nbsp; &nbsp;|
> | --- | --- | --- | --- | --- | --- | --- | --- | --- | --- |
> |  | HR@5 | CR@5 | HC@5 | HR@5 | CR@5 | HC@5 | HR@5 | CR@5 | HC@5 |
> | GRU4Rec | 0.2142 | 0.9266 | 0.2929 | 0.2226 | 0.8864 | 0.2957 | 0.1151 | 0.8380 | 0.1803 |
> | SASRec | 0.2158 | 0.9059 | 0.2923 | 0.3078 | 0.8743 | 0.3612 | 0.1485 | 0.8339 | 0.2190 |
> | Bert4Rec | 0.2191 | 0.9127 | 0.2960 | 0.2372 | 0.8764 | 0.3078 | 0.1391 | 0.8162 | 0.2074 |
> | CL4SRec | 0.2247 | 0.9132 | 0.3012 | 0.2898 | 0.8938 | 0.3516 | 0.1734 | 0.8577 | 0.2469 |
> | Rec4Mit (20%) | 0.2185 | 0.8959 | 0.2876 | 0.2775 | 0.8979 | 0.3427 | 0.1460 | 0.8424 | 0.2166 |
> | HDInt (20%) | 0.2153 | 0.8946 | 0.2906 | 0.3407 | 0.8986 | 0.3875 | 0.1471 | 0.8306 | 0.2168 |
> | PRISM (20%) | 0.1927 | 0.9325 | 0.2727 | 0.2948 | 0.8806 | 0.3531 | 0.1700 | 0.8398 | 0.2418 |
> | Disco (5%) | 0.2617 | 0.9422 | 0.3365 | 0.5179 | 0.9261 | 0.4889 | 0.2127 | 0.9149 | 0.2904 |
> | Disco (10%) | 0.2541 | 0.9476 | 0.3308 | 0.5290 | 0.9266 | 0.4940 | 0.2112 | 0.9217 | 0.2897 |
> | Disco (20%) | 0.2678 | 0.9823 | 0.3466 | 0.5236 | 0.9272 | 0.4918 | 0.2215 | 0.9305 | 0.3001 |
> | Disco (30%) | 0.2704 | 0.9829 | 0.3489 | 0.5196 | 0.9278 | 0.4902 | 0.2207 | 0.9303 | 0.2994 |
> | Disco (50%) | 0.2678 | 0.9842 | 0.3468 | 0.5151 | 0.9284 | 0.4883 | 0.2134 | 0.9331 | 0.2929 |

---

> ### Author Response · Authors · 2025-11-22
> **Response to Reviewer o9ud -- Part 5**
>
> > **Weakness 3: Framework clarification** — “The overall framework is intricate, integrating a disentangled diffusion model, a separate credible subspace projection, and a preference contrast term. The paper does not well explore the intrinsic connections or necessity of coupling all these components, potentially make the design of the model a bit too complex.”
>
> Thank you for your valuable suggestion. To better demonstrate the interconnections and the necessity of integrating all the proposed components, we have **revised the introductory paragraph at the beginning of Section 3** “The Disco Model”. Below is our corresponding revision by **adding more description to emphasize the connections between multiple components.** The updated description is as follows:
>
> “In this section, we first introduce our disentangled diffusion model (Section 3.1), followed by the projection of diffusion targets into a credible subspace (Section 3.2). **These two components jointly enable the learning of credible conditions and credible diffusion targets (i.e., two essential elements in diffusion models)** to guide the model toward credible generation. Subsequently, **to address the more realistic scenario where only a limited portion of content items are labeled with credibility information**, we propose a progressive enhancement mechanism for the credible subspace (Section 3.3). Thereafter, we present the overall optimization objective of our proposed model, which integrates a content disentanglement term and a preference contrast term **to simultaneously enhance recommendation credibility and accuracy** (Section 3.4). Finally, we detail the credible generation and recommendation process after training (Section 3.5). **All components are interlocked to construct a unified diffusion-based framework for accurate and credible content recommendation under limited credibility supervision.**”

---

> ### Author Response · Authors · 2025-11-26
> **A gentle reminder for discussion**
>
> Dear Reviewer o9ud,
>
> Thank you very much for your constructive comments and valuable suggestions.
>
> We have carefully addressed your concerns by **clarifying the selection of baseline methods, conducting additional experiments on embedding dimension, diffusion steps, and credibility label ratio**. These revisions have been incorporated into our updated version.
>
> We are kindly wondering if our responses have addressed your concerns. Your feedback is really important to us, and we are looking forward to your response and are happy to answer any further questions.
>
> &nbsp;
>
> Best regards,
>
> The Authors of Submission 11137

---

> > ### Comment · Reviewer_o9ud · 2025-11-27
> >
> > Thank you for answering my questions. I am still confused about the diffusion steps and how to balance the performance and efficiency. Could you clarify that to achieve the optimal performance, how many diffusion steps are required during training and inference?

---

> > > ### Author Response · Authors · 2025-11-27
> > > **Rebuttal follow-up**
> > >
> > > Thank you very much for your response and follow-up question. To achieve optimal performance, our method uses **2000 diffusion steps during the training stage** (this does **not affect training efficiency**), while the number of **diffusion steps during inference is 20**, which is **significantly smaller than those used in other methods**, including DiffuRec and DreamRec. Our detailed explanation and responses are provided below.
> > >
> > > &nbsp;
> > >
> > > ### **(1) Training stage: The number of diffusion steps does not affect training efficiency.**
> > >
> > > - **Experimental Evidence:**
> > >
> > > We conduct **additional experiments** to record Disco’s training time for each epoch under different total diffusion step $T$ on PolitiFact dataset.
> > >
> > > **Table 5: The training time for epoch under different diffusion step $T$ on PolitiFact datasets.**
> > > |Total diffusion step $T$|100|200|500|1000|2000|
> > > |-|-|-|-|-|-|
> > > |Training time / epoch|12.55 s|12.57 s|12.57 s|12.54 s|12.58 s|
> > >
> > > ### **Key Finding: The diffusion step $T$ does not affect the training efficiency.**
> > >
> > > The training time only fluctuates slightly due to machine load.
> > >
> > > - **Theoretical explanation:**
> > >
> > > To clarify why the number of diffusion steps does not affect training efficiency, we provide the general optimization objective used in DM-based methods:
> > >
> > > $$\mathcal{L}=\mathbb{E}\_{\mathbf{e}\_n^0,\mathbf{c},t\sim U(0,T)}\left[\Vert\mathbf{e}\_n^{0}-f\_{\theta}(\mathbf{e}\_n^t,\mathbf{c},t)\Vert^2\_2\right],$$
> > >
> > > where **$t$ is sampled from a uniform distribution $U(0,T)$**. In this process, **only a sampled step $t$ is involved in optimization, instead of all steps from 0 to $T$.**
> > >
> > > Moreover, the **variable $\mathbf{e}_n^t$ does not need to be computed step-by-step through the chain $\mathbf{e}_n^0\rightarrow\mathbf{e}_n^1\rightarrow\cdots\rightarrow\mathbf{e}_n^{t-1}\rightarrow\mathbf{e}_n^t$. Instead, it can be directly calculated from $\mathbf{e}_n^0$ using the Markov chain principle:**
> > >
> > > $$\mathbf{e}_n^t=\sqrt{\bar{\alpha}_t}\mathbf{e}_n^0+\sqrt{1-\bar{\alpha}_t}\boldsymbol{\epsilon},$$
> > >
> > > where $\bar{\alpha}_t$ is a pre-calculated constant.
> > >
> > > Based on this analysis, we can have the following conclusion:
> > >
> > > - The **computation complexity of $\mathbf{e}_n^t$ and $\mathcal{L}$ does not grow with total diffusion step $T$**;
> > >
> > > ### **Therefore, the training efficiency remains unaffected by the value of $T$.**
> > >
> > > Our method follows exactly this principle; hence, the training efficiency of **Disco is not influenced by the total diffusion steps**.
> > >
> > > &nbsp;
> > >
> > > ### **(2) Inference stage: we utilize DDIM [2] to significantly reduce the generation steps in inference stage.**
> > >
> > > - **Theoretical explanation:**
> > >
> > > The generation process of the DDPM paradigm [3] follows the step-by-step chain $\mathbf{e}_n^T\rightarrow\mathbf{e}_n^{T-1}\rightarrow\cdots\rightarrow\mathbf{e}_n^{1}\rightarrow\mathbf{e}_n^0$, thereby is time-consuming.
> > >
> > > **To improve inference efficiency, in our method, we apply DDIM paradigm** (skip-step=100) whose generation chain becomes: $\mathbf{e}_n^T\rightarrow\mathbf{e}_n^{T-100}\rightarrow\cdots\rightarrow\mathbf{e}_n^{100}\rightarrow\mathbf{e}_n^0$. Thus, the total inference step is $T/100=2000/100=20$ in our method.
> > >
> > > - **More experimental results on inference efficiency.**
> > >
> > > As shown in Table 3 (Part 3 in rebuttal), **even with a much smaller diffusion step $T=100$, our method still substantially outperforms non-DM-based methods** on both PolitiFact and GossipCop datasets.
> > >
> > > Under these settings, the inference costs of Disco and other methods are summarized in Table 6. Since Disco adopts a Transformer as its sequence encoder, we only include Transformer-based baselines for a fair comparison.
> > >
> > > **Table 6: The inference time of Disco ($T=100$) and  no-DM-based methods.**
> > >
> > > |Methods|PolitiFact|GossipCop|MHMisinfo|
> > > |-|-|-|-|
> > > |SASRec (emb_size=64)|2.30 s|89.58 s|4.11 s|
> > > |Bert4Rec (emb_size=64)|2.40 s|93.86 s|4.55 s|
> > > |CL4SRec (emb_size=64)|2.41 s|92.09 s|4.15 s|
> > > |Disco (emb_size=3072)|6.31 s|130.78 s|9.44 s|
> > > |Disco (emb_size=64, $\mathcal{L}_{\texttt{Disco}^*}$)|2.33 s|87.28 s|4.11 s|
> > >
> > > ### **Findings: Disco achieves inference efficiency comparable to non-DM-based methods, especially with the refined version presented in Part 2 of the rebuttal.**
> > >
> > > - Disco **does not substantially increase inference cost, even under much large embedding dimensions**. Although higher embedding dimensions naturally require more computational time, the **additional overhead introduced by Disco remains acceptable, especially considering significant performance gain.**
> > > - The **enhanced version of Disco has comparable efficiency with other methods.**
> > >
> > > &nbsp;
> > >
> > > ### **In summary, Disco significantly outperforms baseline methods in terms of recommendation credibility and accuracy, while still maintaining high efficiency.**
> > > &nbsp;
> > >
> > > [2] Song J, Meng C, Ermon S. Denoising Diffusion Implicit Models. In ICLR. 2020.
> > >
> > > [3] Ho J, Jain A, Abbeel P. Denoising diffusion probabilistic models. In NeurIPS. 2020;33:6840-51.

---

> > > > ### Comment · Reviewer_o9ud · 2025-11-27
> > > >
> > > > Thank you for addressing my concern. I will update my score.

---

> > > > > ### Author Response · Authors · 2025-11-28
> > > > >
> > > > > Thank you very much for your active engagement and follow-up communication throughout the rebuttal process. We sincerely appreciate your support and endorsement.

---

### Author Response · Authors · 2025-11-24
**A gentle reminder for your valuable input for rebuttal and feedback**

Dear all Reviewers,

Thank you once again for your detailed comments and valuable feedback, which have greatly contributed to improving the quality of our work.

We kindly ask if our responses have adequately addressed your concerns. Your feedback is incredibly important to us, and we are looking forward to your responses and are happy to answer any further questions.

If you feel that our revisions and clarifications have adequately resolved the key issues raised, we would deeply appreciate your consideration in updating your evaluation.

Thank you again for your time and constructive comments. We look forward to hearing from you.

&nbsp;

Best regards,

The Authors of Submission 11137

---

### Author Response · Authors · 2025-12-03
**A summary of our rebuttal (Part 2/2)**

Dear AC,

Because Reviewer Vtvf and Reviewer fb2b did not participate in the rebuttal, we summarize their concerns and our responses below for your convenience.

## **Summary of Reviewer Vtvf’s and fb2b’s concerns and our rebuttal**

### **(1). Reviewer Vtvf’s Feedback & Our Reponses**

### **The reviewer has overlooked several important parts of our method**. Specifically:

> The reviewer stated that we did not use a credibility metric :

- **We clarified that we indeed have used a credibility metric CR@K** (Lines 401–402 and 896–908) and a joint accuracy–credibility metric **HC@K** (Lines 402–403 and 909–917).

> The reviewer stated that our method was based on ID embeddings:

- **We clarified that our model uses language embeddings derived from LLaMA2-7B** (Lines 927–929), **rather than ID embeddings as assumed by the reviewer.**

> The reviewer stated that our method primarily involved an disentanglement strategy:

- We clarify that our method not only involving adding a disentanglement strategy. Specifically, In addition to proposing a disentangled diffusion model, we introduce a novel credible subspace projection module (refer to Section 3.2). **These two modules work together to learn credible conditions and diffusion targets**—two essential components of diffusion-model-based recommenders. **To address the challenge posed by limited annotated data**, we further develop a progressive enhancement strategy for the credible projection module (refer to Section 3.3). Moreover, **considering the characteristics of recommendation tasks**, we incorporate a new preference-contrast term into the overall optimization objective (refer to Section 3.4). **All these components are tightly integrated to construct a reliable diffusion framework capable of generating both accurate and credible content recommendations.**


### **We have also carefully addressed other comments.**

> The reviewer suggested us to provide a comparison between existing DM-based methods:

- We have **provided a comparison analysis with existing DM-based methods** from perspective of model architecture and objective formulation.

> The reviewer suggested us to provided a theoretical justification of our credible subspace projection:

- We have **provided a theoretical justification** to support our design.

> The reviewer raised the concern about why not use more datasets:

- We clarified that, **due to the nature of our task (requiring both user-item interaction data and credibility labels of content items), the three datasets we used are the only publicly available datasets** appropriate for evaluating credible content recommendation.

&nbsp;

### **(2). Reviewer fb2b’s Feedback & Our Reponses**

> The reviewer raised the concern about why not use more datasets:

- As with Reviewer Vtvf, we explained that the datasets we used **are the only publicly available datasets which can be used in our task (requiring both user-item interaction data and credibility labels of content items)**.

> The reviewer raised the concern about the performance of our method under varying label ratios:

- We **provided additional experiments** demonstrating that our method consistently outperforms baselines across different label ratios.

> The reviewer raised the concern about whether the credibility labels are used to select the candidate pool and whether data leakage will occur during the data augmentation:

- We clarified that the **credibility labels are not used to select the candidate pool, thereby our evaluation is fair.** Besides, we clarified that **there is no data leakage** in the augmentation process due to the augmentation is performed independently for training and test sets.

> The reviewer suggested us to provided more analysis of one variant in ablation study:

- We provided **additional experiments showing the convergence of loss values and HR@5 metric to demonstrate that this variant actually leads to training instability** as pointed in the part of ablation study.

> The reviewer suggested us to provided more analysis of hyperparameter $w$:

- We provided **additional analysis and some practical guidance for setting $w$ on new datasets.**

&nbsp;

Although Reviewer Vtvf and Reviewer fb2b did not participate in the rebuttal, we believe our responses thoroughly address their concerns and all relevant revisions have been incorporated into the updated manuscript. We would sincerely appreciate your consideration of these points.

Thank you so much for your time and effort.

&nbsp;

Best regards,

The Authors of Submission 11137

---

### Author Response · Authors · 2025-12-03
**A summary of our rebuttal (Part 1/2)**

Dear AC,

We fully understand that due to the recent system incident, all scores were reverted and each paper was reassigned to a new AC. We sincerely appreciate your efforts in taking on the difficult task of reviewing a large number of papers within such a short period. To help minimize your workload, we provide a concise summary of our contributions and novelty as well as the score update in rebuttal.

## **A summary of the key contributions and novelty of our paper:**

### **(1). The first to study credible content recommendation under the setting of limited labels.**

- Our work is the first to study credible content recommendation under conditions where only a limited portion of credibility labels is available, reflecting a more realistic scenario where only a small part of content items are verified. This addresses an important real-world challenge that has been acknowledged and praised by all reviewers.

### **(2). The first work to empirically and theoretically demonstrate existing Diffusion-based recommender are at risk of generating uncredible recommendations.**

- Importantly, our key contribution is not the proposal of an entirely new diffusion paradigm. Rather, it lies in being the first work to empirically and theoretically demonstrate why existing DM-based recommenders are at risk of generating uncredible recommendations, which may have harmful social impacts, and in proposing a novel solution to address this practical and important issue and gap.

### **(3). We propose **Disco**, a novel method designed to enhance recommendation credibility—an important yet underexplored research topic with substantial real-world value.**

- We design a novel disentangled diffusion model that can smoothly separate preference information from uncredible information without requiring any additional disentanglement networks. In addition, we introduce a novel credible subspace projection module. Together, these two components enable the model to effectively learn credible conditions and credible diffusion targets—both of which are essential elements of diffusion-based recommender systems.
- To address the challenge posed by limited annotated data, we further develop a new progressive enhancement strategy for the credible projection module.
- Moreover, considering the characteristics of recommendation tasks, we incorporate a new preference-contrast term into the overall optimization objective.

**All these components are tightly integrated to construct a reliable diffusion framework capable of generating both accurate and credible content recommendations.**

---
&nbsp;

## **Rebuttal Score Changes**

### **Reviewer o9ud: Marginal Below (4) ⟶ Marginal Above (6)**

**Detail:** The reviewer confirmed that all concerns are well addressed and raised the score on 23 Nov (before openreview incident).

### **Reviewer 5evV: Marginal Below (4) ⟶ Marginal Above (6)**

**Detail:** The reviewer responded to us and raised another concern on 27 Nov (before openreview incident). We responded to the reviewer by providing additional theoretical analysis and experiments on 27 Nov (before openreview). The reviewer subsequently raised the score on 28 Nov (after the openrivew incident). Although the score change occurred after the incident, it was in fact a normal and timely continuation of the rebuttal exchange.

### **Reviewer Vtvf: Marginal Below (4) ⟶ No response**

### **Reviewer fb2b: Marginal Below (4) ⟶ No response**

&nbsp;


Only Reviewer o9ud and Reviewer 5evV engaged in the rebuttal. Both confirmed that their concerns were fully addressed and increased their scores accordingly. Evidence of the updated scores is available in their post-rebuttal comments.

Thank you so much for your time and effort.

&nbsp;

Best regards,

The Authors of Submission 11137

---

### Meta-Review · Area_Chair_w4xG · 2026-01-07

**Summary:**

All reviewers agree the paper targets an important and timely problem—mitigating uncredible recommendations in diffusion-based recommenders via a disentangled diffusion formulation plus a null-space projection, and they found the empirical gains promising.
The initial hesitation around acceptance was driven by:
- evaluation scope/generalizability (only three fake-news/misinformation datasets; limited evidence beyond news and content domain).
- method justification: close to prior projection ideas.
- protocol clarity and analysis gaps: candidate pool definition, leakage concerns under sequence augmentation, and insufficient diagnostics for instability observed in ablations.
- practical efficiency: dependence on large embedding dimension and many diffusion steps; multiple coupled components with unclear necessity.

After rebuttal, the authors substantially strengthened the paper with additional experiments (label-ratio sensitivity, fewer diffusion steps, low-dim settings), clarified the evaluation protocol and leakage prevention, added convergence/instability analyses, and improved method motivation/positioning. Given these improvements and the impact of the problem, I recommend Accept.

**Reviewer Concerns:**

Addressed by the rebuttal:
- Label availability sensitivity (o9ud, fb2b): Added experiments across multiple labeled-uncredible ratios (e.g., 5%–50%).
- Efficiency tradeoffs (o9ud): Added studies with fewer diffusion steps and provided evidence that performance remains competitive; also discussed sampling efficiency choices (e.g., DDIM-style acceleration) and convergence behavior.
- Low-dimensional embeddings (o9ud): Added low-dim results (e.g., 64) and analysis; clarified that DM-based methods degrade with reduced dimension, and showed Disco can remain competitive with a minor objective modification, along with a clearer efficiency–effectiveness discussion.
- Writing clarity (5evV): Reorganized and rewrote the earlier sections to define preference-aware vs uncredible-aware embeddings and tasks earlier, improving readability.
- leakage concerns (fb2b): Clarified that the candidate pool includes both credible and uncredible items; explained train/test separation and that augmentation is performed separately to avoid user leakage.
- Instability diagnosis in ablations (fb2b): Added convergence evidence to substantiate the claimed instability when removing the cosine-based term.
- Heuristic concerns for Eq. (4) subtraction and projection (5evV): Added theoretical justification and additional experiments, arguing the chosen form is effective and computationally simple.
- Credibility metrics and whether the method filters uncredible content (Vtvf): Clarified how credibility is defined and highlighted credibility-oriented metrics, plus explained the mechanism for suppressing uncredible signals.
- ID-embedding concern (Vtvf): Clarified that the method uses content embeddings rather than purely ID-level signals, reducing the conceptual mismatch about capturing nuanced credibility.
- Novelty positioning (Vtvf): Added a more explicit comparison in terms of architecture/objective and framed the contribution around credibility-aware generation under limited labels.

Still outstanding:
- Broader-domain generalization: While the authors justify dataset availability constraints, evidence remains concentrated in misinformation-style domains.
- Practical efficiency remains diffusion-dependent: Even with fewer steps / low-dim analyses, diffusion recommenders can be compute-heavy; the rebuttal mitigates but does not eliminate this concern.

**Reviewer Scores:**

Reviewer o9ud (4): expected to be 6. Key questions (low-dim, fewer steps, varying label ratios, component coupling) were directly addressed with added experiments/analysis; reviewer explicitly indicated they would update upward.
Reviewer 5evV (4): expected to be 6. Major concerns on clarity and dataset choice were resolved; reviewer explicitly stated they will raise the score.
Reviewer Vtvf (4): expected to be 6. Rebuttal addresses core clarity issues (credibility metrics, embedding type, novelty vs prior DM recommenders, null-space motivation) reasonably well; remaining concerns mainly relate to broader generalization rather than correctness.
Reviewer fb2b (4): expected to be 6. The rebuttal improves major concerns (protocol/leakage clarification, label-ratio ablation).

---

### Decision · Program_Chairs · 2026-01-26

Accept (Poster)